# Melt-quenched glass formation of a family of metal-carboxylate frameworks

Wen-Long Xue [1,2,11], Guo-Qiang Li [1,11], Hui Chen[1,3,11], Yu-Chen Han[1], Li Feng[1], Lu Wang[1], Xiao-Ling Gu[1], Si-Yuan Hu[1], Yu-Heng Deng[1], Lei Tan[4], Martin T. Dove[5], Wei Li [6] ✉, Jiangwei Zhang [7] ✉, Hongliang Dong[8], Zhiqiang Chen[8], Wei-Hua Deng[9], Gang Xu [9] ✉, Guo Wang[1] & Chong-Qing Wan [1,9,10] ✉

Metal-organic framework (MOF) glasses are an emerging class of glasses which complement traditional inorganic, organic and metallic counterparts due to their hybrid nature. Although a few zeolitic imidazolate frameworks have been made into glasses, how to melt and quench the largest subclass of MOFs, metal carboxylate frameworks, into glasses remains challenging. Here, we develop a strategy by grafting the zwitterions on the carboxylate ligands and incorporating organic acids in the framework channels to enable the glass formation. The charge delocalization of zwitterion-acid subsystem and the densely filled channels facilitate the coordination bonding mismatch and thus reduce the melting temperature. Following melt-quenching realizes the glass formation of a family of carboxylate MOFs (UiO-67, UiO-68 and DUT-5), which are usually believed to be un-meltable. Our work opens up an avenue for melt-quenching porous molecular solids into glasses.

MOFs, composed of organic ligands and metal nodes, are able to be made into glasses[1–5] via the traditional melt-quenching method[6], and the formed glasses show promising ability to overcome the fragility and processability disadvantageous for practical and specific applications of crystalline MOFs[7]. Hitherto, the metal-quenched glass formation has only been achieved for a handful of 3D zeolitic imidazolate frameworks (ZIFs)[7–12], in parallel with a few 1D and 2D metal-phosphates and metal-triazolates[13,14]. For the melting of ZIFs, the coordination bonds between the imidazolate ligands and four-coordinated metal nodes are in a dynamic equilibrium of breaking and reforming[12], which lead to defected structures and destroy the long-range order reminiscent of the scenarios in structurally similar silicates and zeolites[15,16].

Upon heating, the imidazolate linkers dissociate from bound metal nodes and re-coordinate with neighboring ones via configurational adaption in these meltable ZIFs[11,12]. This process obeys the Lindemann criterion[13,17] of melting which requires large enough amplitude of ligand and metal node displacements to exchange with their nearest neighbors[16,18]. In this context, both the high coordination number (CN) and large porosity of metal nodes are detrimental to the melting of MOFs. For the former, the highly coordinated metal node indicates the unease of bond dissociation[19]; while for the later, the large local space

[1]Beijing Key Laboratory for Optical Materials and Photonic Devices, Department of Chemistry, Capital Normal University, 100048 Beijing, China. [2]Anorganische Chemie, Fakultät für Chemie & Chemische Biologie, Technische Universität Dortmund, Otto-Hahn Straße 6, Dortmund 44227, Germany. [3]School of Chemistry and Chemical Engineering, Xi'an University of Architecture and Technology, Xi'an 710055, China. [4]Department of Physics, School of Sciences, Wuhan University of Technology, Wuhan 430070 Hubei, China. [5]College of Computer Science, Sichuan University, Chengdu 610065 Sichuan, China. [6]School of Materials Science and Engineering & Tianjin Key Laboratory of Metal and Molecule-Based Material Chemistry, Nankai University, Tianjin 300350, China. [7]College of Energy Material and Chemistry, Inner Mongolia University, Hohhot 010021, China. [8]Center for High Pressure Science and Technology Advanced Research, Pudong, Shanghai 201203, China. [9]State Key Laboratory of Structural Chemistry, and Fujian Provincial Key Laboratory of Materials and Techniques toward Hydrogen Energy, Fujian Institute of Research on the Structure of Matter, Chinese Academy of Sciences, Fuzhou, Fujian 350002, China. [10]Key Laboratory of Bioorganic Phosphorus Chemistry & Chemical Biology (Ministry of Education), Department of Chemistry, Tsinghua University, 100084 Beijing, China. [11]These authors contributed equally: Wen-Long Xue, Guo-Qiang Li, Hui Chen. ✉e-mail: wl276@nankai.edu.cn; zjw11@tsinghua.org.cn; jwz@imu.edu.cn; gxu@fjirsm.ac.cn; wancq@cnu.edu.cn

means the detached ligands and/or metal nodes may move and even fall apart to form an unstable and collapsed structure un-beneficial to vitrification[12]. Both factors can result in substantially increased energetic barriers during the dissociation of the ligands and metal nodes, leading to great difficulty in the melting of MOFs. This has been particularly evidenced in the facile melting of densely packed ZIF-4 and ZIF-zni[5,8] but un-meltable nature of porous ZIF-8[12]. In addition, porous metal-carboxylate MOFs having high CN more than 4 (6–12) are believed to be un-meltable. Considering the vast number of MOFs are metal-carboxylate frameworks, how to melt these technologically important systems with intrinsic moderate decomposition temperatures ($T_d$, ~200–500 °C)[20] remains a great challenge but is highly sought after[21]. Several carboxylate MOFs, including both carboxylates and bicarboxylates, have been made into glasses very recently, which are derived from discrete solvated complexes or coordination networks through rearrangements of coordination bonds upon desolvation[22–24]. Taking advantage of the flexibility and low symmetry of the aliphatic carboxylate ligands and the lack of crystal field stabilization energy on metal ions, a selection of glasses comprising low oxidation state metals ($Mg^{2+}$, $Mn^{2+}$) and flexible adipate were reported very recently[25]. However, this strategy is not applicable to those carboxylate MOFs bearing with high valence metal ions.

Herein, we propose a feasible strategy to overcome this challenge by covalently bonding zwitterionic groups on the rigid aromatic carboxylate ligands and incorporating the Brønsted acids in the framework channels. On the one hand, the addition of zwitterionic groups and Brønsted acids significantly decrease the porosity, and their strong interactions on the interface reduce the coordination bonding strengths of ligands via charge delocalization[26] (Supplementary Fig. 1). On the other hand, both components exhibit high structural degrees of freedom which substantially increases the configurational entropy and hence reduce the glass transition ($T_g$) and melting temperatures ($T_m$)[27]. UiO-67, UiO-68 and DUT-5 are exemplified because they are prototypical MOFs with high $T_d$ and are commonly believed to be un-meltable due to their ultra-strong Zr-O or Al-O bonds, as well as high CN numbers (6–8) and large porosity[1,20]. Meanwhile, the un-meltable counterparts ILs@UiO-67 were also synthesized as references to show that such a strategy is in stark contrast to the reported guest-host interactions scenario for the melt of ILs@ZIF-8[28].

## Results

UiO-67 is assembled by BPDC ligands and $Zr_6O_4(OH)_4$ clusters in a 12-connected manner (BPDC = biphenyl-4,4′-dicarboxylate)[29]. By substituting a zwitterionic group MIMS (MIMS = (1-methyl-3-imidazolio) propane-3-sulfonate), on the 2-position of BDPC, a reticularly modified analog ZW·UiO-67 crystallized in a relative low symmetric space group $R3$ was obtained. The crystal structure of ZW·UiO-67 was constructed theoretically, and optimized using experimental cell parameters obtained from powder X-ray diffraction (PXRD). The stoichiometric formula of $Zr_6O_4(OH)_4(BPDC\text{-}MIMS)_{5.4}$ is determined via $^1H$ NMR spectra, with a small portion of BPDC-MIMS ligand displaced by the acetate group. This compound is stable even in boiling water and intact in aqueous solution with a wide range of pH values (Fig. 1, Supplementary Figs. 1–4). By incorporating Brønsted acids equal to the total pore volume of the desolvated ZW·UiO-67, a family of ZW·UiO-67·HA frameworks (HA = MSA, TFSA, TFA; MSA = methylsulfonic acid, TFSA = N,N-bis(trifluoromethanesulfonyl) amide, TFA = trifluoromethanesulfonic acid) were obtained. The PXRD patterns confirm ZW·UiO-67·HA are isostructural to the parent UiO-67 (Supplementary Fig. 4). Notably, the ZW·UiO-67·HA family frameworks are fully meltable ($T_m > 120$ °C, Fig. 2a, Supplementary Figs. 5, 6, Supplementary Videos 1–3), which are in stark contrast with the complete unmeltable counterpart ILs@UiO-67 (Supplementary Figs. 1, 8, 9, Supplementary Table 2).

ZW·UiO-67·MSA with a stoichiometric formula of $Zr_6O_4(OH)_4(BPDC\text{-}MIMS)_{5.4}·(O_2CCH_3)_{1.2}·(MSA)_{20.5}$ melted as

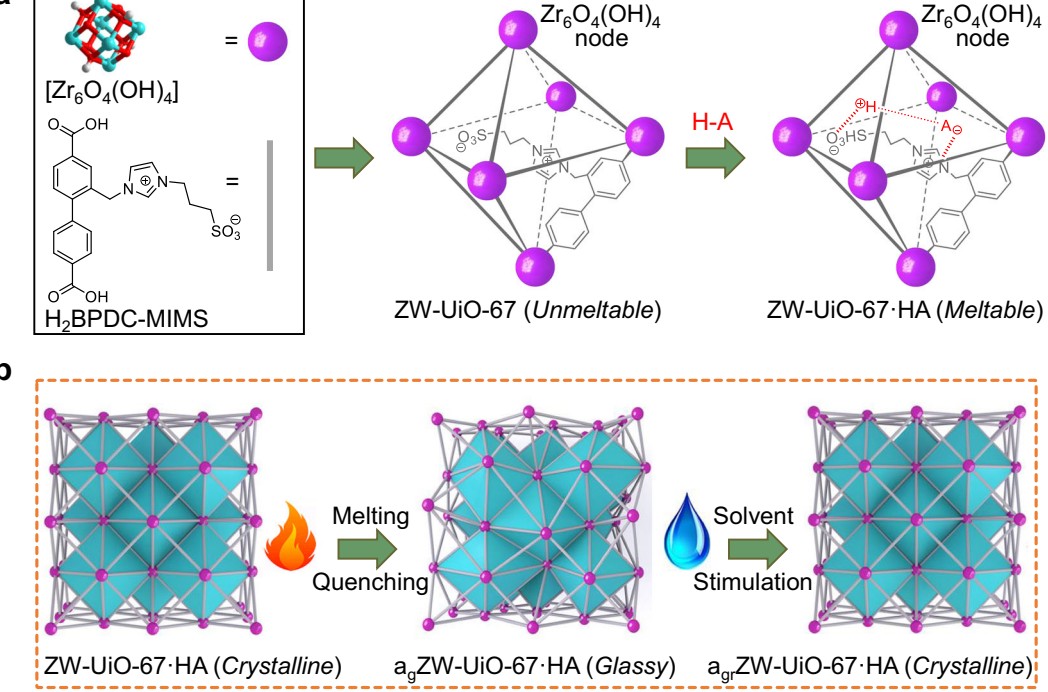

**Fig. 1 | Melt-quenched glass formation of ZW·UiO-67·HA. a** Schematic synthesis of ZW·UiO-67 and ZW·UiO-67·HA. (HA = MSA, TFSA, TFA; MSA = methylsulfonic acid, TFSA = N,N-bis(trifluoromethanesulfonyl) amide, TFA = trifluoromethanesulfonic acid). **b** Transition of crystalline ZW·UiO-67·HA to glassy $a_g$ZW·UiO-67·HA, then back to crystalline $a_{gr}$ZW·UiO-67·HA through melting, quenching and solvent stimulation processes. The purple sphere and gray bond denote the $Zr_6O_4(OH)_4$ node and BPDC-MIMS linker (BPDC = biphenyl-4,4′-dicarboxylate; MIMS = (1-methyl-3-imidazolio) propane-3-sulfonate), respectively. Parent UiO-67 assembled by BPDC is shown in Supplementary Fig. 1 for comparison.

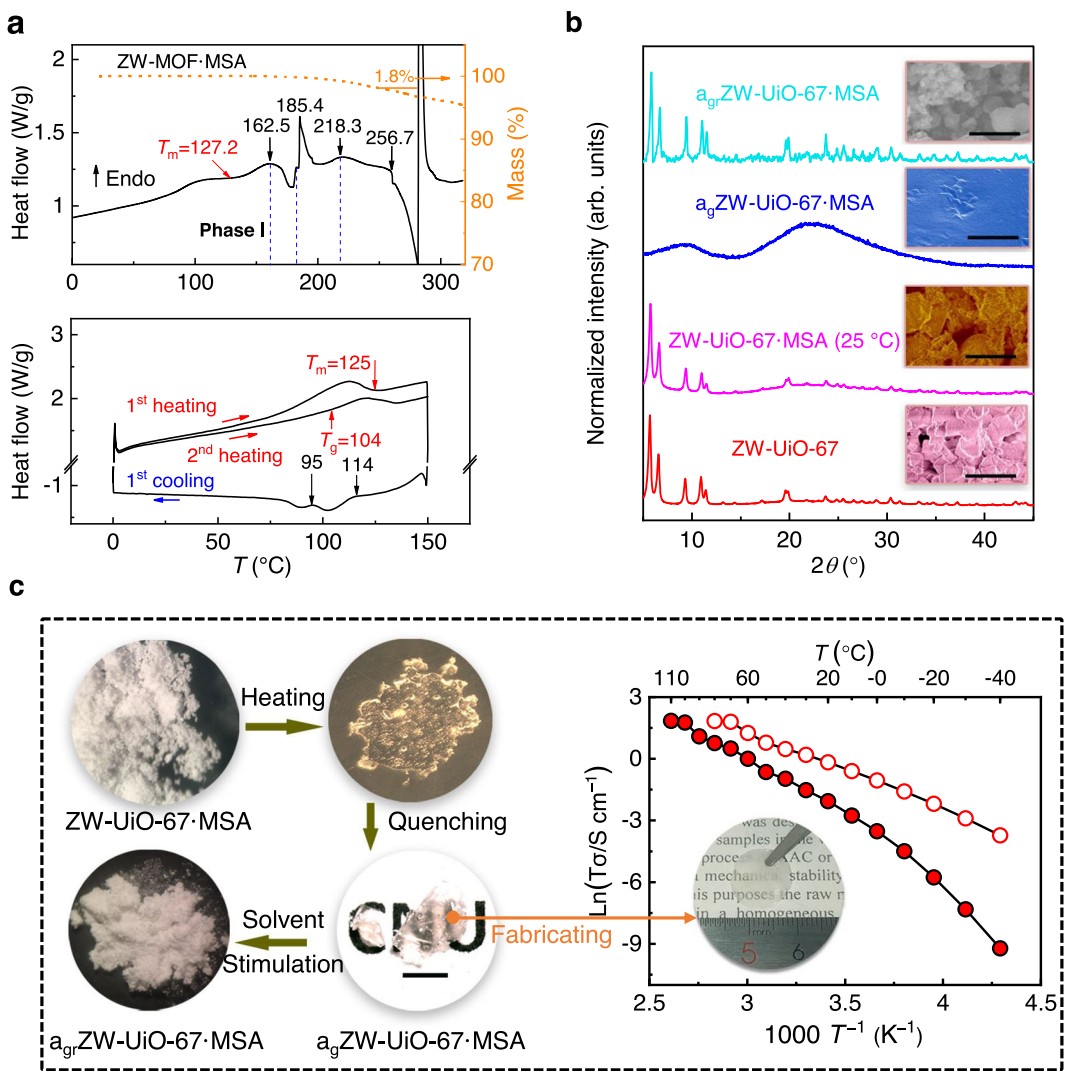

**Fig. 2 | Thermal dynamics of ZW-UiO-67·MSA. a** TGA plot (yellow), DSC curve of ZW-UiO-67·MSA from 0 to 320 °C (top) and cyclic curve from 0 to 150 °C (bottom). The slight difference of $T_m$ is due to the different measurement batches. **b** PXRD patterns of ZW-UiO-67, ZW-UiO-67·MSA, $a_g$ZW-UiO-67·MSA and $a_{gr}$ZW-UiO-67·MSA with inset SEM images (Supplementary Figs. 4, 5, and 10). Bar scale: 2 μm. **c** Microscopical images of the melted, quenched and solvent stimulated products of $a_{gr}$ZW-UiO-67·MSA, and temperature-dependent proton conductivities of $a_g$ZW-UiO-67·MSA glass sheet (red) and ZW-UiO-67·MSA (white).

temperature over 120 °C (Supplementary Video 1, Supplementary Figs. 5, 8, and 11). This $T_m$ is the lowest for known 3D MOF glasses, which is about 250 and 80 °C lower than those of ZIF-62 (370 °C)[9] and a few 1D or 2D metal-phosphates and metal-triazolates (~200 °C), respectively[13,14]. The first broad endothermal peak with no weight loss was recorded in the DSC curve (Fig. 2a), showing a $T_m$ offset temperature around 127 °C, accompanied with an appearance of a flow like liquid (Supplementary Video 1). The melting of ZW-UiO-67·MSA also was confirmed via monitoring the change of the morphology on macroscale and the gradually vanished Bragg diffractions in the PXRD patterns from 25 to 150 °C (Supplementary Figs. 8a and 11). Notably, no melting or evaporating event of MSA ($T_m ≈ 20$ °C, bp ≈ 167 °C) was recorded in the DSC and TGA trace before 180 °C. No loss of MSA implies that the Brønsted acid-zwitterion subsystem MIMS·MSA has been successfully incorporated in ZW-UiO-67·MSA (Supplementary Fig. 1). This is similar to the scenario for the zwitterion-based acidic ionic liquids[26] including H₂BPDC-MIMS·MSA without a thermal event of a separate specie upon heating (Supplementary Fig. 6d). Therefore, the behavior of ZW-UiO-67·MSA is in stark contrast to the thermal properties of neat ZW-UiO-67 or MSA alone. It can be viewed as a solid ionic liquid[30]

featuring the Brønsted acid-zwitterion structure, but not a simple host-guest composite like those ILs@MOFs[28].

For the DSC plot within 0 ~ 320 °C, the first feature of the endo-thermal melting (offset ~127 °C) is broad and weak, which could be related to the lattice fluctuation with multiple and gradual enthalpy changes, being similar to the melts of some porous organic cages with flexible structures upon heating[31]. The measured fusion enthalpy change of 6.35 J/g is also comparable to those of 2D metal-triazolates (5–10 kJ/mol)[14], zeolitic zinc 2-ehtylimidazolate[32] and some silica zeolites[33]. The following exothermal event with onset at 162.5 °C (without weight loss) may be ascribed to polyamorphic phase transitions between distinct structural configurations of neighbor states with different potential energies (Fig. 2a, Supplementary Fig. 5). Therefore, the melting phase of ZW-UiO-67·MSA before 162.5 °C can be regarded as a stable liquid phase I. At 185.4 °C, the endothermal sharp peak closely relates to that event for ZW-UiO-67 at 180.0 °C and ZW-UiO-67·0.5MSA at 186.2 °C, respectively (Supplementary Fig. 5). Considering the formula mass percentage of acetate (⁻OOCCH₃ wt% = 1.4% in ZW-UiO-67·MSA) is close to the ~1.8 wt% weight loss recorded before 250 °C, it could be speculated that the endothermal event without weight loss at 185.4 °C may relate to the de-association of the acetate

group which is followed by less composite decomposition after that temperature. Online gas-phase FT-IR spectra demonstrate that $H_2O$ and $CO_2$ can be detected with the temperature increase (25–320 °C) (Methods and Supplementary Fig. 25).

For clarity, the quenched product from 130 °C (stable phase I, over $T_m$), a transparent solid without any Bragg diffraction, is denoted as $a_g$ZW-UiO-67·MSA ($a_g$ means *amorphous-glass*) for further discussion. The SEM image of the surface of $a_g$ZW-UiO-67·MSA displays a smooth and crack-free texture, markedly different from that before melting (Fig. 2b, Supplementary Fig. 10). Polarized light microscopical image shows no spatially dependent extinction, endorsing its isotropic glassy nature (Supplementary Fig. 13). Cyclic DSC curves of ZW-UiO-67·MSA (within 0 ~ 150 °C) show two exothermal events respectively at 114 and 95 °C during cooling (Fig. 2a, Supplementary Fig. 5b), while a $T_g$ onset appears at 104 °C in the following up-scan. No Bragg diffraction found for $a_g$ZW-UiO-67·MSA signifies such exothermal events could arise from phase transitions, rather than a recrystallization during cooling. Different flowing gas and/or heating rates exhibited slight influence on $T_m$, with the value varying in a narrow range (127 °C–132 °C), but distinct heating rates led to $T_g$ lying within a relative larger region (73 °C–104 °C) for ZW-UiO-67·MSA (Supplementary Fig 7).

Interestingly, $a_g$ZW-UiO-67·MSA gradually recovered to the crystalline phase, $a_{gr}$ZW-UiO-67·MSA ($a_{gr}$ means *amorphous-glass recovery*), upon solvent stimulation (Method). $a_g$ZW-UiO-67·MSA soaked in methanol for 3 min at room temperature only show broad diffractions around (20-1), (202) and (20-4); while further reflux engendered a powdery solid displaying the PXRD pattern identical to that of the parent ZW-UiO-67 (Fig. 2b, Supplementary Fig. 4a). The recovered surface area of such $a_{gr}$ZW-UiO-67·MSA is ~86% of that for ZW-UiO-67, with a slightly broadened pore-size distribution (~8–12 Å, Supplementary Fig. 14, Supplementary Tables 1-2). Elemental analysis demonstrates ~60% MSA was preserved within $a_{gr}$ZW-UiO-67·MSA, with the rest released during reflux. Such embedded MSA and the relative low crystallinity resulting from the solvent stimulation accounts for the reduced porosity. Furthermore, the glass to crystalline phase transformation upon external stimulation can be extended to other kinds of solvents, and polar solvents are advantageous for this process (Supplementary Fig. 15). By exposing to methanol vapor for 30 min, the X-ray diffraction peaks at low angles appeared with a broadened feature. Therefore, $a_g$ZW-UiO-67·MSA is in a metastable amorphous state that is structurally recoverable with a moderate stimulation, and explicitly shows its preserved continuous random network (CRN) structure inherited from the parent MOF[8]. Re-heating and quenching of $a_g$ZW-UiO-67·MSA produced transparent glasses with large lateral sizes. The anhydrous proton conductivity of $a_g$ZW-UiO-67·MSA reaches $1.57 \times 10^{-2}$ Scm$^{-1}$ at 100 °C, acting as a fast ionic conductor with application prospect[30]. Almost one order-of-magnitude increase in conduction from $1.96 \times 10^{-3}$ to $1.65 \times 10^{-2}$ Scm$^{-1}$ was observed as the temperature elevated from 80 to 110 °C. Such a marked change in conduction correlates to the glass transition near $T_g$ (>$T_g$, 104 °C), reminiscent of that observed for the phase transition in reported solid state conductors[34]. Compared to ZW-UiO-67·MSA, the slightly lower ion conduction and non-linear Arrhenius plot signify the largely disordered structure of $a_g$ZW-UiO-67·MSA (Fig. 2c).

The morphologies of ZW-UiO-67 and ZW-UiO-67·0.5MSA remained almost intact even at 300 °C (Supplementary Fig. 9). The preserved PXRD patterns of ZW-UiO-67·0.5MSA and the vanished peaks of ZW-UiO-67·MSA above 150 °C explicitly show their different thermal behaviors, and sufficient MSA molecules are responsible for the meltability of ZW-UiO-67 ($T_d \approx 340$ °C, Supplementary Fig. 16). Here ZW-UiO-67 can be regarded as a zwitterion-decorated MOF with the MIMS groups, and the incorporated MSA pairs with the MIMS group to form a Brønsted acid-base buffer subsystem of MIMS·MSA similar to the scenario of acidic ionic liquids as evidenced by the above

DSC results[26]. The more MSA molecules pair with the zwitterionic ligands through electrostatic and/or hydrogen-bonding interactions on the interface, the greater charge delocalization and separation occur. This substantially disrupts the long range structural order and consequently leads to increased configurational entropy ($S_{conf}$), which dramatically reduces the $T_m$[26,27] and facilitates the solid-to-liquid phase transition, resulting in the melting of ZW-UiO-67·MSA at ~127 °C but unmeltable ZW-UiO-67·0.5MSA due to the insufficient acids incorporated. In this context, the solid zwitterionic EIMS and $H_2$BPDC-MIMS respectively give rise to room temperature acidic ionic liquids EIMS·MSA and $H_2$BPDC-MIMS·MSA as the amount of MSA is 3.78 times that of the zwitterions, while a solid product was observed when half amount of MSA (1.89 times) was incorporated[30] (Supplementary Fig. 1 and Supplementary Table 2). Further theoretical calculations reveal the coordination ability of the carboxylate ligands is weakened as BPDC-MIMS paired with more nearest MSA molecules. The negative charges of four $O_{COO}$ atoms change from −0.64, −0.64, −0.56 and −0.59|e| to −0.62, −0.53, −0.52 and −0.61|e| due to the charge delocalization and intermolecular interactions. It could be speculated the weakened Zr-O bonds may trigger the bond dissociation and atom displacements under the assistance of the acid-zwitterion buffer. However, the densely filled channels in ZW-UiO-67·MSA not only are beneficial for the coordination bonding mismatch between COO$^-$ and metal nodes upon heating, but also enhance the structural disorder and increase the configurational entropy to markedly reduce the energy barrier required for the melting. This assumption is also supported by the extended Zr-O$_{COO}$ distances due to the O$_{COO}$ protonation and temperature increase revealed by the ab initio molecular dynamics (AIMD) simulations (Supplementary Fig. 18). Furthermore, the control ILs@UiO-67 containing the acidic ionic-liquid guests, i.e. counterpart EIMS·MSA@UiO-67 (EIMS = 3-(1-ethyl-3-imidazolium)propane-3-sulfonate) with the molar ratio (EIMS/MSA = 1:3.78) same as that of MIMS/MSA in ZW-UiO-67·MSA (Supplementary Fig. 1, Supplementary Table 2), which remained its crystalline morphology intact at ~300 °C and whereupon thermal decomposition occurred around 355 °C (Supplementary Figs. 9 and 16). This confirms the covalently-bonded MIMS·MSA subsystem is the main cause for reducing the $T_m$ of ZW-UiO-67·MSA.

To track the local structural changes before and after the glass formation, X-ray absorption near edge structure (XANES) spectra of the Zr K edge were collected at ambient conditions. Pre-edge features of ZW-UiO-67, ZW-UiO-67·MSA and $a_g$ZW-UiO-67·MSA resemble that of the UiO-67 parent, indicating that the 8-coordinated Zr$^{4+}$ ions [4 Zr-bridging O/OH (Zr–O$_{\mu3-O}$) and 4 Zr–carboxylate (Zr–O$_{COO}$)][35] existed in $a_g$ZW-UiO-67·MSA. All peaks at about 18,020 eV can be assigned to the dipole-allowed 1s-5p transition (Fig. 3a, b inset). ZW-UiO-67 exhibits a slightly decreased absorption energy compared to that of UiO-67, implying a reduced local symmetry of the 8-coordinated ZrO$_8$ induced[36] by attaching large MIMS groups on BPDC ligands. Similar phenomenon was also found for $a_g$ZW-UiO-67·MSA as compared to ZW-UiO-67·MSA. The decreased local symmetry is consistent with the relative low symmetric R3 space group of ZW-UiO-67 (vs. *Fm-3m* of UiO-67[37]) and increased intensity at 18,003 eV of the dipole-forbidden 1s-4d transition, suggesting a more mixed 4d-5p state. Therefore, the Zr$_6$O$_4$(OH)$_4$ octahedra in ZW-UiO-67, ZW-UiO-67·MSA and $a_g$ZW-UiO-67·MSA are distorted with respect to the perfect geometry in UiO-67. In extended X-ray absorption fine structure (EXAFS) spectra, two major peaks of 1.50–2.50 and 2.90–4.00 Å (phase corrected distances) regions respectively corresponding to the Zr-O and Zr···Zr (Zr-O-Zr, 3.35 Å) scatter contributions are observed. Compared to UiO-67, ZW-UiO-67 and ZW-UiO-67·MSA, scattering path of Zr-O in $a_g$ZW-UiO-67·MSA is enlarged with the slightly weakened intensity. Broaden and weakened Zr···Zr scattering paths in ZW-UiO-67 and ZW-UiO-67·MSA are observed with respect to that of UiO-67, while this phenomenon is more pronounced in $a_g$ZW-UiO-67·MSA with two split peaks at 3.26

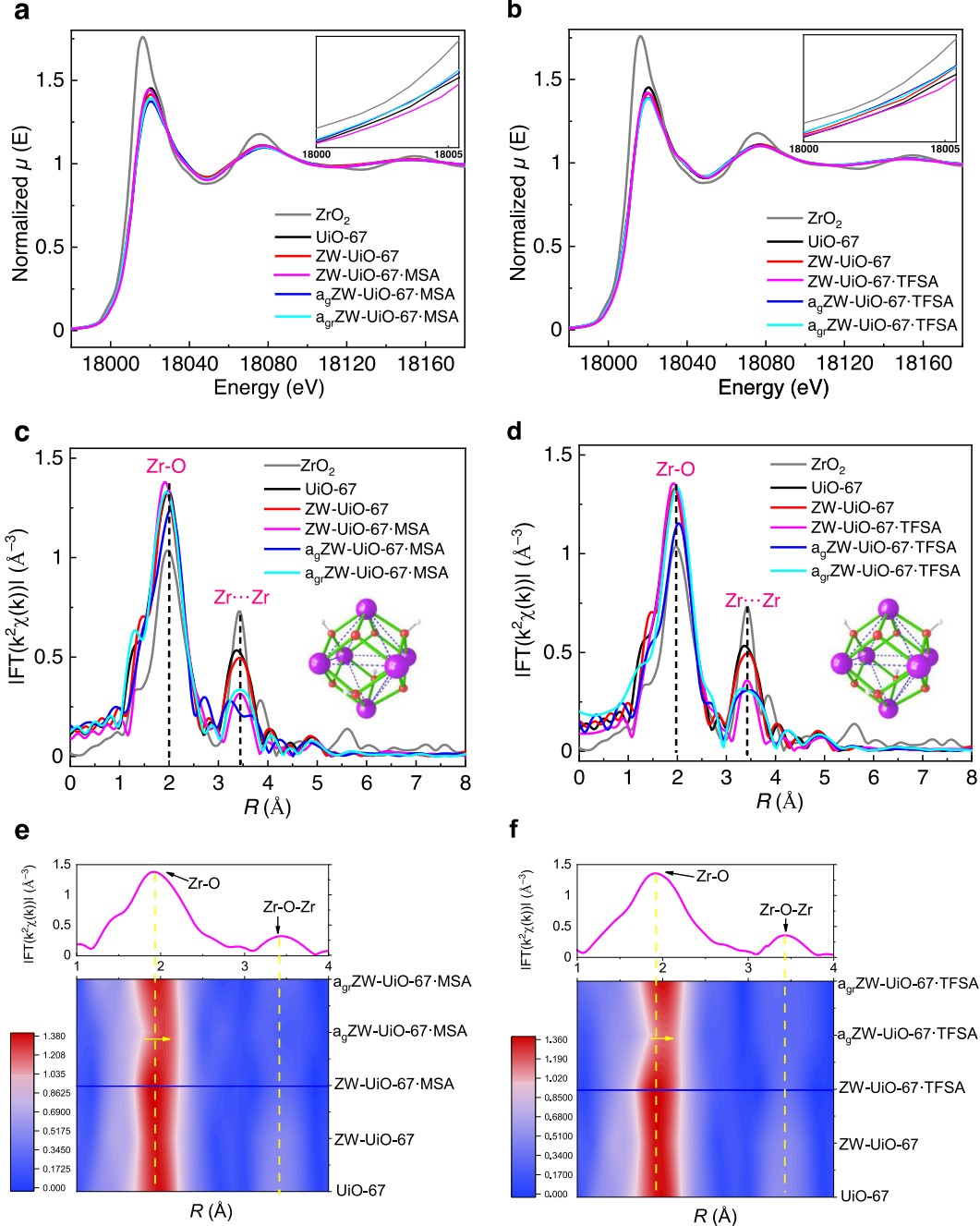

**Fig. 3 | Local structures of ZW-UiO-67·HA before and after the glass formation.**
**a** XANES spectra at the K edge of the Zr atoms for derivates of ZW-UiO-67·MSA and references. **c** Corresponding radial distance χ(R) space spectra from $k^3$-weighted Fourier transform of χ(k)-function from the Zr Kedge EXAFS. Comparison spectra for ZW-UiO-67·TFSA and derivates are shown in (**b**) and (**d**). Insets in (**c**, **d**) show the octahedral $Zr_6O_4(OH)_4$ node. Color codes: Zr, purple; O, red; H, light gray. **e**, **f** Quantitative χ(R) space 2D profiles comparison of before and after vitrificated

ZW-UiO-67·MSA/TFSA (also see Supplementary Figs. 20 and 21). The color scale represents the CN of Zr with different coordination shell during the phase transition. Data of UiO-67 is taken from literature[37]. The yellow dashed lines indicate the radial distance peak position of ZW-UiO-67·HA and the black arrow clarifies the peak type. The yellow arrows provide eye guidance for the shift of the Zr-O distance.

and 3.77 Å. Therefore, $a_gZW$-UiO-67·MSA features the distorted octahedral $Zr_6O_4(OH)_4$ nodes, lengthened Zr-O bonds with partial breakage. Compared the glass to the $a_{gr}ZW$-UiO-67·MSA, the lengthened Zr-O bond recovered with comparable intensity to the pristine state, but the $Zr_6O_4(OH)_4$ node retained during such crystal-glass-crystal transformation. Such cluster changes are also intuitively witnessed by the Quantitative χ(R) space spectra fitting, wavelet transform extended X-ray absorption fine structure (WTEXAFS) and reduced coordination number (CN) of Zr-O bond (7.01–7.72) and Zr···O···Zr scattering path

(2.95–3.18) in $a_gZW$-UiO-67·MSA and $a_{gr}ZW$-UiO-67·MSA as compared the perfect UiO-67 (8, 4) (Fig. 3e, f, Supplementary Figs. 19–21, Supplementary Table 3).

X-ray total scattering experiments reveal similar pair distribution function (PDF) $D(r)$ features below 8 Å for ZW-UiO-67·MSA, $a_gZW$-UiO-67·MSA and $a_{gr}ZW$-UiO-67·MSA, suggesting their similar $Zr_6O_4(OH)_4$ clusters reminiscent of that observed by XANES (Fig. 4). Experimental PDF plots were assigned and compared through the data calculated from UiO-67 and ZW-UiO-67 crystal structures

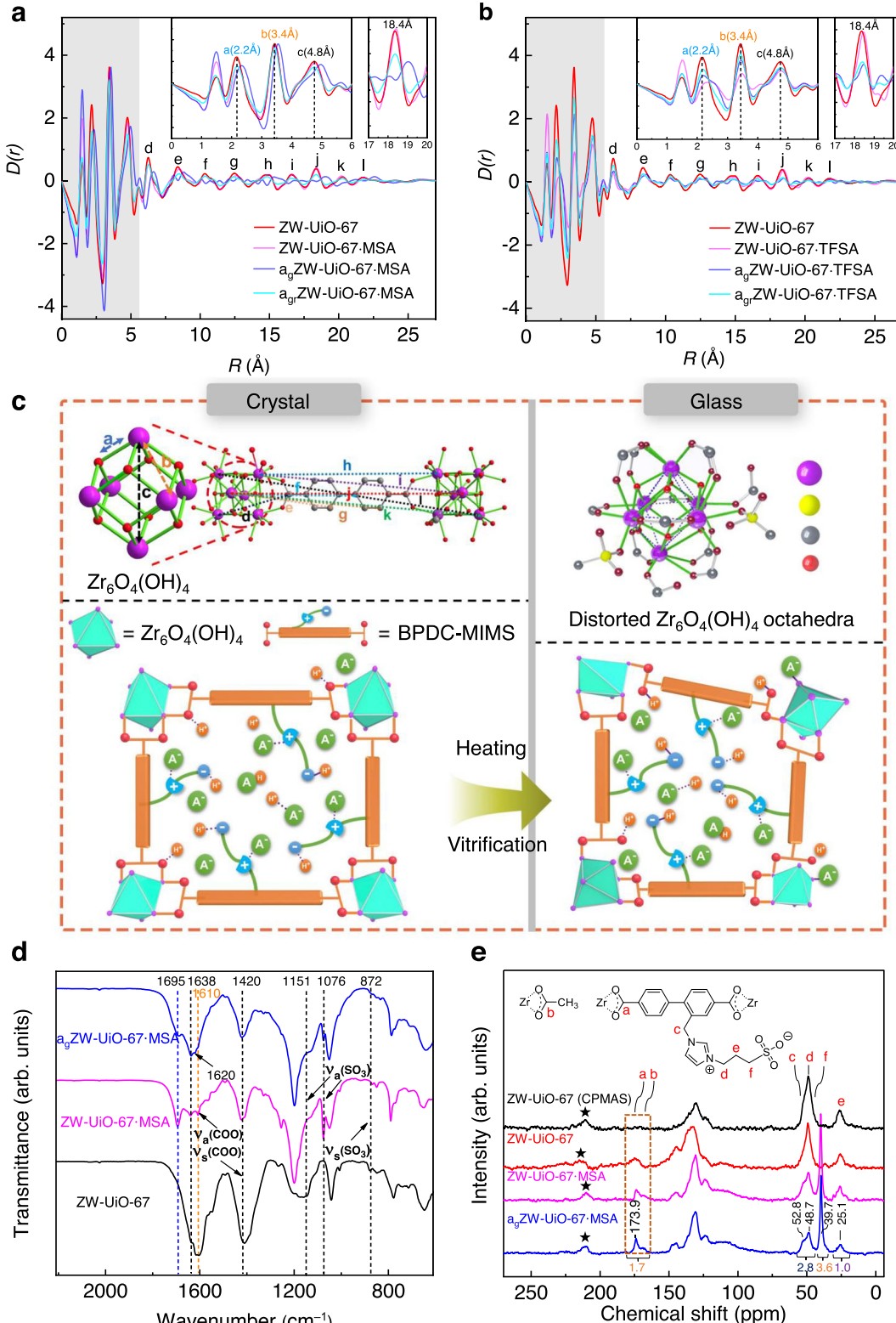

**Fig. 4 | Melt-quenched glass structure and formation mechanism of ZW-UiO-67·MSA. a** PDF $D(r)$ for Zr-O and Zr···Zr distances ($r$ range <6 Å, gray region) and other atom to atom correlations (marked as $a - l$ corresponding to (**c**) panel) in ZW-UiO-67, ZW-UiO-67·MSA, $a_g$ZW-UiO-67·MSA and $a_{gr}$ZW-UiO-67·MSA. **b** Comparison spectra of ZW-UiO-67·TFSA. **c** Proposed microscopic mechanism of ZW-UiO-67·MSA from crystalline to glassy state upon heating. The top left shows the perfect $Zr_6O_4(OH)_4$ octahedra in UiO-67[37], with the atom-to-atom correlations labeled from $a$ to $l$ corresponding to those in (**a**, **b**). Top right presents the distorted $Zr_6O_4(OH)_4$ octahedra within $a_g$ZW-UiO-67·MSA. **d** FT-IR spectra of ZW-UiO-67, ZW-UiO-67·MSA and $a_g$ ZW-UiO-67·MSA, respectively (Supplementary Fig. 24). **e** [1]H-[13]C CP MAS NMR of ZW-UiO-67 (top), and DD MAS of ZW-UiO-67, ZW-UiO-67·MSA and $a_g$ZW-UiO-67·MSA (25 °C, 12 kHz), with adamantane ([13]C, d = 29.5 ppm) as the first reference. Cyclized area shows the carboxylate COO resonance. Spinning sidebands are marked with asterisks (Supplementary Fig. 26).

(Supplementary Fig. 22, Method). The Zr-O bond (-2.2 Å), Zr···Zr distances (-3.4 Å, Zr-O-Zr) and octahedral diagonal (~4.8 Å) of $Zr_6O_4(OH)_4$ were observed with relative weakened intensities compared with those of ZW-UiO-67. For $a_gZW$-UiO-67·MSA, a small extension of Zr-O bond (from 2.2 to 2.3 Å) accompanied by broad peak was observed for the coordination shell of Zr (2.2 Å), indicating the altered Zr-O bonds. So, the second Zr···Zr correlation peak (4.8 Å) extends (to 4.95 Å) and broads due to the distorted $Zr_6O_4(OH)_4$ in the glass. In addition, the broad and weak d (6.6 Å) (marked in Fig. 4c), weakened and split e (8.1, 8.9 Å), f (10.0,10.7 Å) and g (12.1, 12.8 Å) peaks that correlate to Zr···C distances from Zr on distorted $Zr_6$ to the C on linker BPDC-MIMS are observed, which originated from the broken/lengthened $Zr$-$O_{COO}$ bonds as evidenced by the EXAFS and AIMD simulations. Beyond the 15 Å range, Zr···Zr correction peaks (h-l, 15.0 - 22.5 Å) between two $Zr_6$ clusters are weakened with split and/or extended features (Supplementary Table 4). Combined with the preserved $Zr_6$ structure, a continuous random network (CRN) is presented in aZW-UiO-67·MSA. The same differential PDF[38] plots between $a_gZW$-UiO-67·MSA and ZW-UiO-67 or ZW-UiO-67·MSA but strong and broad Zr-O (Zr-$O_{\mu3\text{-}O}$ or Zr-$O_{COO}$) (2.5 Å) and Zr···Zr (Zr-O-Zr) (3.7 Å) peaks after vitrification indicate the preserved but distorted network in $a_gZW$-UiO-67·MSA. Also, the vanished feature ~6.2 Å infers the enlarged Zr···$C_{COO}$ (closest C to Zr-O cluster) and disordered structure upon vitrification (Supplementary Fig. 23). Intriguingly, in $a_{gr}ZW$-UiO-67·MSA, the main peak features (a-k) within <20 Å recover and resemble to those in ZW-UiO-67 (Figs. 4a and 2b).

FT-IR spectra showed a blue-shifted band (C=O) at 1695 $cm^{-1}$ for ZW-UiO-67·MSA compared to ZW-UiO-67, which could be ascribed to the protonated $O_{COO}$ atoms by MSA. In $a_gZW$-UiO-67·MSA, the $v_a$(COO) vibration at 1638 $cm^{-1}$ became significantly broadened and a shoulder band at 1620 $cm^{-1}$ appeared, suggesting the asymmetric stretch of monodentate coordinated COO group (Fig. 4d) dissociated from the *syn-syn-$\mu_2$* bridging mode reminiscent of that in the compressed UiO-66[39]. Solid-state $^{13}C$ NMR spectra with the dipolar dephasing magic-angle-spinning (DD MAS) mode revealed ZW-UiO-67, ZW-UiO-67·MSA and $a_gZW$-UiO-67·MSA featuring almost unchanged COO resonance around 173.9 ppm and aromatic carbon resonances from 120 to 150 ppm (Fig. 4e); meanwhile, more sensitive $^1H$-$^{13}C$ cross polarization magic-angle-spinning (CP MAS) spectra of $a_gZW$-UiO-67·MSA show comparable features to those of unmeltable ZW-UiO-67·0.5MSA, but a slightly different resonance at ~182 ppm (Supplementary Fig. 26b) which implies much small amount of uncoordinated COO groups[36,40] due to that solid-liquid phase transition. In addition, the missed COO resonance (~173 ppm) of CP MAS in ZW-UiO-67 (Fig. 4e) re-appears in $a_gZW$-UiO-67·MSA (Supplementary Fig. 26b), consistent with the protonation of COO groups[41] as suggested from the FT-IR, EXAFS, PDF and AIMD calculations mentioned above: some $Zr$-$O_{COO}$ bonds of the *syn-syn-$\mu_2$* COO bridges were broken, and the local MSA molecules subsequently occupied the vacant sites left by the detached $O_{COO}$ atoms from the $Zr_6O_4(OH)_4$ during melting (Fig. 4c); while, after quenching the $Zr$-$O_{COO}$ bonds reformed and left some monodentate COO groups within the CRN structure of $a_gZW$-UiO-67·MSA. Variable temperature in-situ FT-IR data of ZW-UiO-67·MSA show the gradually broadening and shifting of C=O band from 1693 at room temperature to 1713 $cm^{-1}$ at 130 °C. However, the ~1693 $cm^{-1}$ band feature remained in the melting ZW-UiO-67·MSA and the melt-quenched glass $a_gZW$-UiO-67·MSA (Supplementary Fig. 27). This phenomenon further indicates the coordination network in liquids state which quenched to form the glass structure as mentioned above.

The melting phenomena were successfully extended to ZW-UiO-67·TFSA and ZW-UiO-67·TFA with $T_m$ (offset) of -153 and -157 °C, respectively, displaying comparable fusion enthalpy lying within 2.12–8.65 J/g range (Supplementary Figs. 6, 12, and 13,

Supplementary Tables 5 and 6). Such a slight thermal fluctuation is similar to the almost undetectable configurational heat capacity in some very strong liquids, such as low density amorphous water[42], amorphous Si[43], coordination polymers[14,32] and zeolitic silica[33], indicating a unique low entropy state[44]. Similar glassy $a_gZW$-MOF·TFSA and $a_gZW$-UiO-67·TFA were obtained by melt-quenching at 155 °C and 160 °C, respectively, and according $a_{gr}ZW$-UiO-67·TFSA and $a_{gr}ZW$-UiO-67·TFA also were observed (Supplementary Figs. 4c, 10, 12, and 14). No clear peak was observed in the DSC curve during the cooling down scan, which indicates no phase transition occurs due to the high viscosity of molten sample, similar to that found in plastic crystals[14]. Changes in FT-IR, $^{13}C$ NMR, XANES, EXAFS, PDF, Quantitative χ(R) space spectra fitting and WTEXAFS data of ZW-UiO-67·TFSA before and after vitrification are reminiscent of those of ZW-UiO-67·MSA (Figs. 3 and 4, Supplementary Figs. 20–24, 28 and Supplementary Tables 3 and 4). Small angle X-ray scattering (SAXS) experiments of $a_gZW$-UiO-67·HA showed no scattering comparable to polycarbonate and silicate glasses within the q range from 0.025 to 0.15 $Å^{-1}$, meaning a homogeneous state without fragments/nanoparticles ranging from several nanometers to 25 nm[45]. The SAXS signal $I(q)$ of $a_gZW$-UiO-67·HA crucially continues, rather than that of a monolith blended with nanoparticles[46]. In addition, no detectable wide angle X-ray scattering (WAXS) peaks and the missing electron diffraction in TEM images were recorded for $a_gZW$-UiO-67·HA, exhibiting comparable amorphous features to those of polycarbonate and silicate glasses (Supplementary Figs. 28 and 29).

To exclude the super acid effect on the melting, weak ethanesulfonic acid (ESA, *p*Ka = 1.8) was tried (vs. MSA, TFSA, TFA, *p*Ka: −1.9 ~ −14). As expected, similarly melting ZW-UiO-67·ESA ($T_m$ = 139 °C, $T_g$ of 82 °C, Supplementary Video 4) and recoverable $a_gZW$-UiO-67·ESA to $a_{gr}ZW$-UiO-67·ESA under solvent stimulation were observed (Supplementary Fig. 30); however, all counterpart HA@UiO-67 retained their morphologies even at 320 °C (HA = MSA, TFSA, TFA, ESA), endorsing the acid and even super acid cannot realize the melt of UiO-67 without the functional zwitterions. In addition, controlled ILs@ZW-UiO-67 and ILs@UiO-67 are completely unmeltable even at 320 °C (ILs = EMIM·TFS, EIMS·MSA) (Supplementary Fig. 31, Supplementary Table 5). Therefore, bulk ILs also cannot account for the melt of ZW-UiO-67. This is completely different from the scenario found in ILs@-ZIF-8[28], wherein the interactions between ZIF-8 and ILs weaken the coordination bonds and facilitate the melt[47]. More importantly, successful samples can extend to other metal-carboxylate frameworks such as ZW-DUT-5 bearing the mixed linkers BPDC-MIMS/BPDC (4:1 molar ratio), a zwitterion (MIMS) modified isostructure of DUT-5 ($T_d$ = 450 °C)[48], and ZW-UiO-68. ZW-DUT-5·MSA ($T_m$ ~ 152 °C), ZW-UiO-68·MSA ($T_m$ ~ 156 °C), $a_gZW$-DUT-5·MSA, $a_gZW$-UiO-68·MSA, $a_{gr}ZW$-DUT-5·MSA and $a_{gr}ZW$-UiO-68·MSA analogs were obtained (Supplementary Figs. 32–35, Supplementary Table 6, Supplementary Videos 5 and 6). Notably, the partial zwitterion-modified linkers in ZW-DUT-5 is sufficient to enable the melting, further validating the feasibility of this strategy.

## Discussion

By functionalizing the carboxylate ligands with zwitterionic groups and incorporating the Brønsted acids in the framework channels, a family of metal-carboxylate frameworks (UiO-67, UiO-68 and DUT-5) were successfully made into glasses via the melt-quenched method. Importantly, our strategy could be applied to a range of highly porous MOFs composed of different ligands and metal nodes. In addition, this strategy could also be extended to other kinds of highly porous molecular frameworks such as covalent-organic frameworks. Our findings would also offer opportunities for processing MOFs and other hybrid organic-inorganic crystals by melt-quenching them into glasses, which could enable them to meet the manufacturing requirement by industry.

## Methods

### General materials and instruments

Variable-temperature PXRD of ZW-UiO-67·MSA and EIMS·MSA@UiO-67 were collected on Rigaku SmartLab 9 Kw diffractometer, using Cu Kα radiation ($\lambda = 1.540598$ Å) at 45 kV and 200 mA within the range of $5° \leq 2\theta \leq 30°$ with a continues method, while that for ZW-UiO-67·0.5MSA and ZW-UiO-67·TFSA were collected on PANalytical Empyrean diffractometer at 40 kV and 40 mA. The scan speed of all is set as 4 degree/min, and the temperature-varying rate is 2 K/min. Inert atmosphere $N_2$ was used during the full diffraction experiment. Differential scanning calorimetry (DSC) curves of ZW-UiO-67·MSA and ZW-UiO-67·TFSA within −150 °C ~ 320 °C were recorded on DSC-Q200 TA instruments, while others' were carried out with a METTLER TOLEDO DSC3 instrument with 50 ml/min gas flowing rate. Thermogravimetric analysis (TGA) experiments were carried out on a Bruker TG/DTA 2000 SA, using a heating rate of 5 °C/min from 30 °C to 800 °C under $N_2$ atmosphere with the sample heated in an $Al_2O_3$ crucible. Solid state FT-IR spectra were obtained on a Bruker Equinox 55 FT-IR analyzer using KBr pellets within the wavenumber range of 4000 ~ 400 cm$^{-1}$. In-situ gas-phase FT-IR spectra were recorded from NETZSCH STA449F3 with Ar atmosphere as baseline, operated within 50 to 320 °C range with a 10 K/min rate. Variable temperature in-situ FT-IR was performed by use VERTEX 70 spectrometer with ATR imaging system. $^1H$ NMR spectra were recorded on a Bruker AVANCE III spectrometer at a frequency of 600 MHz at room temperature. $N_2$ adsorption isotherms were obtained at 77 K on a Micromeritics ASAP 2020 HD88 surface area analyzer. A transmission electron microscope (TEM, FEI Tecnai F20) equipped with energy-dispersive X-ray spectroscopy (EDX) was used to characterize the micromorphology, elemental composition and microstructure of the synthesized MOFs and glassy derivates. Scanning electron microscopy (SEM) was performed on a field emission scanning electron microscopy (Hitachi S−4800). Other physicochemical measurements and relevant discussion were detailed in the following sections.

### Structure and component determination

Desolvated ZW-UiO-67 has a formula $Zr_6O_4(OH)_4(L)_{5.4}(O_2CCH_3)_{1.2}$ (L = BPDC-MIMS) with partial acetate $CH_3CO_2^-$ as auxiliary linkers, which was determined via $^1H$-NMR spectrum (dissolved in deuterium chloride and DMSO-$d_6$). Crystalline structure of ZW-UiO-67 was further analyzed via computational simulation, showing the same framework as that of parent UiO-67 (Supplementary Figs. 1–3). The fingerprint patterns of powder x-ray diffraction (PXRD) of ZW-UiO-67 overlapped with that of UiO-67, further endorsing their same lattice framework (Supplementary Fig. 4). Brunaue-Emmet-Teller (BET) surface area (300.23 m$^2$/g) and pore volume (0.426 cm$^3$/g) are markedly reduced as compared to that of UiO-67 (2432 m$^2$/g, 0.954 cm$^3$/g) (Supplementary Table 1) due to the MIMS decoration. The measured pore size distribution of 12 Å and 6 Å (Supplementary Fig. 1) and the triangular window ($\varphi \approx 8$ Å) of the porous structure[29] are large enough for efficient diffusion of MSA (~2 Å), TFSA (~6 Å), TFA (~2 Å) and ESA (~4 Å)[49] into the pores. ZW-UiO-67·HA (HA = MSA, TFSA, TFA, ESA) were obtained through incipient wetness technique by loading Brønsted acid equal to the total pore volume of the desolvated ZW-UiO-67, having the stoichiometric formula $Zr_6O_4(OH)_4(L)_{5.4}·(O_2CCH_3)_{1.2}·(MSA)_{20.5}$, $Zr_6O_4(OH)_4(L)_{5.4}·(O_2CCH_3)_{1.2}·(TFSA)_{6.4}$, $Zr_6O_4(OH)_4(L)_{5.4}·(O_2CCH_3)_{1.2}·(TFA)_{15}$ and $Zr_6O_4(OH)_4(L)_{5.4}·(O_2CCH_3)_{1.2}·(ESA)_{16.3}$, respectively. Measured BET and pore values of ZW-UiO-67·HA are negligible, indicating the efficiently impregnated HA into the MOF (Supplementary Tables 1 and 2). The remained intact PXRD fingerprint patterns of ZW-UiO-67·HA before and after HA incorporation infer the preserved framework at room temperature (Supplementary Fig. 4).

### Structure modeling of ZW-UiO-67

Crystalline structure of ZW-UiO-67 was analyzed using powder X-ray diffraction (PXRD) with Cu Kα radiation in conjunction with multiscale computational simulations and Pawley refinement, by use the Material Studio 2019 software. The integrated intensities were extracted with the Pseudo-Voigt profile. The unit cell parameters $a$, $b$, $c$, $\alpha$, $\beta$, $\gamma$, and FWHM parameters, U, V, W, profile parameters NA, NB, and zero point were refined based on previous studies. It is observed that the obvious peaks at 5.6° and 6.7°, which are assigned to (2 0 −1) and (2 0 2) reflects in a $R_3$ space group. The full profile pattern matching (Pawley) refinements were carried out from the experimental PXRD patterns ($2\theta$ from 5° to 45°). The result of the Pawley refined patterns of ZW-UiO-67 are in good agreement with the that experimentally observed, producing the refined PXRD profile with lattice parameters of $a = b = 35.9719$ Å and $c = 43.2463$ Å, with negligible difference ($R_{wp} = 8.13\%$ and $R_p = 4.83\%$) (Supplementary Fig. 2).

### $^{13}C$ DD MAS and $^{13}C$ CP MAS SSNMR experiment

All experiments were carried out on a JNM-ECZ600R spectrometer at room temperature, with varied spinning rates at 12 kHz. The main magnetic field is 14.1 T, and corresponding Larmor frequency of $^{13}C$ is 150 kHz. All $^{13}C$ DD MAS NMR spectra were acquired using a 3.2 mm probe with a frequency of about 12 kHz, with adamantane ($^{13}C$, d = 29.5 ppm) as the first reference, using 8000-8800 scans with a recycle delay time of 5 s at room temperature (Fig. 4e, Supplementary Fig. 26). A standard solid single pause was employed. $^1H$-$^{13}C$ CP MAS NMR spectra for ZW-UiO-67, a$_g$ZW-UiO-67·MSA and a$_g$ZW-UiO-67·TFSA was recorded by using a rotation frequency of 12 kHz, 3200 scans with a recycle delay time of 5 s at room temperature.

### Differential scanning calorimetry (DSC) and analysis

All samples were preheated at 70 °C to exclude possible moisture from air. Each DSC curve was obtained with a heating rate of 10 °C/min under He atmosphere with the sample placed in Al crucible bearing lid, then cooled down to the low temperature point to start the first up-scan heating. For cyclic DSC heat-cool-heat process, the first up-scan heating is followed by cooling to the low temperature point, and then starting the secondary up-scan heat. DSC curves within −150 ~ 320 °C range were recorded for ZW-UiO-67·MSA, ZW-UiO-67·TFSA and ZW-UiO-68·MSA with aim to investigate the thermal behavior at subzero region, such as possible crystallization, and that thermal event at the high temperature before decomposition of ZW-UiO-67 ($T_d \approx 340$ °C). Similar thermal events were observed for these two composites, and special cyclic curves with $T_m$ (-127.2 °C, -153 °C), $T_g$ (-104 °C, -122 °C) and fusion enthalpy are shown in Supplementary Figs. 5 and 6. The endothermal peak at 185.4 °C for ZW-UiO-67·MSA and 193.2 °C for ZW-UiO-67·TFSA closely relate to that event for ZW-UiO-67 at 180.0 °C and ZW-UiO-67·0.5MSA at 186.2 °C. The third endothermal peak at 218.3 °C of ZW-UiO-67·MSA and 210.6 °C of ZW-UiO-67·TFSA also are similar and close to that at 212.6 °C found in ZW-UiO-67 and ZW-UiO-67·0.5MSA and $H_2$BPDC-MIMS (Supplementary Fig. 5), which may be ascribed to a plastic phase transition reminiscent of that found for solid ionic liquids[50]. The sharpest endothermal peak at 282.9 °C of ZW-UiO-67·MSA and 279.9 °C of ZW-UiO-67·TFSA before 340 °C may be attributed to the dehydroxylated OH of $Zr_6O_4(OH)_4$ as that evidenced for UiO (during 250–300 °C)[29], also being closely similar to that recorded for ZW-UiO-67 at 278.9 °C and ZW-UiO-67·0.5MSA at 292.0 °C. For the last exothermal event with onset temperature 256.7 °C of ZW-UiO-67·MSA (Fig. 2a), it may relate to some thermal decomposition taken place before 282.9 °C (total 3.2 $_{wt}$%). Similar thermal events are also observed for ZW-UiO-67·0.5MSA (onset: ~250 °C) and ZW-UiO-67 (onset: ~266 °C). Moreover, the compared behavior of $H_2$BPDC-MIMS·MSA shows the melting (−45.4 °C) and crystallizing (−45.9 °C) events during the cyclic DCS measurement, exhibiting the marked difference from that for ZW-UiO-67·HA

(Supplementary Fig. 6). For ZW-UiO-67·TFA, ZW-UiO-67·ESA, ZW-UiO-68·MSA and ZW-DUT-5·MSA, see also their DSC plots in Supplementary Figs. 5, 6, 30, 33, and 35. The measured enthalpies of fusion of ZW-UiO-67·HA, ZW-UiO-68·MSA and ZW-DUT-5·MSA lie within 2.12 J/g-11.45 J/g range (Supplementary Table 5), being comparable to that 5–10 kJ/mol for a large number of ZIF[31] and zeolitic silica polymorphs[33].

## Alternating current impedance measurements

a$_g$ZW-UiO-67·MSA was reheated to 90 °C and then compressed into sheet with both faces attached to gold wires. Before test, sample was pre-heated at 60 °C under vacuum overnight. At each temperature point, the measurement was conducted repeatedly with an interval of half an hour until the equilibrium was reached. A Corrtest CS350M electrochemical workstation in the frequency range from 0.01 Hz to 1 MHz with an AC amplitude of 15 mV. The conductivity values were calculated with the equation $\sigma = L/RS$, where $L$: length of sample (cm), $R$: resistance, and $S$: cross-sectional area of a sample (cm$^2$).

## SAXS/WAXS diffraction measurements

SAXS (small-angle X-ray scattering) and WAXS (wide-angle X-ray scattering) measurements were measured on a Xeuss 2.0 SAXS/WAXS system (Xenocs SA, France). Cu K$\alpha$ X-ray source (GeniX3D Cu ULD), generated at 50 kV and 0.6 mA, was utilized to produce X-ray radiation with a wavelength of 1.5418 Å. A semiconductor detector (Pilatus 300 K, DECTRIS, Swiss) with a resolution of 487 × 619 pixels (pixel size = 172 × 172 μm$^2$) was used to collect the scattering signals. For two dimensional (2D) SAXS, the sample-to-detector distance was 1196.26 mm, which was determined by a Silver Behenate (AgC$_{22}$H$_{43}$O$_2$) standard. Each WAXS pattern was collected with an exposure time of 6 min. with 2θ range of 1.8-30 degree. The one-dimensional intensity profiles were integrated from background corrected 2D WAXS patterns (Supplementary Fig. 28). Beamstop: D = 5 mm.

## DFT theoretical study of the acid-zwitterion buffer (BPDC-MIMS·MSA) effect on the ligand coordination ability of BPDC-MIMS to metal ion

Theoretical investigations were performed by using the Gaussian09 program[51]. The hybrid M06-2X density functional[52] and 6-31 + G** basis set were used to better described the weak interaction between BPDC-MIMS$^{2-}$ (deprotonated) and MSA. The coordination capability of the carboxylate ions was focused. Since the MOF has too huge number of atoms that prevent the calculation with quantum chemistry method, the single molecule model BPDC-MIMS$^{2-}$, and it coordinated with one and three MSAs were constructed. Their optimized structures are shown in Supplementary Fig. 18a. For the BPDC-MIMS$^{2-}$ model shown in left, the charges on the four O atoms labeled with 1, 2, 3 and 4 in the two carboxylate ions are −0.64, −0.64, −0.56 and −0.59 |e|, respectively. After coordination with one MSA, the charges become −0.64, −0.64, −0.64 and −0.62 |e|. The negative charges on the O atoms 3 and 4 in a carboxylate ion increase, so the coordination capability with metal cations increases. The coordination with one MSA thus does not weaken the coordination capability of the carboxylate ions. In Supplementary Fig. 18a (middle), there is a hydrogen bond between the MSA and −SO$_3$$^-$ in the MIMS$^{2-}$. Additional weak interaction occurs between an O atom in the MSA and an H atom in the imidazole motif. The distance between the two atoms is 2.08 angstrom. This interaction reduces the charge on the imidazole motif in the zwitterion form. However, the positive charge on the imidazole ring is partially compensated with a benzene ring (the two rings are parallel). This is the reason why the O atoms 3 and 4 have less negative charges before coordination with a MSA. In Supplementary Fig. 18a (right), the interaction between the BPDC-MIMS$^{2-}$ and the first MSA is similar and the distance is 2.01 angstrom. While, the negative charges decrease by the hydrogen bonds formed with the additional two MSA. The charges are −0.62, −0.53, −0.52 and −0.61 |e| for the four carboxylate O atoms,

respectively. The charges on the carboxylate ions are somewhat delocalized on the two MSA, and the coordination capability of the two carboxylate ions are weakened. Furthermore, the steric hindrance from the MSAs in the MOF should also reduce the coordination between the carboxylate ions and metal atoms.

## AIMD simulation methods

Ab initio MD (AIMD) simulations can take detailed host-gust interaction into account, although at higher computational cost[53]. AIMD simulations were carried out with the CP2K simulation package (version 7.1)[54] with a GPW (Gaussian and plane wave) basis set[55]. The *rev*PBE-D3 functional[56] with a plane wave cut-off of at a density functional theory level combined with a DZVP (double-zeta valence polarized) basis set and GTH (Goedecker–Teter–Hutter) pseudopotentials[57] were selected 350 Ry. A $1 \times 1 \times 1$ supercell is employed for ZW-UiO-67·HA of the first principle simulations. As level of theory, the *rev*PBE functional was chosen for its improved performance compared to PBE for the solid-state calculations. Dispersion interactions are incorporated by means of the D3 corrections of Grimme et al.[58,59]. For each guest molecule (MSA and TFSA), cell parameters were obtained by computing time-averaged values from a 2 ps $NpT$ (amount $N$, pressure $p$ and temperature $T$) molecular dynamics simulation at 823 K and atmospheric pressure. Production molecular dynamics runs that were used as input for the chemical-bond analysis were carried out in the $NVT$ (amount $N$, volume $V$ and temperature $T$) ensemble at 823 K for 10 ps. The temperature of the simulations was controlled with a Nosé–Hoover chain thermostat[60,61] consisting of three beads and with a time constant of 1000 wavenumbers. The pressure was controlled with a Martyna–Tobias–Klein barostat[62,63]. A time step of 0.5 fs is employed for integrating the equations of motion. The self-consistent field convergence criterion was set at 10$^{-6}$.

## XAFS measurements and analysis

The X-ray absorption find structure spectra (Zr K-edge) were collected at 1W1B beamline of Beijing Synchrotron Radiation Facility (BSRF) and BL14W1 beamline of Shanghai Synchrotron Radiation Facility (SSRF). The data were collected in fluorescence mode using a Lytle detector while the corresponding reference sample were collected in transmission mode, while the corresponding oxyde reference sample were collected in transmission mode in TableXAFS-500A from Anhui Chuangpu Instrument Technology Co., LTD. with each sample being grinded and uniformly daubed on the special adhesive tape.

The acquired EXAFS data were processed according to the standard procedures using the ATHENA module of Demeter software packages. The EXAFS spectra were obtained by subtracting the post-edge background from the overall absorption and then normalizing with respect to the edge-jump step. Subsequently, the $\chi(k)$ data of were Fourier transformed to real (R) space using a hanning windows (dk = 1.0 Å$^{-1}$) to separate the EXAFS contributions from different coordination shells. To obtain the quantitative structural parameters around central atoms, least-squares curve parameter fitting was performed using the ARTEMIS module of Demeter software packages.

The following EXAFS equation was used:

$$\chi(k) = \sum_{j} \frac{N_j S_0^2 F_j(k)}{k R_j^2} \cdot \exp\left[-2k^2 \sigma_j^2\right] \cdot \exp\left[\frac{-2R_j}{\lambda(k)}\right] \cdot \sin\left[2kR_j + \phi_j(k)\right]$$

(1)

the theoretical scattering amplitudes, phase shifts and the photo-electron mean free path for all paths calculated. $S_0^2$ is the amplitude reduction factor, $F_j(k)$ is the effective curved-wave backscattering amplitude, $N_j$ is the number of neighbors in the jth atomic shell, $R_j$ is the distance between the X-ray absorbing central atom and the atoms

in the jth atomic shell (back scatterer), λ is the mean free path in Å, φ $_j$(k) is the phase shift (including the phase shift for each shell and the total central atom phase shift), σ$_j$ is the Debye-Waller parameter of the jth atomic shell (variation of distances around the average R$_j$). The functions F$_j$(k), λ and φ $_j$(k) were calculated with the ab initio code FEFF9. The additional details for EXAFS simulations are given below.

All fits were performed in the *R* space with *k*-weight of 2 while phase correction was also applied in the first coordination shell to make R value close to the physical interatomic distance between the absorber and shell scatterer. The coordination numbers of model samples were fixed as the nominal values. While $S_0^2$, the internal atomic distances R, Debye-Waller factor $\sigma^2$, and the edge-energy shift Δ were allowed to run freely.

## X-ray total scattering measurements and PDF (pair distribution function) analysis

X-ray total scattering data were measured at BL13W1 beamline in an energy state of 40 keV (0.30996 Å) of the Shanghai Synchrotron Radiation Facility (SSRF). Samples were finely ground before loading into 1.0 mm (outer diameter) quartz capillaries. The empty capillary and also empty instrument data were collected for the background. The collected 2D scattering patterns were masked and azimuthally integrated into 1D diffractograms *I(Q)* by Diamond Light Source Dawn 2.24 package. Then the reduced pair distribution function *D(r)* was obtained by Fourier transforming the total scattering structure function *i(Q)* derived from *I(Q)* using GudrunX software[64], taking into account all necessary corrections and including the standard Lorch modification function within the Fourier transform in order to reduce the effect of ripples albeit at the cost of a slight broadening of the resultant PDF. The instrumental maximum *Q*-max, was set to 16 Å$^{-1}$ and that of *Q*-min was set to 0.1 Å$^{-1}$.

The following *D(r)* equation was used:

$$D(r) = \frac{2}{\pi} \int_0^\infty Qi(Q)\sin(Qr)dQ \qquad (2)$$

## Sample preparation

All chemicals were analytical grade and obtained from commercially available sources, and used without further purification. Ligand H$_2$BPDC-MIMS was synthesized accordingly to the process reported by us recently (*T*$_m$ = 324 °C)[30].

## Synthesis of UiO-67 and ZW-UiO-67

UiO-67 was synthesized following the procedure reported[29]: ZrCl$_4$ (400.76 mg, 1.72 mmol), acetic acid (1.75 mmol), H$_2$BPDC (4,4'-Biphenyldicarboxylic acid) (414.24 mg, 1.72 mmol). ZW-UiO-67 was obtained via the same procedure as that for UiO-67, except ZrCl$_4$ (120 mg, 0.514 mmol) and ligand H$_2$BPDC-MIMS (228.7 mg, 0.514 mmol) were used. After that resulted solid was soaked in methanol for 3 days, the collected solid was subject to further drying under vacuum at 150 °C for 24 h. Yield 72%. The component was confirmed via $^1$H NMR and TGA, with a formula of Zr$_6$O$_4$(OH)$_4$(BPDC-MIMS)$_{5.4}$(O$_2$CCH$_3$)$_{1.2}$ (Supplementary Table 1). The desolvated sample was kept in glovebox for characterization and further use.

## Synthesis of ZW-UiO-67·HA, ZW-UiO-67·0.5MSA and HA@UiO-67 (HA = MSA, TFSA, TFA, ESA)

To exclude moisture from air, all synthesis were performed under nitrogen atmosphere protection. All composites were obtained via the same incipient wetness technique. For example, de-solvated ZW-UiO-67 (80 mg) (0.0256 mmol) was added into the mortar. After forced grinding, 0.0341 ml (0.583 mmol, $\rho = 1.48$ g/ml) methanesulfonic acid (MSA) equaling to the total pore volume of the ZW-UiO-67

(0.426 cm$^3$/g) was added dropwise to the powders with continuously grounding until the MSA was completely absorbed by ZW-UiO-67. The mixture was then sealed into a glass vial and heated in a drying oven at 60 °C for 24 h to distribute the MSA uniformly into the MOF channel. The resulted powder was subject to further dry under vacuum at room temperature for 2 h, obtaining the ZW-UiO-67·MSA. The resultant sample was stored under N$_2$ atmosphere before use without further purification. Other ZW-UiO-67·HA, ZW-UiO-67·0.5MSA, HA@UiO-67 and ILs@UiO-67were obtained following the procedure as that for ZW-UiO-67·MSA by use various HA or ILs. Components of ZW-UiO-67·HA, ZW-UiO-67·0.5MSA and HA@UiO-67 are listed in Supplementary Tables 2 and 5 for comparison.

## Synthesis of acid-zwitterion ionic liquids (ILs = EMIM·TFS, EIMS·MSA, H$_2$BPDC-MIMS·MSA)

Except the EMIM·TFS i.e. (1-ethyl-3-methylimidazolium bis(trifluoromethylsulfonyl)imide) obtained through commercial source, other ionic liquids (ILs) were synthesized following the procedure as that for Bronsted acidic ILs reported[15]. For EIMS·MSA, zwitterion 1-(1-ethyl-3-imidazolio)-propane-3-sulfonate (EIMS) and methanesulfonic acid (MSA) with the zwitterion salt to acid ratio of 1:3.78 were mixed and dissolved in methanol with stirring at 80 °C. After 6 h, the solvent was removed via rotary evaporator under heating. The resultant liquid was further dried at 120 °C overnight under vacuum, which was cooled to room temperature under N$_2$ atmosphere, obtaining the EIMS MSA. See also Supplementary Fig. 1, and Supplementary Table 2.

## Synthesis of ILs@UiO-67 and ILs@ZW-UiO-67 (ILs = EMIM·TFS, EIMS·MSA)

ILs@UiO-67 and ILs@ZW-UiO-67 were obtained and treated following the same procedure as that for ZW-UiO-67·HA by use different ionic liquids (ILs) with the volume equal to the pore of MOF activated. For ILs@UiO-67, EMIM·TFS ($\rho = 1.38$ g/ml) and EIMS·MSA ($\rho = 1.42$ g/ml) were used to produce EMIM·TFS@UiO-67 and EIMS·MSA@UiO-67, respectively. For ILs@ZW-UiO-67, EMIM·TFS was used to obtain EMIM·TFS@ZW-UiO-67. More information are listed in Supplementary Table 5.

## Synthesis of a$_g$ZW-UiO-67·HA (HA = MSA, TFSA, TFA, ESA)

Each ZW-UiO-67·HA sample was heated at a rate of 10 °C/min under N$_2$ atmosphere to form the flow like liquid. For ZW-UiO-67·MSA and ZW-UiO-67·ESA, the heating temperature is up to 130 °C (>*T*$_m$) and 140 °C, respectively, for ZW-UiO-TFSA is to155 °C, for ZW-UiO-67·TFA is 160 °C. The melting samples were then cooled to room temperature, obtaining their transparent glassy solids a$_g$ZW-UiO-67·HA. The glasses were stored under inert atmospheres for further characterization without additional treatment.

## Synthesis of a$_{gr}$ZW-UiO-67·HA

All a$_{gr}$ZW-MOF·HA were obtained through the same solvent stimulation procedure. To view the recoverability of a$_g$ZW-MOF·HA under a moderate condition, soaking the as-made a$_g$ZW-MOF·HA in methanol with stirring for 3–5 min at room temperature was conducted. To fully recover a$_g$ZW-MOF·HA, a reflux treatment under N$_2$ atmospheres is further needed to produce the solid denoted as a$_{gr}$ZW-MOF·HA. For the case of a$_g$ZW-UiO-67·MSA, 80 mg solid was washed with methanol twice, then the solid was collected and soaked in 50 ml flask containing 30 ml methanol with stirring for 3 min at room temperature (yield 93%). After centrifugation, the solid was suspended in fresh methanol and refluxed with stirring at 80 °C for 1 h under N$_2$ atmospheres, and the powder was collected and subjected to further 2 h refluxing with fresh methanol. The resulted was collected via centrifugation for further characterization. 53.6 mg white powder of a$_{gr}$ZW-UiO-67·MSA was obtained, yield 67% based on the mass of starting a$_g$ZW-UiO-67·MSA. Elemental analysis shows C: 35.73%, N: 3.72%, H: 4.18% of

the $a_{gr}$ZW-UiO-67·MSA, meaning a stoichiometry formula of $Zr_6O_4(OH)_4(L)_{5.4}(O_2CCH_3)_{1.2}(MSA)_{12.3}$ (calc. C: 35.50%, N: 3.51%, H: 3.74%) that can be described as ZW-UiO-67·0.6MSA with 60% MSA preserved within the collected powder (Supplementary Table 2). It showed same Bragg diffractions as that of pristine ZW-UiO-67·MSA, and the weight loss may be ascribed to the destroyed framework of ZW-UiO-67 with fragment exfoliated during the melting and the following treatment.

## Synthesis of DUT-5 and ZW-DUT-5

Synthesis for ZW-DUT-5 following the reported direct synthesis procedure for DUT-5[47] by use $H_2$BPDC-MIMS failed. Herein, by using mixed ligands of $H_2$BPDC-MIMS and $H_2$BPDC in 4:1 mole ratio, derivate ZW-DUT-5 was successfully obtained as following. In 50 ml Teflon-liner autoclave, $Al(NO_3)_3·9H_2O$ (260 mg, 0.70 mmol) and 45 equiv of modulator acetic acid were mixed and dissolved in 30 ml DMF by sonication. Then, $H_2$BPDC-MIMS (0.30 mmol) and $H_2$BPDC (0.075 mmol) were added and dissolved. After the Teflon-liner autoclave kept in an oven at 120 °C for 24 h, produce powder was collected and treated and activated as that for these MOFs above. $^1$H NMR spectrum showed the stoichiometric formula $Al(OH)(BPDC)_{0.18}(BPDC\text{-}MIMS)_{0.72}(O_2CCH_3)_{0.2}$ (Supplementary Fig. 32). Yield 71.8% (based on the ligand). Parent DUT-5 was obtained and treated following the same procedure as that for ZW-DUT-5 by solely using $H_2$BPDC ligand, yield 82%.

## Synthesis of ZW-DUT-5·MSA, $a_g$ZW-DUT-5·MSA and $a_{gr}$ZW-DUT-5·MSA

ZW-DUT-5·MSA was obtained and treated through the same procedures as that for ZW-UiO-67·HA. MSA of 6 equivalent to that BPDC-MIMS in ZW-DUT-5 (based on the stoichiometric formula $Al(OH)(BPDC)_{0.18}(BPDC\text{-}MIMS)_{0.72}(O_2CCH_3)_{0.2}$) was used. $a_g$ZW-DUT-5·MSA was obtained via the melt-quenching method by heating ZW-DUT-5·MSA up to 152 °C ($T_m$ = 152 °C), which was suspended within methanol with stirring for 3 min and collected to produce the $a_{gr}$ZW-DUT-5·MSA (Supplementary Fig. 33).

## Synthesis of ligand $H_2$TPDC-MIMS

$H_2$TPDC-MIMS was obtained by de-esterification of the precursor 2,2'-((4,4''-bis(methoxycarbonyl)-[1,1':4',1''-terphenyl]-2',5'-diyl)bis(1H-imidazole-3-ium-1,3-diyl))bis(ethane-1-sulfonate) (P1) received from lab. To the solution of $P_1$ (1.502 g, 2 mmol) dissolved in mixed MeOH/$H_2O$ (v: v = 3:1) (90 ml), LiOH (95.92 mg, 4 mmol) was added with stirring. Then the clear solution was refluxed for 12 h. After cooling to room temperature, the mixture was acidified to pH = 1 with HCl (conc, 37% w/w) to yield a white precipitate, which was filtered, washed with water, and further dried under vacuum to obtain ligand $H_2$TPDC-MIMS (1.372 g, 95%). $^1$H NMR (600 MHz, DMSO-$d_6$): δ 1.97 (4H, m), 2.38 (4H, t, J = 7.04 Hz), 4.18 (4H, t, J = 8.07 Hz), 5.45 (4H, s), 6.77 (2H, s), 6.86 (2H, s), 7.03 (2H, s), 7.32 (2H, s), 7.47 (4H, d, J = 5.62 Hz), 8.01 (4H, d, J = 5.38 Hz), 13.09 (2H, s) (Supplementary Fig. 34).

## Synthesis of ZW-UiO-68, ZW-UiO-68·MSA, $a_g$ZW-UiO-68·MSA and $a_{gr}$ZW-UiO-68·MSA

ZW-UiO-68 was synthesized following the procedure as that for ZW-UiO-67: $ZrCl_4$ (120 mg, 0.514 mmol), ligand $H_2$TPDC-MIMS (371.12 mg, 0.514 mmol). The produced solid was soaked in methanol for 3 days, the collected solid was subject to further drying under vacuum at 150 °C for 24 h. Yield 75%. The desolvated MOF was further confirmed via $^1$H NMR and BET measurement, suggesting a formula of $Zr_6O_4(OH)_4(TPDC\text{-}MIMS)_{5.62}(O_2CCH_3)_{0.76}$, surface area of 694.22 $m^2$/g and pore volume of 0.6826 $cm^3$/g. The desolvated sample was kept in glovebox for characterization and further use. ZW-UiO-68·MSA, $a_g$ZW-UiO-68·MSA and $a_{gr}$ZW-UiO-68·MSA (yield 80%) were respectively obtained through the same procedures as those for the derivatives of ZW-UiO-67. Relevant characteristics are shown in Supplementary Fig. 35.

## Reporting summary

Further information on research design is available in the Nature Portfolio Reporting Summary linked to this article.

## Data availability

All data generated or analyzed during this study are included in this published article and its supplementary information files which are available in the online version of the paper. Source data are provided with this paper.  Correspondence and requests for materials should either be addressed to W. Li, J. Zhang, G. Xu and C. -Q. Wan. Source data are provided with this paper.

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

## Acknowledgements

We thank for support from the National Natural Science Foundation of China (Nos. 22071157, 22171263, 62227815, 2020000052, 21975132, 22325109, 91961115, 12174274, 12204366), National Key Research and Development Program of China (2022YFA1503300) and Beijing Natural Science Foundation (L223010). We also acknowledge support from the State Key Laboratory of Structural Chemistry at Fujian Institute and the Key Laboratory of Life Organic Phosphorus Chemistry and Chemical Biology, Ministry of Education, Tsinghua University. We gratefully

acknowledge BL13W1 and BL14W1 beamline of Shanghai Synchrotron Radiation Facility (SSRF) and 1W1B beamline of Beijing Synchrotron Radiation Facility (BSRF) for providing beam time. Great thanks also for the AIMD calculation study of Prof. Mingbin Gao of Dalian Institute of Chemical Physics, Chinese Academy of Sciences, and Dr. Zhigang Li of School of Materials Science and Engineering & Tianjin Key Laboratory of Metal and Molecule-Based Material Chemistry of Nankai University.

## Author contributions

C.-Q.W. conceived the idea and designed the experiments. W.-L.X., J.Z., Z.C. and H.D. conducted the X-ray absorption spectra (XAS) and X-ray total scattering experiments and data analysis. W.-L.X., G.-Q.L., H.C., L.W., Y.-C.H., S.-Y.H. did the synthesis and characterizations. X.-L.G. analyzed some data and refined the crystal structure. L.F., W.-H.D. and G.X. performed the conduction and SEM measurements. Y.-H.D. conducted the solid-state NMR experiments and analysis. G.W. performed the DFT theoretical calculations. W.-L.X., G.-Q.L., L.T., M.T.D., W.L., and C.-Q.W. contributed to PDF and data analysis and graphic drawing. C.-Q.W. wrote and edited the manuscript, W.L. and G.X. revised the manuscript.

## Competing interests

The authors declare no competing interests.
