## [Peer Review File · Nature Communications]

REVIEWER COMMENTS

Reviewer #1 (Remarks to the Author):

In this work, Xue et al. presents a very comprehensive report of a generalized method for making carboxylate MOF glasses. This method is near universal and can be extended to other MOF systems. The reported strategy is of great scientific significance and could open up a brand-new pathway for making MOFs and other molecular crystals into glasses. In this regard, I highly recommend this manuscript for publication in Nature Communications after a major revision. The authors could consider the following points to strengthen their manuscript:

1. In Figure 1a, the 'Zr-O node' should be changed to 'Zr₆O₄(OH)₄ node' to avoid misunderstanding. Similar changes need to be made in the supporting information.
2. In the SEM images of Figure 2b, the scale bars are not uniform. For better comparison, I suggest the authors make the scale bar in all images with the same length.
3. In my opinion, the proposed strategy could be extended to other molecular frameworks. Covalent-organic frameworks are probably the immediate targets, can the authors give some comments whether this could be possible?
4. The application prospects of the reported new carboxylate MOF glasses were only briefly mentioned, the discussion could be slightly extended.
5. Considering the complexity, I do not believe it is possible to do the reverse Monte Carlo simulations to obtain the structures of the melt-quenched glasses. Nevertheless, I am curious whether the authors have had a go with it.
6. I believe the authors' results are robust enough to support their conclusion of the melt-quenched formation of glasses, however, have they obtained any evidence of excluding the formation of monoliths? This is very important for convincing the community.
7. Ref 42 entitled "Ionic liquid facilitated melting of the metal-organic framework ZIF-8" (Nat. Commun. 2021, 12:5703) is a very important reference, I suggest the authors move it to the very beginning of the manuscript.
8. The influence of incorporated acids on the melting process needs to be discussed. With reference to the simulation work in Ref 3 and Ref 12, the thermodynamic barriers of metal-node dissociation at high temperatures can be briefly discussed, to intuitively reflect the lowering of melting barriers by the acid.

Reviewer #2 (Remarks to the Author):

In this work, the authors developed a Brønsted acid-base strategy for melt-quenched glass formation of metallic carboxylate frameworks by grafting amphiphilic ions onto carboxylate ligands and incorporating organic acids into framework channels. Upon heating, the ordered electrostatic interactions between the attached amphiphilic ions and the incorporated organic acids are stochastically shifted due to charge delocalisation. The melting and glass-forming behaviour is mainly supported by TGA, DSC, PXRD, SEM and other characterization results. This work provides a route to the possibility of melting carboxylate MOFs. Through experiments from different aspects, the process of melting MOF glass is explored, which is very beneficial for understanding the process of MOF glass. Actually, carboxylate MOF glasses have been recently reported by different strategies including the melt-quenching method (10.21203/rs.3.rs-2922761/v1) and desolvation approaches (Angew. Chem. Int. Ed. 2023, e202305942; 10.26434/chemrxiv-2023-05kv3). The authors should discuss the related reports on carboxylate MOF glasses, especially the firstly reported cluster-based carboxylate MOF glass. Based on these considerations, the current manuscript is still too preliminary and major revision is necessary to support its publication.

1. For ZW-UiO-67-MSA in Fig. 2a, "The second endothermic peak at 185.4 oC can be ascribed to the de-associated acetate", I think this should be reconsidered because the coordinated CH₃COO⁻ should be more stable than MSA, and further characterizations are needed, for example TG-MS.
2. The low-density liquid (LDL) to high-density liquid (HDL) transition needs to be proven.
3. "However, the endothermic evidence (218.3 oC) is reminiscent of the phase transition for ZW-UiO-67 (~212.6 oC) (Extended Data Fig. 5)", what is the new phase?
4. Is there a mistake in "(1-methyl-3-imidazolium) propane-3-sulfonate)" in line 85?

5. The authors emphasize that the Zr-O-Zr connection in agZW-UiO-67·MSA and agZW-UiO-67·TFSA is still reserved, but the Zr-O-Zr region in Extended Data Fig.19 is negligible. They need more clear evidences to support their points.
6. agZW-UiO-67·MSA was immersed in methanol and refluxed to obtain agrZW-UiO-67·MSA. Is it because agZW-UiO-67·MSA was dissolved in methanol and then re-formed nanocrystals? Are other solvents used (polar solvents, non-polar solvents, H₂O, etc.) to rule out the possibility of ligand dissolution leading to recrystallization?
7. From the relative low porosity of agrZW-UiO-67·MSA than ZW-UiO-67·MSA, authors guess the 60% of the MSA still remains in the framework. What is the degree of crystallinity for the agrZW-UiO-67·MSA? This is another possible reason for the lower porosity.
8. ZW-UiO-67·0.5MSA and ZW-UiO-67·MSA have the different thermodynamic behaviors. ZW-UiO-67·0.5MSA cannot melt. The authors should provide further experiments to discuss the role of MSA in the melting behaviors.
9. As a control experiment, UiO-67 is mixed directly with EIMS-MSA to check the melting behaviors. VT-PXRD shows that some peaks of the mixture disappear during heating in Extended Data Fig.7. Is this possibly caused by uneven mixing and not complete melting? Is there any evidences that EIMS-MSA is evenly distributed in UiO-67?
10. In Extended Data Fig. 24, the NMR spectrum is not clearly discussed. The peak at 182.3 ppm for agZW-UiO-67·MSA is difficult to observe.

Reviewer #3 (Remarks to the Author):

In this manuscript, the authors reported the realization of melting and the preparation of quenched glass of three typical metal-carboxylate MOF of UiO-67, UiO-68 and DUT-5. The research follows one of the important trends in MOF chemistry, that is transforming traditional crystalline MOFs into their liquid or glassy states. Relative efforts have gained success firstly in a limited numbers of ZIFs and some 1D/2D coordination polymers, and recently in traditional MOFs with both single metal and cluster as nodes. Even so, the area is quite new with a lot of puzzles remain to be explored. From this point, the presented result is interesting and significant to attract wide readership.

For the investigated content, all the three compounds have long been known as quite thermal stable MOF constructed from [Zr₆] node, yet decomposition rather than melting occur upon heating. Such fact indeed shows the current situation that most MOFs, especially the vast number of metal-carboxylate MOFs, can not be melted. This work developed a Brønsted acid-base strategy by grafting the zwitterions on the carboxylate ligands and incorporating organic acids in the framework channels to enable the melt-quenched glass formation of UiO-67, UiO-68 and DUT-5. The method somewhat draw lessons from traditional ionic liquid with long research history, and similar strategy has been successful in ZIF-8 (Ref. 42) which is also known without melting. Even so, present research results are no doubt important and valuable to enhance understanding to MOF glass preparation. Especially, the research is performed in a comprehensive and thorough way with abundant characterization and experimental evidences, which is infrequent and laudable in peer study. Hence, I think this paper is a suitable candidate for publication in Nat. Commun. Some queries needed to be answered are listed as following:

1. As a great aspiration, the authors intend to construct a "generally applicable strategy" to vitrification of wide range of MOFs which are "composed of different ligands and metal nodes" and "not limited to the structure type and porosity". However, presented three MOFs are in fact quite similar either in composition and structure. Particularly, they possess very rigid framework to endure zwitterions grafting and organic acids inclusion. While most known metal-carboxylate framework did not have such high structural stability. The method applicability is open to doubt, or at least need much more examples. On this situation, I think the presented title is too broad and the conclusion is exaggerated.
2. All the DSC results are quite strange comparing to known MOF glasses and also other traditional glass system. These abnormal data further lead to great queries in several important thermal parameters of T_m and T_g for all reported compounds. I noticed two instruments of DSC-Q200 and METTLER TOLEDO DSC3 were applied to collected data. How about the data repeatability, as well as the influence factor of flowing gas, heating rate?
3. For the typical DSC of ZW-UIO-67·MSA in figure 2a, firstly the really melting-range is indeed

confusing for readers. Why the T_m point is in a platform position? More importantly, the authors listed three temperature points of 162.5, 185.4 and 218.3, which seems located at TG range without weight loss. As well known, the melt liquid goes in a thermal stable energy state as homogeneous phase with steady DSC trace. What happens at these points with evident endothermic and exothermic signals? The two-paragraph discussion on thermal behaviors in Page 6 are convoluted.

4. The authors said "the following exothermal and another event with onset 256.7 oC may be polymorphic phase transitions from the low-density liquid (LDL) to high-density liquid (HDL) driven by different potential energies between distinct structural configurations of neighbor states". However, comparing to Ref-5, the signals at 256.7 oC shows completely different characters. The dramatic exothermic trends are really strange. Without sufficient and credible evidence, such statement of LDL and HDL is improper.

5. In page 5, the formula for ZW-UiO-67-MSA seems incorrect, the BPDC should contain the grafted MIMS. In addition, why 1MSA and 0.5MSA per ZW-UiO-67 in page 7. Considering there are a numbers of compounds in the paper, a list showing all the abbreviation, accurate formula in necessary for readership.

6. For the description of proton conduction in Page 7, the reported data is not clear for which sample.

7. For the recovery experiments, how about the solvent-dependent? Excepting methanol, can other solvent induce the glass-to-crystalline transformation. In addition, to maintain the molding film, is vapor possible to facilitate the recovery?

8. It seems enough content of incorporating organic acids is crucial for the success of melting, how about the critical point for the three MOFs?

9. On the whole, the organization of different parts as well as discussion in the same part is not sufficiently good, which leads to easy lost of main idea of the authors. Though it may be ascribed to the large experimental data, further improvement on the organization and language is necessary.

Reviewer #4 (Remarks to the Author):

There is a great deal of current interest in the new family of glasses derived from metal organic frameworks, but most of the ones that have been reported to date are based on organic linkers that bind to metals via nitrogen. For example, the most widely studied MOF glasses are those based on zeolites imidazolate frameworks (ZIFs). The manuscript submitted by Yue et al describes an ingenious way of making glasses from MOFs based upon carboxylate linkers, which constitute the largest family of MOFs. This work will therefore be of considerable interest to the MOF community.

The new carboxylate glasses are mainly formed by starting with derivatives of MOF UiO-67 that have been functionalized by substituting zwitterionic amines onto the aromatic linkers and then introducing a complementary Brønsted acid, such as methylsulfonic acid, into the MOF cavity. The amine substituent and the Brønsted acid form zwitterionic pairs that facilitate the low temperature melting of the MOF. The melts can then in turn be quenched to yield glasses. One of the interesting features of these glasses is that the crystalline MOF precursor can be retrieved from the glass by leaching the acid out of the glass by using a simple organic solvent such as methanol. This reversibility is not usually found in ZIF glasses.

The characterisation of the MOF glasses in the manuscript is very thorough, even though the structures of the crystalline material precursors depend upon powder X-ray diffraction data. The use of other techniques, such as EXAFS, total scattering XRD, vibrational spectroscopy and solid state NMR, has been used to probe the local structures of the glasses and compare them to the crystalline precursors. Overall, I think that the work has been carried out very carefully and that the claims of glass formation are well supported by the data.

I think that this interesting work could be published in Nature Communications after a couple of important points have been addressed:

1. The authors claim that they have developed a universal strategy for making glasses from carboxylate MOFs (see pages 3 and 14). I think that this claim is exaggerated and needs to be toned down. For example, the MOF cavities need to be large enough to accommodate both the zwitterionic amine substituents AND the complementary Brønsted acid molecules. This approach can only therefore be used for MOFs with large cavities such as the UiO family. Carboxylate MOFs with smaller cavities, of which there are very many, will be unsuitable for this approach. The claim of universality must be replaced by a more suitable and justifiable claim.

2. Most of the work concerned MOFs from the UiO family and these are well-known to be prone to having large concentrations of defects in the form of missing octahedra (from the work of Goodwin). This should be mentioned in the manuscript since these large defect vacancies may play a role in accommodating the amine substituents and the Brønsted acid molecules. I also wonder if any of the characterisation methods reveal any evidence for the presence of defects in the crystalline precursors.

If these two points are properly addressed, I think that the paper could be accepted for publication in Nature Communications.

RESPONSE TO REVIEWERS' COMMENTS

Reviewer #1 (Remarks to the Author):

In this work, Xue et al. presents a very comprehensive report of a generalized method for making carboxylate MOF glasses. This method is near universal and can be extended to other MOF systems. The reported strategy is of great scientific significance and could open up a brand-new pathway for making MOFs and other molecular crystals into glasses. In this regard, I highly recommend this manuscript for publication in Nature Communications after a major revision. The authors could consider the following points to strengthen their manuscript:

Response: We sincerely thank the reviewer for the enthusiastic comments on the novelty, significance and universality of our work.

1. In Figure 1a, the 'Zr-O node' should be changed to 'Zr₆O₄(OH)₄ node' to avoid misunderstanding. Similar changes need to be made in the supporting information.

Response: We have replaced 'Zr-O node' with 'Zr₆O₄(OH)₄ node' in both Fig. 1a and Supplementary Fig. 1.

2. In the SEM images of Figure 2b, the scale bars are not uniform. For better comparison, I suggest the authors make the scale bar in all images with the same length.

Response: We have reorganized Figure 2b to make all the scale bars uniform.

3. In my opinion, the proposed strategy could be extended to other molecular frameworks. Covalent-organic frameworks are probably the immediate targets, can the authors give some comments whether this could be possible?

Response: We fully agree with the reviewer that our strategy can be extended to other molecular systems. Our recent results have demonstrated that covalent-organic frameworks can indeed be made into glasses using this strategy. Corresponding discussions and prospect of this strategy have been added to the last paragraph of the revised manuscript as the following:

“Importantly, our strategy could be applied to a range of highly porous MOFs composed of different ligands and metal nodes. In addition, this strategy could also be extended to other kinds of highly porous molecular frameworks such as covalent-organic frameworks.”

4. The application prospects of the reported new carboxylate MOF glasses were only briefly mentioned, the discussion could be slightly extended.

Response: We have slightly extended our discussion about the application prospect in the “Discussion” section as the following: “...Our findings would also offer new opportunities for processing MOFs and other hybrid organic-inorganic crystals by melt-quenching them into glasses, which could enable them to meet the manufacturing requirement by industry...”

5. Considering the complexity, I do not believe it is possible to do the reverse Monto Carlo simulations to obtain the structures of the melt-quenched glasses. Nevertheless, I am curious whether the authors have had a go with it.

Response: We thank the reviewer for this insightful comment. We have indeed spent a year and a half in trying the RMC simulations on the ZW-UiO-67·HA glasses with many different approaches.

However, we were unable to achieve this goal due to lacking of reasonable starting structural models, even with enormous endeavor by the inventor of the RMC simulation software package (Prof. Martin T. Dove, who is also a co-author of this revision). Specifically, the RMC models of ZIF glasses are based on the SiO₂ glass by replacing Si atom with the imidazolate ligand, while the SiO₂ glass model is derived from the amorphous Si by inserting O atoms between adjacent Si atoms. Since the majority of MOFs are not four-connected (including ours), the RMC simulations have only been successfully done for a few ZIF type MOFs (please see Scheme. R1 below) up to date.

Scheme R1. Explanation of the unsuccessful RMC simulations for non 4-connected MOFs.

6. I believe the authors' results are robust enough to support their conclusion of the melt-quenched formation of glasses, however, have they obtained any evidence of excluding the formation of monoliths? This is very important for convincing the community.

Response: We thank the reviewer for raising this valuable comment. To exclude the formation of monoliths, comprehensive small angle X-ray scattering (SAXS), wide angle X-ray scattering (WAXS) and electron diffraction experiments have been performed on ZW-UiO-67 and a_gZW-UiO-67·HA. The results have been compared with those of polycarbonate and silicate glasses, which clearly show the melt-quenched a_gZW-UiO-67·HA glasses are not the monoliths of nanoparticles blended within organic acids (Supplementary Figs. 27-28). For a_gZW-DUT-5·MSA and a_gZW-UiO-68·MSA, the optical microscopic images, SEM images, TEM images and electron diffraction patterns from the glasses also have been carefully recollected and shown in Supplementary Figs. 29, 32 and 34. As expected, no electron diffraction of a monolith with nanocrystals or nanoparticles was observed, explicitly implying the homogenous amorphous states of these melt-quenched carboxylate a_gMOFs. Corresponding discussion has been added to page 17 of the revised manuscript as the following:

“The SAXS signal $I(q)$ of a_gZW-UiO-67·HA crucially continues, rather than that of a monolith blended with nanoparticles^[45]. In addition, no detectable wide angle X-ray scattering (WAXS) peaks and the missing electron diffraction in TEM images were recorded for a_gZW-UiO-67·HA, exhibiting comparable amorphous features to those of polycarbonate and silicate glasses (Supplementary Fig. 27-28).”

7. Ref 42 entitled “Ionic liquid facilitated melting of the metal-organic framework ZIF-8” (Nat. Commun. 2021, 12:5703) is a very important reference, I suggest the authors move it to the very beginning of the manuscript.

Response: We have moved this important reference to the beginning of the revised manuscript. In addition, related discussion has been added to the introduction section on page 4.

“Meanwhile, the un-meltable counterparts ILs@UiO-67 were also synthesized as references to show that such an universal strategy is in stark contrast to the reported guest-host interactions scenario for the melt of ILs@ZIF-8^[27].”

8. The influence of incorporated acids on the melting process needs to be discussed. With reference to the simulation work in Ref 3 and Ref 12, the thermodynamic barriers of metal-node dissociation at high temperatures can be briefly discussed, to intuitively reflect the lowering of melting barriers by the acid.

Response: We thank the reviewer for this important suggestion. The long-range order of ZW-UiO-67·HA was disrupted to lead to increased configurational entropy and reduced energy barrier beneficial for melting due to the following synergistic reasons: the charge delocalization and separation effect of the acid-zwitterion buffer system (MIMS·MSA), the weakened coordination bonds between -COO^- and metal nodes, and the dense structures with the fully filled channels that facilitate the break and mismatch of the coordination bonds. Additional discussions have been added on page 9 in the revised manuscript as follows:

“...theoretical calculations reveal the coordination ability of the carboxylate ligands is weakened as BPDC-MIMS paired with more nearest MSA molecules. The negative charges of four O_{COO} atoms change from -0.64 , -0.64 , -0.56 and $-0.59|e|$ to -0.62 , -0.53 , -0.52 and $-0.61|e|$ due to the charge delocalization and intermolecular interactions. It could be speculated the weakened Zr-O bonds may trigger the bond dissociation and atom displacements under the assistance of the acid-zwitterion buffer. However, the densely filled channels in ZW-UiO-67·MSA not only are beneficial for the coordination bonding mismatch between COO^- and metal nodes upon heating, but also enhance the structural disorder and increase the configurational entropy to markedly reduce the energy barrier required for the melting...”

Reviewer #2 (Remarks to the Author):

In this work, the authors developed a Bronsted acid-base strategy for melt-quenched glass formation of metallic carboxylate frameworks by grafting amphiphilic ions onto carboxylate ligands and incorporating organic acids into framework channels. Upon heating, the ordered electrostatic interactions between the attached amphiphilic ions and the incorporated organic acids are stochastically shifted due to charge delocalisation. The melting and glass-forming behaviour is mainly supported by TGA, DSC, PXRD, SEM and other characterization results. This work provides a route to the possibility of melting carboxylate MOFs. Through experiments from different aspects, the process of melting MOF glass is explored, which is very beneficial for understanding the process of MOF glass. Actually, carboxylate MOF glasses have been recently reported by different strategies including the melt-quenching method (10.21203/rs.3.rs-2922761/v1) and desolvation approaches (Angew. Chem. Int. Ed. 2023, e202305942; 10.26434/chemrxiv-2023-05kv3). The authors should discuss the related reports on carboxylate MOF glasses, especially the firstly reported cluster-based carboxylate MOF glass. Based on these considerations, the current manuscript is still too preliminary and major revision is necessary to support its publication.

Response: We thank the reviewer for the positive and critical comments on our work. We are sorry for not citing the most recently developed metal-carboxylate MOF glasses. We have cited these important work as references 22, 23 and 24. Additional discussions have been included in page 3 of the revised manuscript as shown in the following:

“...porous metal-carboxylate MOFs having high CN more than 4 (6 to 12) are believed to be unmeltable...Several melt-quenched metal complexes with ligands bearing carboxylate and a secondary coordination group (pyridyl/azolate) can be obtained from discrete solvated complexes through rearrangements of metal-ligand bonds upon desolvation^[22-23]. Taking advantage of the flexibility and low symmetry of the aliphatic carboxylate ligands and the lack of crystal field stabilization energy on metal ions, a selection of glasses comprising low oxidation state metals (Mg^{2+} , Mn^{2+}) and flexible adipate were reported very recently^[24]. However, this strategy is not applicable to those carboxylate MOFs bearing with high valence metal ions.”

Furthermore, careful revisions have been done and detailed in our responses set below.

1. For ZW-UiO-67·MSA in Fig. 2a, "The second endothermic peak at 185.4 oC can be ascribed to the de-associated acetate", I think this should be reconsidered because the coordinated CH_3COO^- should be more stable than MSA, and further characterizations are needed, for example TG-MS.

Response: We thank the reviewer for this valuable suggestion. In the original manuscript, the second endothermal peak at 185.4 °C of the DSC plot was temporarily attributed to the decomposition of the acetate group because similar thermal behavior was reported in $Zr(CH_3COO)_4$ upon heating to ~200 °C (*J. Clust. Sci.* 2016, 27, 1553). Also, such a sharp peak is closely related to that event found for neat ZW-UiO-67 at 180.0 °C and ZW-UiO-67·0.5MSA at 186.2 °C, respectively (Supplementary Fig. 5). Considering the formula mass percentage of the acetate ($-OOCCH_3$ wt% = 1.4 wt% in ZW-UiO-67·MSA) is close to ~1.8wt% weight loss recorded before 250 °C, we speculated such endothermal event without weight loss may be related to the de-association of the acetate which is followed by less composite decomposition after that temperature. Online gas-phase FT-IR spectra detected H_2O and CO_2 with the temperature increase

(Supplementary Fig. 25), but no MSA was recorded even up to 320 °C (total weight loss ~4.5 wt%). For more clarity, careful analysis combining DSC, TGA and FT-IR were done, and the corresponding results were reorganized and discussed in the last paragraph highlighted on page 7 as:

“...At 185.4 °C, the endothermal sharp peak closely relates to that event for ZW-UiO-67 at 180.0 °C and ZW-UiO-67·0.5MSA at 186.2 °C, respectively (Supplementary Fig. 5). Considering the formula mass percentage of acetate (-OOCCH_3 wt% = 1.4% in ZW-UiO-67·MSA) is close to the ~1.8wt% weight loss recorded before 250 °C, it could be speculated that the endothermal event without weight loss at 185.4 °C may relate to the de-association of the acetate group which is followed by less composite decomposition after that temperature. Online gas-phase FT-IR spectra demonstrate that H_2O and CO_2 can be detected with the temperature increase (25 °C to 320 °C) (Methods and Supplementary Fig. 25).”

More detailed thermal events are also provided in Methods (page 22-23) for further review.

Supplementary Fig. 5 | DSC curves of the $\text{H}_2\text{BPDC-MIMS}$ linker and ZW-UiO-67 with and without incorporation of MSA. a, Up-scan curve of ZW-UiO-67·MSA from -150 to 320 °C, with thermogravimetric analysis (TGA) trace shown as yellow dashed line, being consistent with Fig. 1a. Inset photography is the soaking of melt-quenched solids (from 130 °C) in methanol for 3 minutes at room temperature, obtaining precipitates with weakened Bragg scattering as shown in Supplementary Fig. 4a. **b**, Special cyclic DSC curve of ZW-UiO-67·MSA from 0 to 150 °C shown with T_m , T_g phase transition temperatures and fusion enthalpy. **c-d**, Compared curves of ZW-UiO-67·0.5MSA and ZW-UiO-67 ($0 \sim 320$ °C) (up-scan) along with TGA traces of yellow dashed lines

(25 ~ 800 °C). The plot of ligand H₂BPDC-MIMS (blue line) is presented as a reference, with asterisk inferring the phase transition and the dashed-lines guiding that similar events occurred for ZW-UiO-67-0.5MSA and ZW-UiO-67 at this temperature. Please see the detailed discussion and analysis of DSC in the Method section.

Supplementary Fig. 25 | Online gas-phase FT-IR spectra with temperature increasing. In-situ gas-phase FT-IR of ZW-UiO-67·MSA measured from 25 to 320 °C, with the dominant stretching vibration bands of H₂O (1250 -2000 cm⁻¹, 3400-4000 cm⁻¹) and CO₂ (~2200 cm⁻¹).

2. The low-density liquid (LDL) to high-density liquid (HDL) transition needs to be proven.

Response: We thank the reviewer for this professional comment. This event relates to two exothermal events with onset temperature at 162.5 and 256.7 °C in the DSC plot of ZW-UiO-67·MSA (Fig. 2a, Supplementary Fig. 5). Due to the lack and difficulty of related characterizations for the phase transitions from the low-density liquid (LDL) to high-density liquid (HDL), we have reconsidered and carefully analyzed the TGA, DSC and FT-IR data, and rewritten the discussion to give a more appropriate explanation in the revised manuscript. For the first exothermal event (162.5 °C) without weight loss, it may be ascribed to the polyamorphic phase transitions between distinct structural configurations of neighbor states with different potential energies, as stated in the last paragraph of page 7:

“...The following exothermal event with onset at 162.5 °C (without weight loss) may be ascribed to polyamorphic phase transitions between distinct structural configurations of neighbor states with different potential energies (Fig. 2a, Supplementary Fig. 5)....”

For the second exothermal event with the onset temperature at 256.7 °C, it may relate to some thermal decomposition taken place before 282.9 °C (total 3.2_{wt}%) because similar thermal events are also observed for ZW-UiO-67-0.5MSA (onset: ~250 °C) and neat ZW-UiO-67 (onset: ~266 °C). Corresponding discussion has been added and highlighted in Method on page 23 as:

“...For the last exothermal event with onset temperature 256.7 °C of ZW-UiO-67·MSA (Fig. 2a), it may relate to some thermal decomposition taken place before 282.9 °C (total 3.2_{wt}%). Similar thermal events are also observed for ZW-UiO-67-0.5MSA (onset: ~250 °C) and ZW-UiO-67 (onset:

~266 °C)...”

3. “However, the endothermal evidence (218.3 °C) is reminiscent of the phase transition for ZW-UiO-67 (~212.6 °C) (Supplementary Fig. 5)”, what is the new phase?

Response: The third endothermal peak for ZW-UiO-67·MSA and ZW-UiO-67·TFSA is at 218.3 and 210.6 °C, respectively (Fig. 2a, Supplementary Fig. 5). In addition, it closes to the feature at 212.6 °C found in ZW-UiO-67, ZW-UiO-67·0.5MSA and H₂BPDC-MIMS (Supplementary Figs. 5). Therefore, we temporarily ascribe it to a plastic phase transition (*Phys. Chem. Chem. Phys.* 2010, 12, 11291–11298) reminiscent of that found for solid ionic liquids (*Korean J. Chem. Eng.* 2006, 23, 940–947) due to the flexible molecular symmetries of the zwitterion-acid subsystem and complex intermolecular interactions (i.e. H-bonding, Coulomb force etc.) within ZW-UiO-67·MSA. Related discussion is shown and highlighted on page 23 as:

“...The third endothermal peak at 218.3 °C of ZW-UiO-67·MSA and 210.6 °C of ZW-UiO-67·TFSA also are similar and close to that at 212.6 °C found in ZW-UiO-67 and ZW-UiO-67·0.5MSA and H₂BPDC-MIMS (Supplementary Fig. 5), which may be ascribed to a plastic phase transition reminiscent of that found for solid ionic liquids^[49]...”

4. Is there a mistake in "(1-mthyl-3-imidazolio) propane-3-sulfonate)" in line 85?

Response: We have corrected this mistake with '(1-methyl-3-imidazolio)propane-3-sulfonate' in the revised manuscript.

5. The authors emphasize that the Zr-O-Zr connection in agZW-UiO-67·MSA and agZW-UiO-67·TFSA is still reserved, but the Zr-O-Zr region in Extended Data Fig.19 is negligible. They need more clear evidences to support their points.

Response: We sincerely thanks for your thoughtful consideration of the Wavelet transform EXAFS data. As you said, it is true that the EXAFS data of agZW-UiO-67·MSA and agZW-UiO-67·TFSA have weak peaks of Zr-O-Zr after the wavelet transformations in Extended Data Fig.19, but we don't think that they can be considered as a negligible. Firstly, Zr-O-Zr belongs to second coordination shell, the scating path of which is instinctively weaker than that of the first coordination shell of Zr-O, as that shown radial distance $\chi(R)$ space spectra for crystalline UiO-67 and ZW-UiO-67 in Fig.3c,d. While, this phenomenon is pronounced in agZW-UiO-67·MSA and agZW-UiO-67·TFSA due to the distorted octahedron Zr₆O₄(OH)₄ along with some broken Zr-O bonds and decreased coordination number (2.95, 3.46) (Extended Data Table 3) upon vitrification. For more clarity, the strengths of different coordination shell path contribution of Zr-O-Zr, Zr-O and Zr-O-C in $\chi(R)$ space spectra of UiO-67, agZW-UiO-67·MSA and agZW-UiO-67·TFSA are compared and shown in Supplementary Fig 19a of new version. So, the wavelet transform EXAFS spectra (Extended Data Fig.19 you mentioned) shows weak region of the Zr-O-Zr. Secondly, Quantitative $\chi(R)$ space 2D profiles comparison in Fig. 3e,f also explicitly shows that weaken but not negligible Zr-O-Zr path contribution. Related discussion in page 11 as:

“...Such cluster changes are also intuitively witnessed by the Quantitative $\chi(R)$ space spectra fitting, wavelet transform extended X-ray absorption fine structure (WTEXAFS) and reduced coordination number (CN) of Zr-O bond (7.01-7.72) and Zr...O...Zr scattering path (2.95-3.18) in agZW-UiO-67·MSA and a_{gr}ZW-UiO-67·MSA as compared the perfect UiO-67 (8, 4) (Fig. 3e-f, Supplementary

Figs.20-21, Table 3)...”.

Supplementary Fig. 19 | XANES and $\chi(R)$ space spectra in comparison. **a**, The strength of different coordination shell path contribution of Zr-O-Zr and Zr-O in $\chi(R)$ space spectra in comparison. **b**, XANES $\mu(E)$ spectra observed at the K edge of the Zr atoms in UiO-67, ZrO_2 , **ZW-UiO-67** and **ZW-UiO-67·0.5MSA** at 298 K.

6. agZW-UiO-67·MSA was immersed in methanol and refluxed to obtain agrZW-UiO-67·MSA. Is it because agZW-UiO-67·MSA was dissolved in methanol and then re-formed nanocrystals? Are other solvents used (polar solvents, non-polar solvents, H₂O, etc.) to rule out the possibility of ligand dissolution leading to recrystallization?

Response: We thank the reviewer for this professional comment. Firstly, agZW-UiO-67·MSA is indissolvable in methanol, and the special short-time solvent stimulation (3 minutes) at room temperature can facilitate the recovery of the Bragg diffraction of the melt-quenched agZW-UiO-67·MSA, as shown in Supplementary Figs. 4 and 5. A UiO-67 framework is unable to be reconstructed at such a moderate condition in a short time. Therefore, such a recovered structure is derived from a preserved framework of the parent MOF, rather than a reaction or recrystallization of a dissolved/decomposed structure. In this context, the methanol may remove part of the organic acid (60% MSA remained, Method) embedded in the distorted framework of the formed glass and stimulate the phase transition to a more ordered structure. To further rule out the possibility of ligand dissolution leading to recrystallization, methanol vapor was used, and the low angle X-ray diffraction peaks recovered after 30 min exposure (Supplementary Fig. 15 as shown below). Furthermore, other different kinds of polar and non-polar solvents are also used to investigate the recovery, and polar solvents would be beneficial for this process. Related detail discussion has been included and updated in middle of page 8 in the revised manuscript:

“...Elemental analysis demonstrates ~60% MSA was preserved within **agrZW-UiO-67·MSA**, with the rest released during reflux. Such embedded MSA and the relative low crystallinity resulting from the solvent stimulation accounts for the reduced porosity. Furthermore, the glass to crystalline phase transformation upon external stimulation can be extended to other kinds of solvents, and polar solvents are advantageous for this process (Supplementary Fig. 15). By exposing to methanol vapor for 30 min, the X-ray diffraction peaks at low angles appeared with a broadened feature...”.

Supplementary Fig. 15 PXRD patterns of $a_{gr}ZW-UiO-67 \cdot MSA$ after solvent stimulations in comparison. Products were obtained via soaking $a_{gr}ZW-UiO-67 \cdot MSA$ in different solvents with stirring at room temperature for 20 minutes. Compared to the patterns of the sample soaked of 3 minutes in methanol (Supplementary Fig. 4), peaks at high 2θ degree appear for those of 20 minutes in various solvents. Methanol vapor stimulation is done by exposing sample to vapor for 30 minutes.

7. From the relative low porosity of $a_{gr}ZW-UiO-67 \cdot MSA$ than $ZW-UiO-67 \cdot MSA$, authors guess the 60% of the MSA still remains in the framework. What is the degree of crystallinity for the $a_{gr}ZW-UiO-67 \cdot MSA$? This is another possible reason for the lower porosity.

Response: We thank the reviewer for this constructive comment. We fully agree with the reviewer that the low porosity could be caused by the low crystallinity in this case. As mentioned previously, 60% MSA remained within $a_{gr}ZW-UiO-67 \cdot MSA$, which mainly contributes to the reduced porosity. However, low crystallinity is also an important reason for the low porosity in $a_{gr}ZW-UiO-67 \cdot MSA$, and we have included additional discussion in page 8 of new version as:

“...Such embedded MSA and the relative low crystallinity resulting from the solvent stimulation accounts for the reduced porosity...”

8. $ZW-UiO-67 \cdot 0.5MSA$ and $ZW-UiO-67 \cdot MSA$ have the different thermodynamic behaviors. $ZW-UiO-67 \cdot 0.5MSA$ cannot melt. The authors should provide further experiments to discuss the role of MSA in the melting behaviors.

Response: We thank the reviewer for this insightful comment. For better clarity, we have rewritten and highlighted the melting mechanism section on page 9.

Firstly, from the point of view of acidic ionic liquids, $ZW-UiO-67$ can be regarded as a zwitterion-decorated MOF with the MIMS groups. The incorporated MSA acid pairs with the MIMS group to form a Brønsted acid-base buffer subsystem, MIMS·MSA, as evidenced by the DSC data of them and acidic ionic liquids (Ref. 25, *Chem. Rev.* 2016, 116, 6133–6183). The more MSA molecules pair with the zwitterionic ligands through electrostatic and/or hydrogen-bonding interactions, the greater charge delocalization and separation occur. This substantially disrupts the long-range structural order and consequently leads to increased configurational entropy (S_{conf}), which

dramatically reduces the T_m (Ref. 26, *J. Non-Cryst. Solids* 2017, 463, 175–188) and facilitates the solid-to-liquid phase transition, resulting in the melting of ZW-UiO-67·MSA at ~ 127 °C but unmeltable ZW-UiO-67·0.5MSA due to its insufficient acid. This conclusion has been further demonstrated by the formation of zwitterion-based acidic ionic liquids: the solid zwitterionic EIMS and H₂BPDC-MIMS respectively give rise to room temperature acidic ionic liquids EIMS·MSA and H₂BPDC-MIMS·MSA as the amount of MSA is 3.78 times that of zwitterions (Supplementary Fig. 1 and Table 2); while a solid product, i.e. a relative high melting temperature, is observed since half amount of MSA (1.89 times) is insufficient to lead to the melting as observed in our previous work (Ref. 29, *Angew. Chem. Int. Ed.* 2021, 60, 1290–1297). Meanwhile, the densely filled channels in ZW-UiO-67·MSA not only are beneficial for the coordination bonding mismatch between COO⁻ and metal nodes upon heating, but also enhance the disordered structure and increase the configurational entropy to markedly reduce the energy barrier facilitating the melting.

Secondarily, to further reason the role of MSA in ZW-UiO-67·0.5MSA, more sensitive ¹H-¹³C cross polarization magic-angle-spinning (CP MAS) spectra have been complemented. The missed COO resonance (~ 173 ppm) of CP MAS in ZW-UiO-67 re-appears in the unmeltable ZW-UiO-67·0.5MSA (Fig. 4e, Supplementary Fig. 26b). Such a resonance signal indicates the protonation of the COO groups in the half-filled ZW-UiO-67·0.5MSA, but the insufficient acid cannot realize the melting as that for ZW-UiO-67·MSA. The updated CP MAS spectra is shown in Supplementary Fig. 26b, and related discussion is presented in middle of Page 14 revised as the follows:

“...around 173.9 ppm and aromatic carbon resonances from 120 to 150 ppm (Fig. 4e). Meanwhile, more sensitive ¹H-¹³C cross polarization magic-angle-spinning (CP MAS) spectra of a_gZW-UiO-67·MSA show comparable features to those of unmeltable ZW-UiO-67·0.5MSA, but a slightly different resonance at ~ 182 ppm (Supplementary Fig. 26b) which implies much small amount of uncoordinated COO groups^[35,39] due to that solid-liquid phase transition...”

Supplementary Fig. 26 | Solid-state ¹³C NMR spectra at room temperature. **a**, Solid-state ¹³C NMR spectra of CP MAS (at 12 kHz) of ZW-UiO-67, and DD MAS (at 12 kHz) of ZW-UiO-67, ZW-UiO-67·TFSA and a_gZW-UiO-67·TFSA. Enclosed area indicates the missed CP MAS resonance of C_{COO} in ZW-UiO-67 and the appeared DD MAS signal in comparison. Spinning sideband are marked with asterisks. Signals signed with triangle are ¹³C resonances of DMF (*N,N*-dimethylformamide) solvent. 121.2 ppm is the resonance signal of -CF₃ in TFSA (*N,N*-

bis(trifluoromethanesulfonyl) amide). **b**, Appeared C_{COO} resonance in CP MAS of ^{13}C NMR spectra of $a_gZW-UiO-67 \cdot MSA$ and $a_gZW-UiO-67 \cdot TFSA$ and $ZW-UiO-67 \cdot 0.5MSA$ due to protonated C_{COO} . Enclosed rectangle indicates the appeared C_{COO} resonance, as compared to that missed signal of $ZW-UiO-67$ (See also Fig. 4e).

9. As a control experiment, UiO-67 is mixed directly with EIMS-MSA to check the melting behaviors. VT-PXRD shows that some peaks of the mixture disappear during heating in Extended Data Fig.7. Is this possibly caused by uneven mixing and not complete melting? Is there any evidences that EIMS-MSA is evenly distributed in UiO-67?

Response: We agree with the reviewer and have reconsidered this critical point. We are inclined to believe this is due to a very tiny amount of residual inhomogeneous mixing during the preparation (the acids diffuse into the MOF channels) at 60 °C (Method). So the Bragg peaks between 5-10 degrees continue to become weaker with increasing temperature which can be evidenced in Supplementary Fig. 8. For clarity, a caption for Supplementary Fig. 8 “The weakening of the low-angle peaks in panel **d** may be due to the complete homogenization of ILs@UiO-67 resulted from high temperature heating” is presented.

Moreover, we do not think this is due to ‘incomplete melting’ due to the following two reasons. Firstly, other high-angle Bragg peaks show almost no changes in intensity; secondly, the melting behavior of ILs@UiO-67 in the thermal analysis was unable to be detected. Nevertheless, we must clarify that EIMS-MSA was indeed almost completely diffused into UiO-67, as evidenced by the BET and pore size distributions before and after the impregnation of MSA, i.e. ignorable surface area was detected, as shown in Supplementary Fig. 14 and Supplementary Table 1. Furthermore, more kinds of HA/ILs@UiO-67 have been investigated, which all exhibited as unmeltable solids similar to that of EIMS-MSA@UiO-67, as shown in Supplementary Table 5:

Supplementary Table 5 | Compared melting behaviors of HA/ILs@ZW-UiO-67 to that of HA/ILs@UiO-67 (see also Supplementary Figs. 5-6, 29, Table 1, 2, 4).

MOF ILs/HA	Meltability of ILs@ZW-UiO-67 and ILs@UiO-67 (heated to 320 °C)		Meltability/ T_m (°C) of ZW-UiO-67·HA and HA@UiO-67			
	EMIM·TFS*	EIMS·MSA*	MSA	TFSA	TFA	ESA
ZW-UiO-67 ^a	No	/	Yes /127.2	Yes /149.6	Yes /157	Yes /139
UiO-67 ^b	No	No	No/-	No/-	No/-	No/-
ρ (g/ml) ^c	1.387	1.42	1.48	1.892	1.696	1.388

pK_a	/	/	-1.9	-12.2	-14	1.8
--------	---	---	------	-------	-----	-----

* EMIM·TFS = 1-ethyl-3-methylimidazolium trifluoromethanesulfonate, EIMS = 1-(1-ethyl-3-imidazolium)propane-3-sulfonate. ^a For synthesis of **EMIM·TFS@ZW-UiO-67**, de-solvated **ZW-UiO-67** (80 mg) and 34 μ L ILs (**EMIM·TFS**) (equal to the pore volume of MOF) were used. ^b For **ILs@UiO-67** and **HA@UiO-67**, activated UiO-67 (80 mg) and 81 μ L (equal to the pore volume) HA or ILs were respectively used. All samples were obtained by the same incipient wetness technique (see also Method).

10. In Extended Data Fig. 24, the NMR spectrum is not clearly discussed. The peak at 182.3 ppm for a_g ZW-UiO-67·MSA is difficult to observe.

Response: We thank the reviewer for pointing out this question. This weak peak at 182.3 pm may be due to the very small amount of uncoordinated COO groups which only gives to limited NMR signal. For clarity, ^1H - ^{13}C cross polarization magic-angle-spinning (CP MAS) spectra of a_g ZW-UiO-67·MSA was re-collected with a long-time scan, as shown in the updated Supplementary Fig. 24b. For further comparison, CP MAS spectra of ZW-UiO-67·0.5MSA was additionally measured and compared, which clearly showed the absence of a 182.3 ppm peak (Supplementary Fig. 26b).

Reviewer #3 (Remarks to the Author):

In this manuscript, the authors reported the realization of melting and the preparation of quenched glass of three typical metal-carboxylate MOF of UiO-67, UiO-68 and DUT-5. The research follows one of the important trends in MOF chemistry, that is transforming traditional crystalline MOFs into their liquid or glassy states. Relative efforts have gained success firstly in a limited numbers of ZIFs and some 1D/2D coordination polymers, and recently in traditional MOFs with both single metal and cluster as nodes. Even so, the area is quite new with a lot of puzzles remain to be explored. From this point, the presented result is interesting and significant to attract wide readership.

For the investigated content, all the three compounds have long been known as quite thermal stable MOF constructed from [Zr6] node, yet decomposition rather than melting occur upon heating. Such fact indeed shows the current situation that most MOFs, especially the vast number of metal-carboxylate MOFs, can not be melted. This work developed a Brønsted acid-base strategy by grafting the zwitterions on the carboxylate ligands and incorporating organic acids in the framework channels to enable the melt-quenched glass formation of UiO-67, UiO-68 and DUT-5. The method somewhat draw lessons from traditional ionic liquid with long research history, and similar strategy has been successful in ZIF-8 (Ref. 42) which is also known without melting. Even so, present research results are no doubt important and valuable to enhance understanding to MOF glass preparation. Especially, the research is performed in a comprehensive and thorough way with abundant characterization and experimental evidences, which is infrequent and laudable in peer study. Hence, I think this paper is a suitable candidate for publication in Nat. Commun. Some queries needed to be answered are listed as following:

Response: We thank the reviewer for these very positive comments, which will significantly inspire our future work.

1. As a great aspiration, the authors intend to construct a “generally applicable strategy” to vitrification of wide range of MOFs which are “composed of different ligands and metal nodes” and “not limited to the structure type and porosity”. However, presented three MOFs are in fact quite similar either in composition and structure. Particularly, they possess very rigid framework to endure zwitterions grafting and organic acids inclusion. While most known metal-carboxylate framework did not have such high structural stability. The method applicability is open to doubt, or at least need much more examples. On this situation, I think the presented title is too broad and the conclusion is exaggerated.

Response: We agree that the three MOFs presented in our study are indeed quite similar in terms of both composition and structure, possessing a high level of structural stability required for zwitterion grafting and organic acid inclusion. We acknowledge that this may raise doubts regarding the applicability of our method to a broader range of MOFs. As suggested by the reviewer, the title has been changed to ‘Melt-Quenched Glass Formation of a Family of Metal-Carboxylate Frameworks’. The conclusion in page 16 has also been revised accordingly as the following:

“...a family of metal-carboxylate frameworks (UiO-67, UiO-68 and DUT-5) were successfully made into glasses *via* the melt-quenched method. Importantly, our strategy could be applied to a range of highly porous MOFs composed of different ligands and metal nodes. In addition, this strategy could also be extended to other kinds of highly porous molecular frameworks such as covalent-organic frameworks. Our findings would also offer new opportunities for processing MOFs and other hybrid organic-inorganic crystals by melt-quenching them into glasses, which

could enable them to meet the manufacturing requirement by industry.”

2. All the DSC results are quite strange comparing to known MOF glasses and also other traditional glass system. These abnormal data further lead to great queries in several important thermal parameters of T_m and T_g for all reported compounds. I noticed two instruments of DSC-Q200 and METTLER TOLEDO DSC3 were applied to collected data. How about the data repeatability, as well as the influence factor of flowing gas, heating rate?

Response: We thank the reviewer for this constructive comment. Firstly, for the DSC data and repeatability, the top DSC plot of Fig. 2a was recorded on an instrument (DSC-Q200) able to decrease the temperature to $-150\text{ }^\circ\text{C}$ with the aim to detect the possible phase transition at the low temperature region; however, the DSC curve in the bottom panel was measured via a different instrument (METTLER TOLEDO DSC3) for recording the cyclic plot only from $0\text{ }^\circ\text{C}$. According to the reviewer’s comment, different flowing gas and/or heating rate for the DSC measurements of ZW-UiO-67·MSA, ZW-UiO-67·TFSA and ZW-UiO-67·TFA have been done. For ZW-UiO-67·MSA, different samples were used to repeat the DSC results with the aim to view the possible effect of different batches. The T_m was slightly influenced by the heating and gas flowing rate, with the value varying in a narrow range ($127\text{ }^\circ\text{C}$ - $132\text{ }^\circ\text{C}$), but distinct heating rates led to T_g lying within a relative larger region ($73\text{ }^\circ\text{C}$ - $104\text{ }^\circ\text{C}$). For ZW-UiO-67·TFSA and ZW-UiO-67·TFA, the carefully re-recorded plots of up scans with variable heating rates. Those newly repeated DSC data were all collected from instrument METTLER TOLEDO DSC3, showing a good repeatability (see Supplementary Fig. 7). Related discussion has been updated in the bottom of first paragraph of page 8 as:

“...Different flowing gas and/or heating rates exhibited slight influence on T_m , with the value varying in a narrow range ($127\text{ }^\circ\text{C}$ - $132\text{ }^\circ\text{C}$), but distinct heating rates led to T_g lying within a relative larger region ($73\text{ }^\circ\text{C}$ - $104\text{ }^\circ\text{C}$) for ZW-UiO-67·MSA (Supplementary Fig 7)...”

Secondly, for the strange feature of DSC, here we show a glass family derived from a MOF incorporated with the Brønsted acid, i.e. a solid ionic liquid with T_m over room temperature, rather than a porous crystalline MOF such as ZIF-62 etc. Here, ZW-UiO-67·HA bears the rigid coordination framework and acid-zwitterion buffer subsystem, MIMS·HA, with complex intermolecular interactions and more flexible molecular configurations. Therefore, such a soft-in-hard structure displays a more complex thermal behavior markedly different from that of porous crystalline ZIFs. This leads to a broad distribution of molecular motions and relaxation processes during the heating process, showing a broad and weak melting endothermal peak and T_g feature. That may be reminiscent of that of porous organic cages with flexible structures upon heating (*J. Mater. Chem A*, 2021, 9, 19807-19816; please also see page 7 revised). Furthermore, the efficiently grinding, completely and homogeneously diffusing of liquid acids into the channels of MOFs may also influence the final complete structural homogeneity of ZW-UiO-67·HA (vs. crystal) which subsequently affects the DSC features because the sample is obtained via an incipient wetness technique (Method). To obtain the repeatable DSC results in the revised manuscript, all those re-recorded samples were obtained under nitrogen atmosphere protection. The moisture from air may also effect the components of ionic ZW-UiO-67·HA. Special mention was presented at their synthesis of Method at page 27 of new version.

Supplementary Fig. 7 | Different flowing gas and/or heating rate for DSC of ZW-UiO-67·MSA and ZW-UiO-67·TFSA. **a**, Repeated DSC measurements by use different samples to show the effect of heating rate and different batch on the T_m , and **b**, the effect on T_g of ZW-UiO-67·MSA. Asterisk infer the gas flowing rate of 80 ml/min that is different from those of all others (50 mL/min). **c**, Repeated cyclic DSC plot from 0 °C to 170 °C and the following down to 0 °C, and then up scan to 170 °C (black line) with heating rate of 10 °C/min for ZW-UiO-67·TFSA. The red and turquoise plots are for the second and third cyclic data with different heating rates, while the blue line is the last up scan of the a_g ZW-UiO-67·TFSA. **d**, comparison of four up scan plots for ZW-UiO-67·TFA as that in panel **c**.

3. For the typical DSC of ZW-UiO-67·MSA in figure 2a, firstly the really melting-range is indeed confusing for readers. Why the T_m point is in a platform position? More importantly, the authors listed three temperature points of 162.5, 185.4 and 218.3, which seems located at TG range without weight loss. As well known, the melt liquid goes in a thermal stable energy state as homogeneous phase with steady DSC trace. What happens at these points with evident endothermic and exothermic signals? The two-paragraph discussion on thermal behaviors in Page 6 are convoluted.

Response: We thank the reviewer for these critical comments. For the first comment, in the bottom panel of Fig. 2a, the platform position of the offset T_m point is due to the system error between two different instruments, with the 127.2 °C for the top and 125 °C for the bottom. In the revised manuscript, the T_m point of the bottom figure is marked as its actual position (~125 °C) and mentioned at the Fig. 2a caption. For the second comment, please see our responses to questions 1,

2 and 3 of the reviewer 2. More careful analysis based on TGA, DSC and FT-IR spectra has been done, and a more appropriate explanation has been given and highlighted on page 7 and 23 of the revised manuscript. Lastly, we reorganized that two-paragraph discussion on thermal behaviors in page 6 of original manuscript. New version of which is presented as two paragraphs, with the first for the component and stability discussion and the followed paragraph for DSC analysis, as shown in page 7 of revised manuscript as:

“(Supplementary Figs. 8a, 11). Notably, no melting or evaporating event of MSA ($T_m \approx 20^\circ\text{C}$, $b_p \approx 167^\circ\text{C}$) was recorded in the DSC and TGA trace before 180°C . No loss of MSA implies that the Brønsted acid-zwitterion subsystem MIMS·MSA has been successfully incorporated in **ZW-UiO-67·MSA** (Supplementary Fig. 1). This is similar to the scenario for the zwitterion-based acidic ionic liquids^[25] including H₂BPDC-MIMS·MSA which shows no thermal event of a separate specie upon heating (Supplementary Fig. 6d). Therefore, the behavior of **ZW-UiO-67·MSA** is in stark contrast to the thermal properties of neat **ZW-UiO-67** or MSA alone. It can be viewed as a solid ionic liquid^[29] featuring the Brønsted acid-zwitterion structure, but not a simple host-guest composite like those ILs@MOFs^[27].

For the DSC plot within $0\sim 320^\circ\text{C}$, the first feature of the endothermal melting (offset $\sim 127^\circ\text{C}$) is broad and weak, which could be related to the lattice fluctuation with multiple and gradual enthalpy changes, being similar to the melts of some porous organic cages with flexible structures upon heating^[30]. The measured fusion enthalpy change of 6.35 J/g is also comparable to those of 2D metal-triazolates ($5\text{-}10\text{ kJ/mol}$)^[14], zeolitic zinc 2-ethylimidazole^[31] and some silica zeolites^[32]. The following exothermal event with onset at 162.5°C (without weight loss) may be ascribed to polyamorphic phase transitions between distinct structural configurations of neighbor states with different potential energies (Fig. 2a, Supplementary Fig. 5). Therefore, the melting phase of **ZW-UiO-67·MSA** before 162.5°C can be regarded as a stable liquid phase I. At 185.4°C , the endothermal sharp peak closely relates to that event for **ZW-UiO-67** at 180.0°C and **ZW-UiO-67·0.5MSA** at 186.2°C , respectively (Supplementary Fig. 5). Considering the formula mass percentage of acetate ($-\text{OOCCH}_3$ wt% = 1.4% in **ZW-UiO-67·MSA**) is close to the $\sim 1.8\text{wt}\%$ weight loss recorded before 250°C , it could be speculated that the endothermal event without weight loss at 185.4°C may relate to the de-association of the acetate group which is followed by less composite decomposition after that temperature. Online gas-phase FT-IR spectra demonstrate that H₂O and CO₂ can be detected with the temperature increase (25°C to 320°C) (Methods and Supplementary Fig. 25).”

4. The authors said “the following exothermal and another event with onset 256.7°C may be polyamorphic phase transitions from the low-density liquid (LDL) to high-density liquid (HDL) driven by different potential energies between distinct structural configurations of neighbor states”. However, comparing to Ref-5, the signals at 256.7°C shows completely different characters. The dramatic exothermic trends are really strange. Without sufficient and credible evidence, such statement of LDL and HDL is improper.

Response: We thank the reviewer for this critical comment. Please see our response to question 2 from Reviewer 2.

5. In page 5, the formular for ZW-UiO-67·MSA seems incorrect, the BPDC should contain the grafted MIMS. In addition, why 1MSA and 0.5MSA per ZW-UiO-67 in page 7. Considering there

are a numbers of compounds in the paper, a list showing all the abbreviation, accurate formular in necessary for readership.

Response: The formula for ZW-UiO-67·MSA has been changed to $Zr_6O_4(OH)_4(BPDC-MIMS)_{5.4} \cdot (O_2CCH_3)_{1.2} \cdot (MSA)_{20.5}$. In addition, we acknowledge the importance of providing a comprehensive list of abbreviations and accurate formulas for all compounds mentioned in the paper. To address this, we have included a dedicated Supplementary Table 2 that lists all abbreviations used for the complexes discussed throughout the manuscript.

Supplementary Table 2 | Compositions of ZW-UiO-67, ZW-UiO-67·0.5MSA, ZW-UiO-67·HA, a_{gr} ZW-UiO-67·MSA, and EIMS·MSA@UiO-67. HA = MSA, TFSA, TFA, ESA.

	ZW-UiO-67 (mg)	HA (μ l)	Stoichiometry Formula ^a (L = BPDC-MIMS)	Formula weight	ZrO ₂ (wt%) ^b Calc./meas.	
ZW-UiO-67	/	/	$Zr_6O_4(OH)_4(L)_{5.4} (O_2CCH_3)_{1.2}$	~3124.8	~23.7/27.6	
ZW-UiO-67·MSA	80	34	$Zr_6O_4(OH)_4(L)_{5.4} \cdot (O_2CCH_3)_{1.2} \cdot (MSA)_{20.5}$	~5092.8	~14.5/15.7	
ZW-UiO-67·0.5MSA	80	17	$Zr_6O_4(OH)_4(L)_{5.4} \cdot (O_2CCH_3)_{1.2} \cdot (MSA)_{10.2}$	~4104.0	~18.0/20.1	
ZW-UiO-67·TFSA	80	34	$Zr_6O_4(OH)_4(L)_{5.4} \cdot (O_2CCH_3)_{1.2} \cdot (HTFSA)_6$	~4923.2	~15.0/16.0	
ZW-UiO-67·TFA	80	34	$Zr_6O_4(OH)_4(L)_{5.4} \cdot (O_2CCH_3)_{1.2} \cdot (TFA)_{15.0}$	~5374.8	~13.8/14.1	
ZW-UiO-67·ESA	80	34	$Zr_6O_4(OH)_4(L)_{5.4} \cdot (O_2CCH_3)_{1.2} \cdot (ESA)_{16.3}$	~4916.6	~15.0/26	
EIMS·MSA@UiO-67	UiO-67 (80 mg) & ILs (81 μ L)		$Zr_6O_4(OH)_4(BPDC)_{5.4} \cdot (O_2CCH_3)_{1.2} \cdot [EIMS]_{4.93} [MSA]_{18.7}$	~4935.7	~15.0/8.73	
a_{gr}ZW-UiO-67·MSA	/	/	$Zr_6O_4(OH)_4(L)_{5.4} \cdot (O_2CCH_3)_{1.2} \cdot (MSA)_{12.3}$	~4305.6	Unmeasured	
HA or ILs used in this study						
HA/ILs^c	EIMS·MSA	EMIM·TFA	MSA	TFSA ^d	TFA	ESA
ρ (g/ml)	1.42	1.38	1.48	1.892	1.696	1.35

^aStoichiometry formula is based on ZW-UiO-67 which was determined through ¹H NMR (Supplementary Fig. 2), and volume of HA or ionic liquid equals to the pore of ZW-UiO-67 used.

^b Data is obtained from the TGA traces (Supplementary Fig.16). ^c Ionic liquids (ILs) were synthesized as that illustrated in Supplementary Fig. 1. ^d Melt point of TFSA (bis(trifluoromethane)sulfonimide) is 52 °C, here used as a liquid under 55 °C.

6. For the description of proton conduction in Page 7, the reported data is not clear for which sample.

Response: We are sorry for this misunderstanding. The reported data are for a_gZW-UiO-67·MSA, which has been explicitly mentioned in page 8-9 of the revised manuscript.

“...The anhydrous proton conductivity of a_gZW-UiO-67·MSA reaches $1.57 \times 10^{-2} \text{ Scm}^{-1}$ at 100 °C, acting as a fast ionic conductor with application prospect^[29]. Almost one order-of-magnitude increase in conduction from 1.96×10^{-3} to $1.65 \times 10^{-2} \text{ Scm}^{-1}$ was observed as the temperature elevated from 80 to 110 °C. Such a marked change in conduction correlates to the glass transition near $T_g (> T_g, 104 \text{ °C})$, reminiscent of that observed for the phase transition in reported solid state conductors^[33]...”.

7. For the recovery experiments, how about the solvent-dependent? Excepting methanol, can other solvent induce the glass-to-crystalline transformation. In addition, to maintain the molding film, is vapor possible to facilitate the recovery?

Response: Thanks for your professional comments and insightful suggestion. We appreciate your interest in the solvent-dependent recovery experiments. Based on our research, we have observed partial recovery of the low 2θ degree X-ray diffraction for a_grZW-UiO-67·MSA when exposed to methanol vapor for 30 minutes. In response to your question, we have also extended our investigations to include other polar and non-polar solvents such as ethanol, water, ethyl acetate, dichloromethane, and acetone. Our findings suggest that a polar solvent is more beneficial for the recovery process. Related updated Supplementary Fig. 15 demonstrates the reappearance of high-degree X-ray diffractions after immersing the samples in the solvent for 20 minutes with stirring at room temperature. The revised manuscript now includes a detailed discussion on this topic, specifically on page 8. Please also see that response to question 6 of reviewer 2 above.

8. It seems enough content of incorporating organic acids is crucial for the success of melting, how about the critical point for the three MOFs?

Response: The reviewer is right. Sufficient amounts of organic acids are critical, as illustrated by the difference in thermal behavior between ZW-UiO-67·MSA and ZW-UiO-67·0.5MSA. Regarding the critical points of the three MOFs, in this study we mainly focused on exploring the addition of same volume of organic acids as the pores of MOFs. To fully characterize the melting of a series of incorporated MOF·HA compounds, more structure determinations need to be involved, including X-ray absorption near edge structure (XANES) spectra, total scattering measurements, morphology and thermal behavior characterizations. Due to the resource and time constraints, we regret to say that our revision is unable to include all the demanding experiments in this context. However, we totally agree with the reviewer that this is a very worthwhile direction to explore as well as investigating the potential linear relationship between organic acid content and T_m/T_g . We will try to look into this in depth in future studies.

9. On the whole, the organization of different parts as well as discussion in the same part is not sufficiently good, which leads to easy lost of main idea of the authors. Though it may be ascribed to the large experimental data, further improvement on the organization and language is necessary.

Response: As the reviewer suggested, we have made several revisions to improve the organization and language of the paper. Specifically, we have reorganized certain paragraphs, such as the

component and stability section, as well as the thermal behavior discussion on page 7. Additionally, we have addressed the recovery phenomenon and conducted a more comprehensive study of the glass on page 8. Furthermore, we have revised our explanation of the melting mechanism, providing a clearer elucidation on page 9. We believe that these revisions contribute to a more effective communication of our research findings.

Reviewer #4 (Remarks to the Author):

There is a great deal of current interest in the new family of glasses derived from metal organic frameworks, but most of the ones that have been reported to date are based on organic linkers that bind to metals via nitrogen. For example, the most widely studied MOF glasses are those based on zeolites imidazolate frameworks (ZIFs). The manuscript submitted by Yue et al describes an ingenious way of making glasses from MOFs based upon carboxylate linkers, which constitute the largest family of MOFs. This work will therefore be of considerable interest to the MOF community. The new carboxylate glasses are mainly formed by starting with derivatives of MOF UiO-67 that have been functionalized by substituting zwitterionic amines onto the aromatic linkers and then introducing a complementary Brønsted acid, such as methylsulfonic acid, into the MOF cavity. The amine substituent and the Brønsted acid form zwitterionic pairs that facilitate the low temperature melting of the MOF. The melts can then in turn be quenched to yield glasses. One of the interesting features of these glasses is that the crystalline MOF precursor can be retrieved from the glass by leaching the acid out of the glass by using a simple organic solvent such as methanol. This reversibility is not usually found in ZIF glasses.

The characterisation of the MOF glasses in the manuscript is very thorough, even though the structures of the crystalline material precursors depend upon powder X-ray diffraction data. The use of other techniques, such as EXAFS, total scattering XRD, vibrational spectroscopy and solid state NMR, has been used to probe the local structures of the glasses and compare them to the crystalline precursors. Overall, I think that the work has been carried out very carefully and that the claims of glass formation are well supported by the data.

I think that this interesting work could be published in Nature Communications after a couple of important points have been addressed:

Response: We thank the reviewer for these enthusiastic comments on the significance of our work.

1. The authors claim that they have developed a universal strategy for making glasses from carboxylate MOFs (see pages 3 and 14). I think that this claim is exaggerated and needs to be toned down. For example, the MOF cavities need to be large enough to accommodate both the zwitterionic amine substituents AND the complementary Brønsted acid molecules. This approach can only therefore be used for MOFs with large cavities such as the UiO family. Carboxylate MOFs with smaller cavities, of which there are very many, will be unsuitable for this approach. The claim of universality must be replaced by a more suitable and justifiable claim.

Response: We agree with the reviewer that we should tone down the applicability of our strategy. We have reorganized our statements to make our claim more appropriate, such as the modified article title “Melt-Quenched Glass Formation of a Family of Metal-Carboxylate Frameworks” and the revised wording “By functionalizing the carboxylate ligands with zwitterionic groups and incorporating the Brønsted acids in the framework channels, a family of metal-carboxylate frameworks (UiO-67, UiO-68 and DUT-5) were successfully made into glasses via the melt-quenched method...” in the Discussion of the last paragraph of the revised manuscript.

2. Most of the work concerned MOFs from the UiO family and these are well-known to be prone to having large concentrations of defects in the form of missing octahedra (from the work of Goodwin).

This should be mentioned in the manuscript since these large defect vacancies may play a role in accommodating the amine substituents and the Brønsted acid molecules. I also wonder if any of the characterisation methods reveal any evidence for the presence of defects in the crystalline precursors.

Response: According to Goodwin's work (*Nat. Commun.* 2014, **5**, 4176), we have thoroughly examined our WAXS data for crystalline ZW-UiO-67 (Supplementary Fig. 27b). The analysis reveals that the fluctuations between 2.5 and 5 degrees (2θ) are nearly negligible, suggesting that cluster defects may be of minimal significance (*Nat. Chem.* 2019, **11**, 622–628). Furthermore, ZW-UiO-67 exhibits a structure with ligand-dominated defects, which has been characterized using ^1H NMR spectra. The stoichiometric formula of ZW-UiO-67 is denoted as $\text{Zr}_6\text{O}_4(\text{OH})_4(\text{BPDC-MIMS})_{5.4}$, with a portion of the BPDC-MIMS ligand being displaced by an acetate anion. Although we cannot dismiss the possibility of a small number of cluster defects in ZW-UiO-67, we believe that the majority of defects can be attributed to ligand defects identified through ^1H NMR, and such defects are not responsible for the meltability of ZW-UiO-67·HA because control ILs@ ZW-UiO-67 is unmeltable. According discussion on the ligand-based defect has been included in the middle of page 5 of the revised version as the following:

“...The stoichiometric formula of $\text{Zr}_6\text{O}_4(\text{OH})_4(\text{BPDC-MIMS})_{5.4}$ is determined via ^1H NMR spectra, with a small portion of BPDC-MIMS ligand displaced by the acetate group...”

Supplementary Fig. 27b. PXRD patterns of WAXS of crystalline ZW-UiO-67, amorphous a_gZW-UiO-67·HA, polycarbonate and silicate glass.

If these two points are properly addressed, I think that the paper could be accepted for publication in Nature Communications.

REVIEWER COMMENTS

Reviewer #1 (Remarks to the Author):

The authors have addressed all the issues, and thus I recommend it be accepted for publish without further change.

Reviewer #2 (Remarks to the Author):

Point-by-point revisions by the author are highly welcome. The author has added new citations for some metal-carboxylate framework glasses. Additional discussions for metal-carboxylate framework glasses have been added. But does the ambiguous naming of "Several melt-quenched metal complexes" give the impression that the authors are deliberately leading readers to ignore closely related published works? After reading these papers, I firmly believe that they are carboxylate MOF glasses. In addition, I have noticed that another paper based on porous dicarboxylate-bridged MOF glasses has just been published (Chem. Commun., 2023, 10.1039/D3CC04518H). The authors need to make appropriate discussions and revisions.

In addition, agrZW-UiO-67-MSA was obtained by refluxing agrZW-UiO-67-MSA in methanol. The current discussion, which is mainly based on PXRD, cannot support that agZW-UiO-67-MSA is a UiO-67-type framework. That is, it cannot be completely ruled out that the material gradually dissolves and recrystallizes in the refluxing methanol. In fact, it can be seen from the blurred SEM in Fig. 2b that agrZW-UiO-67-MSA are nanocrystals, quite different from that of agZW-UiO-67-MSA. The authors need to provide clear SEMs to study the morphological evolution of the particles and surfaces of agrZW-UiO-67-MSA to agZW-UiO-67-MSA.

The authors also found that other solvents are advantageous for the recrystallization process. However, PXRD showed that these solvents caused agZW-UiO-67-MSA to crystallize into a different phase than grZW-UiO-67-MSA. The authors need to try to resolve and discuss this crystalline phase.

I am curious if the coordination structure is maintained after melting? In situ analyses by PDF, XAFS, FT-IR, etc. may provide evidence. Investigation of the structure after liquefaction helps to understand what the real structure of agZW-UiO-67-MSA is.

Authors need to thoroughly address the above key issues, otherwise the work is not recommended for publication in such a high-quality journal.

Reviewer #3 (Remarks to the Author):

After carefully reading the authors response and revised manuscript, most of my previous queries have been well addressed, based on some complementary experiments and further analysis of all the observed results. I also happy to see the organization, writing and quality of the paper have been improved, benefited from the comments of other reviewers. I personally think the current version is suitable for publication in Nat. Commun.

Reviewer #4 (Remarks to the Author):

I am pleased that the authors have addressed the two important points made in my original review and I am satisfied with the changes that they have made in response to them. I also feel that they have responded effectively to the comments made by the other reviewers. I am therefore happy to recommend publication of the revised manuscript in Nature Communications.

RESPONSE TO REVIEWERS' COMMENTS

Reviewer #1 (Remarks to the Author):

The authors have addressed all the issues, and thus I recommend it be accepted for publish without further change.

Response: We thank the reviewer for the very positive comment.

Reviewer #2 (Remarks to the Author):

Q1

Point-by-point revisions by the author are highly welcome. The author has added new citations for some metal-carboxylate framework glasses. Additional discussions for metal-carboxylate framework glasses have been added. But does the ambiguous naming of "Several melt-quenched metal complexes" give the impression that the authors are deliberately leading readers to ignore closely related published works? After reading these papers, I firmly believe that they are carboxylate MOF glasses. In addition, I have noticed that another paper based on porous dicarboxylate-bridged MOF glasses has just been published (Chem. Commun., 2023, 10.1039/D3CC04518H). The authors need to make appropriate discussions and revisions.

Response: We agree with the reviewer we need to extend our discussions on the recently developed metal-carboxylate MOF glasses. We have cited the mentioned paper as reference 24. Also, related discussions have been reorganized on page 3 in the revised manuscript as the following:

“...Several carboxylate MOFs, including both carboxylates and bicarboxylates, have been made into glasses very recently, which are derived from discrete solvated complexes or coordination networks through rearrangements of coordination bonds upon desolvation^[22-24]....”

Q2

In addition, agrZW-UiO-67-MSA was obtained by refluxing agrZW-UiO-67-MSA in methanol. The current discussion, which is mainly based on PXRD, cannot support that agZW-UiO-67-MSA is a UiO-67-type framework. That is, it cannot be completely ruled out that the material gradually dissolves and recrystallizes in the refluxing methanol. In fact, it can be seen from the blurred SEM in Fig. 2b that agrZW-UiO-67-MSA are nanocrystals, quite different from that of agZW-UiO-67-MSA. The authors need to provide clear SEMs to study the morphological evolution of the particles and surfaces of agrZW-UiO-67-MSA to agZW-UiO-67-MSA.

Response: We are afraid that the reviewer confused the glass (agZW-UiO-67-MSA) and recovered crystalline phase (agrZW-UiO-67-MSA). Firstly, the structure of the agZW-UiO-67-MSA glass was characterized by EXAFS, PDF, solid NMR and FT-IR spectra and *ab initio* molecular dynamics (AIMD) simulations etc, which showed the distorted $Zr_6O_4(OH)_4$ nodes with some broken Zr-O_{COO} bonds in the *syn-syn-μ₂* COO bridges. Such a defected structure with continue random network (CRN) is reminiscent of the scenarios in structurally similar silicates and reported MOF based glasses (*Chem. Rev.* 2022, 122, 4163). Additional variable-temperature in-situ FT-IR of ZW-UiO-67·MSA also can prove the network of agZW-UiO-67·MSA resemble the UiO-67-type structure in ZW-UiO-67·MSA. Please also see the following response for Q4.

The reviewer mentioned ‘...material gradually dissolves and recrystallizes in the refluxing methanol’. In fact, agZW-UiO-67·MSA is indissolvable in methanol, which is shown in

Supplementary Fig. 5a. So, soaking the as-made a_g ZW-UiO-67·MSA in methanol with stirring at room temperature for 3 minutes led to high yield (97%) of a_{gr} ZW-UiO-67·MSA powder with partially recovered Bragg peaks, as shown in Supplementary Fig. 4a and Method Section in Page 29. The less weight loss may be due to the departure of some methylsulfonate acid MSA. To further rule out the possible recrystallization of a_{gr} ZW-UiO-67·MSA from the gradually dissolving of a_g ZW-UiO-67·MSA, the Soxhlet Extractor method was used. The results showed the a_{gr} ZW-UiO-67·MSA product with partially recovered Bragg diffractions (concentrated between 5 and 7.5°, Supplementary Fig. 15 as shown below). Combined the stimulation of methanol vapor and other kinds of solvents and the EXAFS, PDF, solid-state NMR and FT-IR spectra, it can be concluded that a_{gr} ZW-UiO-67·MSA is a UiO-67-type framework and a_{gr} ZW-UiO-67·MSA inherited the a_g ZW-UiO-67·MSA structure which is distorted and can be recovered under external stimulation.

Furthermore, the blurred SEM images in Fig. 2b have been updated with better resolution. According to the reviewer's comment, cross-sectional SEM images of a_g ZW-UiO-67·MSA, a_{gr} ZW-UiO-67·MSA obtained from short-time (3 min) soaking in methanol at room temperature and refluxing methanol (3 h) are shown in Supplementary Figs. 10d-f. In addition, the morphological evolution from a_g ZW-UiO-67·MSA to a_{gr} ZW-UiO-67·MSA can be clearly visualized in from Supplementary Figs. 10c-f.

Supplementary Fig. 15 | PXRD patterns of a_{gr} ZW-UiO-67·MSA after solvent stimulations in comparison. Products were obtained via soaking a_g ZW-UiO-67·MSA in different solvents with stirring at room temperature for 20 minutes. Compared to the patterns of the sample soaked of 3 minutes in methanol (Supplementary Fig. 4), peaks at high 2θ degree appear for those of 20 minutes in various solvents. Methanol vapor stimulation is done by exposing sample to vapor for 30 minutes. Soxhlet Extractor method with methanol as solvent is used to rule out the possibility of a_{gr} ZW-UiO-67·MSA from the gradually dissolving a_g ZW-UiO-67·MSA and recrystallization. Except for the effect of different solvents, the different crystalline phases of a_{gr} ZW-UiO-67·MSA can also be explained by the effect of different batches, including distinct composites obtained via the incipient wetness technique, uncertain

portion departure of MSA and different conditions such as with or without stirring etc.

Supplementary Fig. 10 | SEM images. a, ZW-UiO-67. b, ZW-UiO-67-MSA. c, a_g ZW-UiO-67-MSA. d, Cross-sectional image of a_g ZW-UiO-67-MSA. PTFE polymer is used as support. e, a_g ZW-UiO-67-MSA obtained via soaking in methanol with stirring at room temperature. f, a_g ZW-UiO-67-MSA resulted from refluxing methanol. g,

a_gZW-UiO-67-TFSA. h, a_gZW-UiO-67-TFA. The different false colors are shown for clarity.

Q3

The authors also found that other solvents are advantageous for the recrystallization process. However, PXRD showed that these solvents caused a_gZW-UiO-67-MSA to crystallize into a different phase than grZW-UiO-67-MSA. The authors need to try to resolve and discuss this crystalline phase.

Response: We thank the reviewer for this professional comment about the recrystallization process. It is true that other solvents can cause recrystallization as the reviewer mentioned, however, it is somewhat different from the case of methanol (Supplementary Fig. 15). After being stimulated by other solvents, the treated a_gZW-UiO-67-MSA tends to show diffraction peaks preferentially at higher angles (19°–25°); while with methanol, the treated a_gZW-UiO-67-MSA tends to have diffraction peaks at low angles (5°–12°). The presence of low-angle diffraction peaks could imply a better formation of the MOF framework. This could be attributed to the fact that methanol has the optimal molecular volume, polarity, and affinity for recovering this family of MOF glasses into their crystalline parents. Nevertheless, we also recognize that in addition to the effect of different solvents, the different re-crystalline behaviours of a_gZW-UiO-67-MSA can also be explained by the effect of different sample batches, including distinct composites obtained *via* the incipient wetness technique, uncertain portion departure of MSA during stimulation and different conditions such as with or without stirring etc.

Furthermore, in order to address the attribution of peaks emerging from other solvent stimulation, we have tried to Pawley fit the PXRD patterns of DCM stimulated sample (please see the following Fig. R1), which has the highest Bragg intensity in Supplementary Fig. 15 compared to those from other samples. The peak positions and the intensities can be reasonably fitted by the lattice parameters of ZW-UiO-67. We therefore consider the emerged phase could be ZW-UiO-67 with low crystallinity. However, the crystallinity is so low here that we cannot define the exact phase situation. But this is a very interesting phenomenon and we will explore this topic in depth in subsequent work.

For a more accurate description, we have made the following changes to the explanation in page 8 of the new version:

“...Furthermore, the glass to crystalline phase transformation upon external stimulation can be extended to other kinds of solvents, and polar solvents are advantageous for this process (Supplementary Fig. 15)...”

Supplementary Fig. 15 | PXRD patterns of $a_{gr}ZW-UiO-67-MSA$ after solvent stimulations in comparison.

Products were obtained via soaking $a_{gr}ZW-UiO-67-MSA$ in different solvents with stirring at room temperature for 20 minutes. Compared to the patterns of the sample soaked of 3 minutes in methanol (Supplementary Fig. 4), peaks at high 2θ degree appear for those of 20 minutes in various solvents. Methanol vapor stimulation is done by exposing sample to vapor for 30 minutes. Soxhlet Extractor method with methanol as solvent is used to rule out the possibility of $a_{gr}ZW-UiO-67-MSA$ from the gradually dissolving $a_{gr}ZW-UiO-67-MSA$ and recrystallization. Except for the effect of different solvents, the different crystalline phases of $a_{gr}ZW-UiO-67-MSA$ can also be explained by the effect of different batches, including distinct composites obtained via the incipient wetness technique, uncertain portion departure of MSA and different conditions such as with or without stirring etc.

Fig. R1 | Pawley fit of $a_{gr}ZW-UiO-67-MSA$ (DCM). The crystal parameters taken from ZW-UiO-67.

Q4

I am curious if the coordination structure is maintained after melting? In situ analyses by PDF, XAFS, FT-IR, etc. may provide evidence. Investigation of the structure after liquefaction helps to

understand what the real structure of agZW-UiO-67-MSA is.

Response:

We thank the reviewer for the valuable suggestion. For convenience, variable temperature in-situ FT-IR of ZW-UiO-67-MSA was conducted using a VERTEX 70 spectrometer with the ATR imaging system. All IR peaks of a_gZW-UiO-67-MSA were maintained after melting. Additional discussion has been added at the bottom of Page 14 as the following:

“...Variable temperature in-situ FT-IR data of ZW-UiO-67-MSA show the gradually broadening and shifting of C=O band from 1693 at room temperature to 1713 cm⁻¹ at 130 °C. However, the ~1693 cm⁻¹ band feature remained in the melting ZW-UiO-67-MSA and the melt-quenched glass a_gZW-UiO-67-MSA (Supplementary Fig. 27). This phenomenon further indicates the coordination network in liquids state which quenched to form the glass structure as mentioned above”

Supplementary Fig. 27 | Variable temperature in-situ FT-IR of ZW-UiO-67-MSA. The right panel of the spectra from 1300 cm⁻¹ to 1800 cm⁻¹ range, with the guiding eye lines to show the change of the carboxylate C=O band.

Reviewer #3 (Remarks to the Author):

After carefully reading the authors response and revised manuscript, most of my previous queries have been well addressed, based on some complementary experiments and further analysis of all the observed results. I also happy to see the organization, writing and quality of the paper have been improved, benefited from the comments of other reviewers. I personally think the current version is suitable for publication in Nat. Commun.

Response: We thank the reviewer for the positive comment.

Reviewer #4 (Remarks to the Author):

I am pleased that the authors have addressed the two important points made in my original review and I am satisfied with the changes that they have made in response to them. I also feel that they

have responded effectively to the comments made by the other reviewers. I am therefore happy to recommend publication of the revised manuscript in Nature Communications.

Response: We thank the reviewer for the enthusiastic comment on this work.

REVIEWERS' COMMENTS

Reviewer #2 (Remarks to the Author):

I am very pleased with the authors' excellent and detailed responses to my comments and those of the other two reviewers. I appreciate the way they have revised the paper and believe it will provide an important breakthrough for future work in glass chemistry and other fields.

RESPONSE TO REVIEWERS' COMMENTS

Reviewer #1 (Remarks to the Author):

In this work, Xue et al. presents a very comprehensive report of a generalized method for making carboxylate MOF glasses. This method is near universal and can be extended to other MOF systems. The reported strategy is of great scientific significance and could open up a brand-new pathway for making MOFs and other molecular crystals into glasses. In this regard, I highly recommend this manuscript for publication in Nature Communications after a major revision. The authors could consider the following points to strengthen their manuscript:

Response: We sincerely thank the reviewer for the enthusiastic comments on the novelty, significance and universality of our work.

1. In Figure 1a, the 'Zr-O node' should be changed to 'Zr₆O₄(OH)₄ node' to avoid misunderstanding. Similar changes need to be made in the supporting information.

Response: We have replaced 'Zr-O node' with 'Zr₆O₄(OH)₄ node' in both Fig. 1a and Supplementary Fig. 1.

2. In the SEM images of Figure 2b, the scale bars are not uniform. For better comparison, I suggest the authors make the scale bar in all images with the same length.

Response: We have reorganized Figure 2b to make all the scale bars uniform.

3. In my opinion, the proposed strategy could be extended to other molecular frameworks. Covalent-organic frameworks are probably the immediate targets, can the authors give some comments whether this could be possible?

Response: We fully agree with the reviewer that our strategy can be extended to other molecular systems. Our recent results have demonstrated that covalent-organic frameworks can indeed be made into glasses using this strategy. Corresponding discussions and prospect of this strategy have been added to the last paragraph of the revised manuscript as the following:

"Importantly, our strategy could be applied to a range of highly porous MOFs composed of different ligands and metal nodes. In addition, this strategy could also be extended to other kinds of highly porous molecular frameworks such as covalent-organic frameworks."

4. The application prospects of the reported new carboxylate MOF glasses were only briefly mentioned, the discussion could be slightly extended.

Response: We have slightly extended our discussion about the application prospect in the "Discussion" section as the following: "...Our findings would also offer new opportunities for processing MOFs and other hybrid organic-inorganic crystals by melt-quenching them into glasses, which could enable them to meet the manufacturing requirement by industry..."

5. Considering the complexity, I do not believe it is possible to do the reverse Monte Carlo simulations to obtain the structures of the melt-quenched glasses. Nevertheless, I am curious whether the authors have had a go with it.

Response: We thank the reviewer for this insightful comment. We have indeed spent a year and a half in trying the RMC simulations on the ZW-UiO-67·HA glasses with many different approaches. However, we were unable to achieve this goal due to lacking of reasonable starting structural models,

even with enormous endeavor by the inventor of the RMC simulation software package (Prof. Martin T. Dove, who is also a co-author of this revision). Specifically, the RMC models of ZIF glasses are based on the SiO₂ glass by replacing Si atom with the imidazolate ligand, while the SiO₂ glass model is derived from the amorphous Si by inserting O atoms between adjacent Si atoms. Since the majority of MOFs are not four-connected (including ours), the RMC simulations have only been successfully done for a few ZIF type MOFs (please see Scheme. R1 below) up to date.

Scheme R1. Explanation of the unsuccessful RMC simulations for non 4-connected MOFs.

6. I believe the authors' results are robust enough to support their conclusion of the melt-quenched formation of glasses, however, have they obtained any evidence of excluding the formation of monoliths? This is very important for convincing the community.

Response: We thank the reviewer for raising this valuable comment. To exclude the formation of monoliths, comprehensive small angle X-ray scattering (SAXS), wide angle X-ray scattering (WAXS) and electron diffraction experiments have been performed on ZW-UiO-67 and a_gZW-UiO-67·HA. The results have been compared with those of polycarbonate and silicate glasses, which clearly show the melt-quenched a_gZW-UiO-67·HA glasses are not the monoliths of nanoparticles blended within organic acids (Supplementary Figs. 27-28). For a_gZW-DUT-5·MSA and a_gZW-UiO-68·MSA, the optical microscopic images, SEM images, TEM images and electron diffraction patterns from the glasses also have been carefully recollected and shown in Supplementary Figs. 29, 32 and 34. As expected, no electron diffraction of a monolith with nanocrystals or nanoparticles was observed, explicitly implying the homogenous amorphous states of these melt-quenched carboxylate a_gMOFs. Corresponding discussion has been added to page 17 of the revised manuscript as the following:

“The SAXS signal $I(q)$ of a_gZW-UiO-67·HA crucially continues, rather than that of a monolith blended with nanoparticles^[45]. In addition, no detectable wide angle X-ray scattering (WAXS) peaks and the missing electron diffraction in TEM images were recorded for a_gZW-UiO-67·HA, exhibiting comparable amorphous features to those of polycarbonate and silicate glasses (Supplementary Fig. 27-28).”

7. Ref 42 entitled “Ionic liquid facilitated melting of the metal-organic framework ZIF-8” (Nat. Commun. 2021, 12:5703) is a very important reference, I suggest the authors move it to the very beginning of the manuscript.

Response: We have moved this important reference to the beginning of the revised manuscript. In addition, related discussion has been added to the introduction section on page 4.

“Meanwhile, the un-meltable counterparts ILs@UiO-67 were also synthesized as references to

show that such an universal strategy is in stark contrast to the reported guest-host interactions scenario for the melt of ILs@ZIF-8^[27].”

8. The influence of incorporated acids on the melting process needs to be discussed. With reference to the simulation work in Ref 3 and Ref 12, the thermodynamic barriers of metal-node dissociation at high temperatures can be briefly discussed, to intuitively reflect the lowering of melting barriers by the acid.

Response: We thank the reviewer for this important suggestion. The long-range order of ZW-UiO-67·HA was disrupted to lead to increased configurational entropy and reduced energy barrier beneficial for melting due to the following synergistic reasons: the charge delocalization and separation effect of the acid-zwitterion buffer system (MIMS·MSA), the weakened coordination bonds between COO^- and metal nodes, and the dense structures with the fully filled channels that facilitate the break and mismatch of the coordination bonds. Additional discussions have been added on page 9 in the revised manuscript as follows:

“...theoretical calculations reveal the coordination ability of the carboxylate ligands is weakened as BPDC-MIMS paired with more nearest MSA molecules. The negative charges of four O_{COO} atoms change from -0.64 , -0.64 , -0.56 and $-0.59|e|$ to -0.62 , -0.53 , -0.52 and $-0.61|e|$ due to the charge delocalization and intermolecular interactions. It could be speculated the weakened Zr-O bonds may trigger the bond dissociation and atom displacements under the assistance of the acid-zwitterion buffer. However, the densely filled channels in ZW-UiO-67·MSA not only are beneficial for the coordination bonding mismatch between COO^- and metal nodes upon heating, but also enhance the structural disorder and increase the configurational entropy to markedly reduce the energy barrier required for the melting...”

Reviewer #2 (Remarks to the Author):

In this work, the authors developed a Brownsted acid-base strategy for melt-quenched glass formation of metallic carboxylate frameworks by grafting amphiphilic ions onto carboxylate ligands and incorporating organic acids into framework channels. Upon heating, the ordered electrostatic interactions between the attached amphiphilic ions and the incorporated organic acids are stochastically shifted due to charge delocalisation. The melting and glass-forming behaviour is mainly supported by TGA, DSC, PXRD, SEM and other characterization results. This work provides a route to the possibility of melting carboxylate MOFs. Through experiments from different aspects, the process of melting MOF glass is explored, which is very beneficial for understanding the process of MOF glass. Actually, carboxylate MOF glasses have been recently reported by different strategies including the melt-quenching method (10.21203/rs.3.rs-2922761/v1) and desolvation approaches (Angew. Chem. Int. Ed. 2023, e202305942; 10.26434/chemrxiv-2023-05kv3). The authors should discuss the related reports on carboxylate MOF glasses, especially the firstly reported cluster-based carboxylate MOF glass. Based on these considerations, the current manuscript is still too preliminary and major revision is necessary to support its publication.

Response: We thank the reviewer for the positive and critical comments on our work. We are sorry for not citing the most recently developed metal-carboxylate MOF glasses. We have cited these important work as references 22, 23 and 24. Additional discussions have been included in page 3 of the revised manuscript as shown in the following:

“...porous metal-carboxylate MOFs having high CN more than 4 (6 to 12) are believed to be un-meltable...Several melt-quenched metal complexes with ligands bearing carboxylate and a secondary coordination group (pyridyl/azolate) can be obtained from discrete solvated complexes through rearrangements of metal-ligand bonds upon desolvation^[22-23]. Taking advantage of the flexibility and low symmetry of the aliphatic carboxylate ligands and the lack of crystal field stabilization energy on metal ions, a selection of glasses comprising low oxidation state metals (Mg^{2+} , Mn^{2+}) and flexible adipate were reported very recently^[24]. However, this strategy is not applicable to those carboxylate MOFs bearing with high valence metal ions.”

Furthermore, careful revisions have been done and detailed in our responses set below.

1. For ZW-UiO-67·MSA in Fig. 2a, "The second endothermic peak at 185.4 oC can be ascribed to the de-associated acetate", I think this should be reconsidered because the coordinated CH_3COO^- should be more stable than MSA, and further characterizations are needed, for example TG-MS.

Response: We thank the reviewer for this valuable suggestion. In the original manuscript, the second endothermal peak at 185.4 °C of the DSC plot was temporarily attributed to the decomposition of the acetate group because similar thermal behavior was reported in $Zr(CH_3COO)_4$ upon heating to ~200 °C (*J. Clust. Sci.* 2016, 27, 1553). Also, such a sharp peak is closely related to that event found for neat ZW-UiO-67 at 180.0 °C and ZW-UiO-67·0.5MSA at 186.2 °C, respectively (Supplementary Fig. 5). Considering the formula mass percentage of the acetate ($-OOCCH_3$ wt% = 1.4 wt% in ZW-UiO-67·MSA) is close to ~1.8wt% weight loss recorded before 250 °C, we speculated such endothermal event without weight loss may be related to the de-association of the acetate which is followed by less composite decomposition after that temperature. Online gas-phase FT-IR spectra detected H_2O and CO_2 with the temperature increase

(Supplementary Fig. 25), but no MSA was recorded even up to 320 °C (total weight loss ~4.5 wt%). For more clarity, careful analysis combining DSC, TGA and FT-IR were done, and the corresponding results were reorganized and discussed in the last paragraph highlighted on page 7 as:

“...At 185.4 °C, the endothermal sharp peak closely relates to that event for ZW-UiO-67 at 180.0 °C and ZW-UiO-67-0.5MSA at 186.2 °C, respectively (Supplementary Fig. 5). Considering the formula mass percentage of acetate (-OOCCH_3 wt% = 1.4% in ZW-UiO-67-MSA) is close to the ~1.8wt% weight loss recorded before 250 °C, it could be speculated that the endothermal event without weight loss at 185.4 °C may relate to the de-association of the acetate group which is followed by less composite decomposition after that temperature. Online gas-phase FT-IR spectra demonstrate that H_2O and CO_2 can be detected with the temperature increase (25 °C to 320 °C) (Methods and Supplementary Fig. 25).”

More detailed thermal events are also provided in Methods (page 22-23) for further review.

Supplementary Fig. 5 | DSC curves of the $\text{H}_2\text{BPDC-MIMS}$ linker and ZW-UiO-67 with and without incorporation of MSA. a, Up-scan curve of ZW-UiO-67-MSA from -150 to 320 °C, with thermogravimetric analysis (TGA) trace shown as yellow dashed line, being consistent with Fig. 1a. Inset photography is the soaking of melt-quenched solids (from 130 °C) in methanol for 3 minutes at room temperature, obtaining precipitates with weakened Bragg scattering as shown in Supplementary Fig. 4a. **b**, Special cyclic DSC curve of ZW-UiO-67-MSA from 0 to 150 °C shown with T_m , T_g phase transition temperatures and fusion enthalpy. **c-d**, Compared curves of ZW-UiO-67-0.5MSA and ZW-UiO-67 ($0 \sim 320$ °C) (up-scan) along with TGA traces of yellow dashed lines

(25 ~ 800 °C). The plot of ligand H₂BPDC-MIMS (blue line) is presented as a reference, with asterisk inferring the phase transition and the dashed-lines guiding that similar events occurred for ZW-UiO-67-0.5MSA and ZW-UiO-67 at this temperature. Please see the detailed discussion and analysis of DSC in the Method section.

Supplementary Fig. 25 | Online gas-phase FT-IR spectra with temperature increasing. In-situ gas-phase FT-IR of ZW-UiO-67·MSA measured from 25 to 320 °C, with the dominant stretching vibration bands of H₂O (1250 -2000 cm⁻¹, 3400-4000 cm⁻¹) and CO₂ (~2200 cm⁻¹).

2. The low-density liquid (LDL) to high-density liquid (HDL) transition needs to be proven.

Response: We thank the reviewer for this professional comment. This event relates to two exothermal events with onset temperature at 162.5 and 256.7 °C in the DSC plot of ZW-UiO-67·MSA (Fig. 2a, Supplementary Fig. 5). Due to the lack and difficulty of related characterizations for the phase transitions from the low-density liquid (LDL) to high-density liquid (HDL), we have reconsidered and carefully analyzed the TGA, DSC and FT-IR data, and rewritten the discussion to give a more appropriate explanation in the revised manuscript. For the first exothermal event (162.5 °C) without weight loss, it may be ascribed to the polyamorphic phase transitions between distinct structural configurations of neighbor states with different potential energies, as stated in the last paragraph of page 7:

“...The following exothermal event with onset at 162.5 °C (without weight loss) may be ascribed to polyamorphic phase transitions between distinct structural configurations of neighbor states with different potential energies (Fig. 2a, Supplementary Fig. 5)...”

For the second exothermal event with the onset temperature at 256.7 °C, it may relate to some thermal decomposition taken place before 282.9 °C (total 3.2_{wt}%) because similar thermal events are also observed for ZW-UiO-67-0.5MSA (onset: ~250 °C) and neat ZW-UiO-67 (onset: ~266 °C). Corresponding discussion has been added and highlighted in Method on page 23 as:

“...For the last exothermal event with onset temperature 256.7 °C of ZW-UiO-67·MSA (Fig. 2a), it may relate to some thermal decomposition taken place before 282.9 °C (total 3.2_{wt}%). Similar thermal events are also observed for ZW-UiO-67-0.5MSA (onset: ~250 °C) and ZW-UiO-67 (onset: ~266 °C).”

~266 °C)...”

3. “However, the endothermal evidence (218.3 °C) is reminiscent of the phase transition for ZW-UiO-67 (~212.6 °C) (Supplementary Fig. 5)”, what is the new phase?

Response: The third endothermal peak for ZW-UiO-67·MSA and ZW-UiO-67·TFSA is at 218.3 and 210.6 °C, respectively (Fig. 2a, Supplementary Fig. 5). In addition, it closes to the feature at 212.6 °C found in ZW-UiO-67, ZW-UiO-67·0.5MSA and H₂BPDC-MIMS (Supplementary Figs. 5). Therefore, we temporarily ascribe it to a plastic phase transition (*Phys. Chem. Chem. Phys.* 2010, 12, 11291–11298) reminiscent of that found for solid ionic liquids (*Korean J. Chem. Eng.* 2006, 23, 940–947) due to the flexible molecular symmetries of the zwitterion-acid subsystem and complex intermolecular interactions (i.e. H-bonding, Coulomb force etc.) within ZW-UiO-67·MSA. Related discussion is shown and highlighted on page 23 as:

“...The third endothermal peak at 218.3 °C of ZW-UiO-67·MSA and 210.6 °C of ZW-UiO-67·TFSA also are similar and close to that at 212.6 °C found in ZW-UiO-67 and ZW-UiO-67·0.5MSA and H₂BPDC-MIMS (Supplementary Fig. 5), which may be ascribed to a plastic phase transition reminiscent of that found for solid ionic liquids^[49]...”

4. Is there a mistake in "(1-mthyl-3-imidazolio) propane-3-sulfonate)" in line 85?

Response: We have corrected this mistake with '(1-methyl-3-imidazolio)propane-3-sulfonate' in the revised manuscript.

5. The authors emphasize that the Zr-O-Zr connection in agZW-UiO-67·MSA and agZW-UiO-67·TFSA is still reserved, but the Zr-O-Zr region in Extended Data Fig.19 is negligible. They need more clear evidences to support their points.

Response: We sincerely thanks for your thoughtful consideration of the Wavelet transform EXAFS data. As you said, it is true that the EXAFS data of agZW-UiO-67·MSA and agZW-UiO-67·TFSA have weak peaks of Zr-O-Zr after the wavelet transformations in Extended Data Fig.19, but we don't think that they can be considered as a negligible. Firstly, Zr-O-Zr belongs to second coordination shell, the scattering path of which is instinctively weaker than that of the first coordination shell of Zr-O, as that shown radial distance $\chi(R)$ space spectra for crystalline UiO-67 and ZW-UiO-67 in Fig.3c,d. While, this phenomenon is pronounced in agZW-UiO-67·MSA and agZW-UiO-67·TFSA due to the distorted octahedron Zr₆O₄(OH)₄ along with some broken Zr-O bonds and decreased coordination number (2.95, 3.46) (Extended Data Table 3) upon vitrification. For more clarity, the strengths of different coordination shell path contribution of Zr-O-Zr, Zr-O and Zr-O-C in $\chi(R)$ space spectra of UiO-67, agZW-UiO-67·MSA and agZW-UiO-67·TFSA are compared and shown in Supplementary Fig 19a of new version. So, the wavelet transform EXAFS spectra (Extended Data Fig.19 you mentioned) shows weak region of the Zr-O-Zr. Secondly, Quantitative $\chi(R)$ space 2D profiles comparison in Fig. 3e,f also explicitly shows that weaken but not negligible Zr-O-Zr path contribution. Related discussion in page 11 as:

“...Such cluster changes are also intuitively witnessed by the Quantitative $\chi(R)$ space spectra fitting, wavelet transform extended X-ray absorption fine structure (WTEXAFS) and reduced coordination number (CN) of Zr-O bond (7.01-7.72) and Zr...O...Zr scattering path (2.95-3.18) in agZW-UiO-67·MSA and agZW-UiO-67·MSA as compared the perfect UiO-67 (8, 4) (Fig. 3e-f, Supplementary

Figs.20-21, Table 3)...”.

Supplementary Fig. 19 | XANES and $\chi(R)$ space spectra in comparison. **a**, The strength of different coordination shell path contribution of Zr-O-Zr and Zr-O in $\chi(R)$ space spectra in comparison. **b**, XANES $\mu(E)$ spectra observed at the K edge of the Zr atoms in UiO-67, ZrO_2 , **ZW-UiO-67** and **ZW-UiO-67·0.5MSA** at 298 K.

6. agZW-UiO-67·MSA was immersed in methanol and refluxed to obtain agrZW-UiO-67·MSA. Is it because agZW-UiO-67·MSA was dissolved in methanol and then re-formed nanocrystals? Are other solvents used (polar solvents, non-polar solvents, H₂O, etc.) to rule out the possibility of ligand dissolution leading to recrystallization?

Response: We thank the reviewer for this professional comment. Firstly, agZW-UiO-67·MSA is indissolvable in methanol, and the special short-time solvent stimulation (3 minutes) at room temperature can facilitate the recovery of the Bragg diffraction of the melt-quenched agZW-UiO-67·MSA, as shown in Supplementary Figs. 4 and 5. A UiO-67 framework is unable to be reconstructed at such a moderate condition in a short time. Therefore, such a recovered structure is derived from a preserved framework of the parent MOF, rather than a reaction or recrystallization of a dissolved/decomposed structure. In this context, the methanol may remove part of the organic acid (60% MSA remained, Method) embedded in the distorted framework of the formed glass and stimulate the phase transition to a more ordered structure. To further rule out the possibility of ligand dissolution leading to recrystallization, methanol vapor was used, and the low angle X-ray diffraction peaks recovered after 30 min exposure (Supplementary Fig. 15 as shown below). Furthermore, other different kinds of polar and non-polar solvents are also used to investigate the recovery, and polar solvents would be beneficial for this process. Related detail discussion has been included and updated in middle of page 8 in the revised manuscript:

“...Elemental analysis demonstrates ~60% MSA was preserved within agrZW-UiO-67·MSA, with the rest released during reflux. Such embedded MSA and the relative low crystallinity resulting from the solvent stimulation accounts for the reduced porosity. Furthermore, the glass to crystalline phase transformation upon external stimulation can be extended to other kinds of solvents, and polar solvents are advantageous for this process (Supplementary Fig. 15). By exposing to methanol vapor for 30 min, the X-ray diffraction peaks at low angles appeared with a broadened feature...”.

Supplementary Fig. 15 PXRD patterns of $a_{gr}ZW-UiO-67 \cdot MSA$ after solvent stimulations in comparison. Products were obtained via soaking $a_{gr}ZW-UiO-67 \cdot MSA$ in different solvents with stirring at room temperature for 20 minutes. Compared to the patterns of the sample soaked of 3 minutes in methanol (Supplementary Fig. 4), peaks at high 2θ degree appear for those of 20 minutes in various solvents. Methanol vapor stimulation is done by exposing sample to vapor for 30 minutes.

7. From the relative low porosity of $a_{gr}ZW-UiO-67 \cdot MSA$ than $ZW-UiO-67 \cdot MSA$, authors guess the 60% of the MSA still remains in the framework. What is the degree of crystallinity for the $a_{gr}ZW-UiO-67 \cdot MSA$? This is another possible reason for the lower porosity.

Response: We thank the reviewer for this constructive comment. We fully agree with the reviewer that the low porosity could be caused by the low crystallinity in this case. As mentioned previously, 60% MSA remained within $a_{gr}ZW-UiO-67 \cdot MSA$, which mainly contributes to the reduced porosity. However, low crystallinity is also an important reason for the low porosity in $a_{gr}ZW-UiO-67 \cdot MSA$, and we have included additional discussion in page 8 of new version as:

“...Such embedded MSA and the relative low crystallinity resulting from the solvent stimulation accounts for the reduced porosity...”

8. $ZW-UiO-67 \cdot 0.5MSA$ and $ZW-UiO-67 \cdot MSA$ have the different thermodynamic behaviors. $ZW-UiO-67 \cdot 0.5MSA$ cannot melt. The authors should provide further experiments to discuss the role of MSA in the melting behaviors.

Response: We thank the reviewer for this insightful comment. For better clarity, we have rewritten and highlighted the melting mechanism section on page 9.

Firstly, from the point of view of acidic ionic liquids, $ZW-UiO-67$ can be regarded as a zwitterion-decorated MOF with the MIMS groups. The incorporated MSA acid pairs with the MIMS group to form a Brønsted acid-base buffer subsystem, $MIMS \cdot MSA$, as evidenced by the DSC data of them and acidic ionic liquids (Ref. 25, *Chem. Rev.* 2016, 116, 6133–6183). The more MSA molecules pair with the zwitterionic ligands through electrostatic and/or hydrogen-bonding interactions, the greater charge delocalization and separation occur. This substantially disrupts the long-range structural order and consequently leads to increased configurational entropy (S_{conf}), which

dramatically reduces the T_m (Ref. 26, *J. Non-Cryst. Solids* 2017, 463, 175–188) and facilitates the solid-to-liquid phase transition, resulting in the melting of ZW-UiO-67·MSA at $\sim 127^\circ\text{C}$ but unmeltable ZW-UiO-67·0.5MSA due to its insufficient acid. This conclusion has been further demonstrated by the formation of zwitterion-based acidic ionic liquids: the solid zwitterionic EIMS and H₂BPDC-MIMS respectively give rise to room temperature acidic ionic liquids EIMS·MSA and H₂BPDC-MIMS·MSA as the amount of MSA is 3.78 times that of zwitterions (Supplementary Fig. 1 and Table 2); while a solid product, i.e. a relative high melting temperature, is observed since half amount of MSA (1.89 times) is insufficient to lead to the melting as observed in our previous work (Ref. 29, *Angew. Chem. Int. Ed.* 2021, 60, 1290–1297). Meanwhile, the densely filled channels in ZW-UiO-67·MSA not only are beneficial for the coordination bonding mismatch between COO⁻ and metal nodes upon heating, but also enhance the disordered structure and increase the configurational entropy to markedly reduce the energy barrier facilitating the melting.

Secondarily, to further reason the role of MSA in ZW-UiO-67·0.5MSA, more sensitive ¹H-¹³C cross polarization magic-angle-spinning (CP MAS) spectra have been complemented. The missed COO resonance (~ 173 ppm) of CP MAS in ZW-UiO-67 re-appears in the unmeltable ZW-UiO-67·0.5MSA (Fig. 4e, Supplementary Fig. 26b). Such a resonance signal indicates the protonation of the COO groups in the half-filled ZW-UiO-67·0.5MSA, but the insufficient acid cannot realize the melting as that for ZW-UiO-67·MSA. The updated CP MAS spectra is shown in Supplementary Fig. 26b, and related discussion is presented in middle of Page 14 revised as the follows:

“...around 173.9 ppm and aromatic carbon resonances from 120 to 150 ppm (Fig. 4e). Meanwhile, more sensitive ¹H-¹³C cross polarization magic-angle-spinning (CP MAS) spectra of a_gZW-UiO-67·MSA show comparable features to those of unmeltable ZW-UiO-67·0.5MSA, but a slightly different resonance at ~ 182 ppm (Supplementary Fig. 26b) which implies much small amount of uncoordinated COO groups^[35,39] due to that solid-liquid phase transition...”

Supplementary Fig. 26 | Solid-state ¹³C NMR spectra at room temperature. a, Solid-state ¹³C NMR spectra of CP MAS (at 12 kHz) of ZW-UiO-67, and DD MAS (at 12 kHz) of ZW-UiO-67, ZW-UiO-67·TFSA and a_gZW-UiO-67·TFSA. Enclosed area indicates the missed CP MAS resonance of C_{COO} in ZW-UiO-67 and the appeared DD MAS signal in comparison. Spinning sideband are marked with asterisks. Signals signed with triangle are ¹³C resonances of DMF (*N,N*-dimethylformamide) solvent. 121.2 ppm is the resonance signal of -CF₃ in TFSA (*N,N*-

bis(trifluoromethanesulfonyl) amide). **b**, Appeared C_{COO} resonance in CP MAS of ^{13}C NMR spectra of $a_gZW-UiO-67 \cdot MSA$ and $a_gZW-UiO-67 \cdot TFSA$ and $ZW-UiO-67 \cdot 0.5MSA$ due to protonated C_{COO} . Enclosed rectangle indicates the appeared C_{COO} resonance, as compared to that missed signal of $ZW-UiO-67$ (See also Fig. 4e).

9. As a control experiment, UiO-67 is mixed directly with EIMS-MSA to check the melting behaviors. VT-PXRD shows that some peaks of the mixture disappear during heating in Extended Data Fig.7. Is this possibly caused by uneven mixing and not complete melting? Is there any evidences that EIMS-MSA is evenly distributed in UiO-67?

Response: We agree with the reviewer and have reconsidered this critical point. We are inclined to believe this is due to a very tiny amount of residual inhomogeneous mixing during the preparation (the acids diffuse into the MOF channels) at 60 °C (Method). So the Bragg peaks between 5-10 degrees continue to become weaker with increasing temperature which can be evidenced in Supplementary Fig. 8. For clarity, a caption for Supplementary Fig. 8 “The weakening of the low-angle peaks in panel **d** may be due to the complete homogenization of ILs@UiO-67 resulted from high temperature heating” is presented.

Moreover, we do not think this is due to ‘incomplete melting’ due to the following two reasons. Firstly, other high-angle Bragg peaks show almost no changes in intensity; secondly, the melting behavior of ILs@UiO-67 in the thermal analysis was unable to be detected. Nevertheless, we must clarify that EIMS-MSA was indeed almost completely diffused into UiO-67, as evidenced by the BET and pore size distributions before and after the impregnation of MSA, i.e. ignorable surface area was detected, as shown in Supplementary Fig. 14 and Supplementary Table 1. Furthermore, more kinds of HA/ILs@UiO-67 have been investigated, which all exhibited as unmeltable solids similar to that of EIMS-MSA@UiO-67, as shown in Supplementary Table 5:

Supplementary Table 5 | Compared melting behaviors of HA/ILs@ZW-UiO-67 to that of HA/ILs@UiO-67 (see also Supplementary Figs. 5-6, 29, Table 1, 2, 4).

	Meltability of ILs@ZW-UiO-67 and ILs@UiO-67 (heated to 320 °C)		Meltability/ T_m (°C) of ZW-UiO-67·HA and HA@UiO-67			
MOF ILs/HA	EMIM·TFS*	EIMS·MSA *	MSA	TFSA	TFA	ESA
ZW-UiO-67 ^a	No	/	Yes /127.2	Yes /149.6	Yes /157	Yes /139
UiO-67 ^b	No	No	No/-	No/-	No/-	No/-
ρ (g/ml) ^c	1.387	1.42	1.48	1.892	1.696	1.388

pK_a	/	/	-1.9	-12.2	-14	1.8
--------	---	---	------	-------	-----	-----

* EMIM·TFS = 1-ethyl-3-methylimidazolium trifluoromethanesulfonate, EIMS = 1-(1-ethyl-3-imidazolium)propane-3-sulfonate. ^a For synthesis of **EMIM·TFS@ZW-UiO-67**, de-solvated **ZW-UiO-67** (80 mg) and 34 μ L ILs (**EMIM·TFS**) (equal to the pore volume of MOF) were used. ^b For **ILs@UiO-67** and **HA@UiO-67**, activated UiO-67 (80 mg) and 81 μ L (equal to the pore volume) HA or ILs were respectively used. All samples were obtained by the same incipient wetness technique (see also Method).

10. In Extended Data Fig. 24, the NMR spectrum is not clearly discussed. The peak at 182.3 ppm for $agZW-UiO-67 \cdot MSA$ is difficult to observe.

Response: We thank the reviewer for pointing out this question. This weak peak at 182.3 pm may be due to the very small amount of uncoordinated COO groups which only gives to limited NMR signal. For clarity, $^1H-^{13}C$ cross polarization magic-angle-spinning (CP MAS) spectra of $agZW-UiO-67 \cdot MSA$ was re-collected with a long-time scan, as shown in the updated Supplementary Fig. 24b. For further comparison, CP MAS spectra of $ZW-UiO-67 \cdot 0.5MSA$ was additionally measured and compared, which clearly showed the absence of a 182.3 ppm peak (Supplementary Fig. 26b).

Reviewer #3 (Remarks to the Author):

In this manuscript, the authors reported the realization of melting and the preparation of quenched glass of three typical metal-carboxylate MOF of UiO-67, UiO-68 and DUT-5. The research follows one of the important trends in MOF chemistry, that is transforming traditional crystalline MOFs into their liquid or glassy states. Relative efforts have gained success firstly in a limited numbers of ZIFs and some 1D/2D coordination polymers, and recently in traditional MOFs with both single metal and cluster as nodes. Even so, the area is quite new with a lot of puzzles remain to be explored. From this point, the presented result is interesting and significant to attract wide readership.

For the investigated content, all the three compounds have long been known as quite thermal stable MOF constructed from [Zr6] node, yet decomposition rather than melting occur upon heating. Such fact indeed shows the current situation that most MOFs, especially the vast number of metal-carboxylate MOFs, can not be melted. This work developed a Brønsted acid-base strategy by grafting the zwitterions on the carboxylate ligands and incorporating organic acids in the framework channels to enable the melt-quenched glass formation of UiO-67, UiO-68 and DUT-5. The method somewhat draw lessons from traditional ionic liquid with long research history, and similar strategy has been successful in ZIF-8 (Ref. 42) which is also known without melting. Even so, present research results are no doubt important and valuable to enhance understanding to MOF glass preparation. Especially, the research is performed in a comprehensive and thorough way with abundant characterization and experimental evidences, which is infrequent and laudable in peer study. Hence, I think this paper is a suitable candidate for publication in Nat. Commun. Some queries needed to be answered are listed as following:

Response: We thank the reviewer for these very positive comments, which will significantly inspire our future work.

1. As a great aspiration, the authors intend to construct a “generally applicable strategy” to vitrification of wide range of MOFs which are “composed of different ligands and metal nodes” and “not limited to the structure type and porosity”. However, presented three MOFs are in fact quite similar either in composition and structure. Particularly, they possess very rigid framework to endure zwitterions grafting and organic acids inclusion. While most known metal-carboxylate framework did not have such high structural stability. The method applicability is open to doubt, or at least need much more examples. On this situation, I think the presented title is too broad and the conclusion is exaggerated.

Response: We agree that the three MOFs presented in our study are indeed quite similar in terms of both composition and structure, possessing a high level of structural stability required for zwitterion grafting and organic acid inclusion. We acknowledge that this may raise doubts regarding the applicability of our method to a broader range of MOFs. As suggested by the reviewer, the title has been changed to ‘Melt-Quenched Glass Formation of a Family of Metal-Carboxylate Frameworks’. The conclusion in page 16 has also been revised accordingly as the following:

“...a family of metal-carboxylate frameworks (UiO-67, UiO-68 and DUT-5) were successfully made into glasses *via* the melt-quenched method. Importantly, our strategy could be applied to a range of highly porous MOFs composed of different ligands and metal nodes. In addition, this strategy could also be extended to other kinds of highly porous molecular frameworks such as covalent-organic frameworks. Our findings would also offer new opportunities for processing MOFs and other hybrid organic-inorganic crystals by melt-quenching them into glasses, which

could enable them to meet the manufacturing requirement by industry.”

2. All the DSC results are quite strange comparing to known MOF glasses and also other traditional glass system. These abnormal data further lead to great queries in several important thermal parameters of T_m and T_g for all reported compounds. I noticed two instruments of DSC-Q200 and METTLER TOLEDO DSC3 were applied to collected data. How about the data repeatability, as well as the influence factor of flowing gas, heating rate?

Response: We thank the reviewer for this constructive comment. Firstly, for the DSC data and repeatability, the top DSC plot of Fig. 2a was recorded on an instrument (DSC-Q200) able to decrease the temperature to $-150\text{ }^\circ\text{C}$ with the aim to detect the possible phase transition at the low temperature region; however, the DSC curve in the bottom panel was measured via a different instrument (METTLER TOLEDO DSC3) for recording the cyclic plot only from $0\text{ }^\circ\text{C}$. According to the reviewer’s comment, different flowing gas and/or heating rate for the DSC measurements of ZW-UiO-67·MSA, ZW-UiO-67·TFSA and ZW-UiO-67·TFA have been done. For ZW-UiO-67·MSA, different samples were used to repeat the DSC results with the aim to view the possible effect of different batches. The T_m was slightly influenced by the heating and gas flowing rate, with the value varying in a narrow range ($127\text{ }^\circ\text{C}$ - $132\text{ }^\circ\text{C}$), but distinct heating rates led to T_g lying within a relative larger region ($73\text{ }^\circ\text{C}$ - $104\text{ }^\circ\text{C}$). For ZW-UiO-67·TFSA and ZW-UiO-67·TFA, the carefully re-recorded plots of up scans with variable heating rates. Those newly repeated DSC data were all collected from instrument METTLER TOLEDO DSC3, showing a good repeatability (see Supplementary Fig. 7). Related discussion has been updated in the bottom of first paragraph of page 8 as:

“...Different flowing gas and/or heating rates exhibited slight influence on T_m , with the value varying in a narrow range ($127\text{ }^\circ\text{C}$ - $132\text{ }^\circ\text{C}$), but distinct heating rates led to T_g lying within a relative larger region ($73\text{ }^\circ\text{C}$ - $104\text{ }^\circ\text{C}$) for ZW-UiO-67·MSA (Supplementary Fig 7)...”

Secondly, for the strange feature of DSC, here we show a glass family derived from a MOF incorporated with the Brønsted acid, i.e. a solid ionic liquid with T_m over room temperature, rather than a porous crystalline MOF such as ZIF-62 etc. Here, ZW-UiO-67·HA bears the rigid coordination framework and acid-zwitterion buffer subsystem, MIMS·HA, with complex intermolecular interactions and more flexible molecular configurations. Therefore, such a soft-in-hard structure displays a more complex thermal behavior markedly different from that of porous crystalline ZIFs. This leads to a broad distribution of molecular motions and relaxation processes during the heating process, showing a broad and weak melting endothermal peak and T_g feature. That may be reminiscent of that of porous organic cages with flexible structures upon heating (*J. Mater. Chem A*, 2021, 9, 19807-19816; please also see page 7 revised). Furthermore, the efficiently grinding, completely and homogeneously diffusing of liquid acids into the channels of MOFs may also influence the final complete structural homogeneity of ZW-UiO-67·HA (vs. crystal) which subsequently affects the DSC features because the sample is obtained via an incipient wetness technique (Method). To obtain the repeatable DSC results in the revised manuscript, all those re-recorded samples were obtained under nitrogen atmosphere protection. The moisture from air may also effect the components of ionic ZW-UiO-67·HA. Special mention was presented at their synthesis of Method at page 27 of new version.

Supplementary Fig. 7 | Different flowing gas and/or heating rate for DSC of ZW-UiO-67·MSA and ZW-UiO-67·TFSA. **a**, Repeated DSC measurements by use different samples to show the effect of heating rate and different batch on the T_m , and **b**, the effect on T_g of ZW-UiO-67·MSA. Asterisk infer the gas flowing rate of 80 ml/min that is different from those of all others (50 mL/min). **c**, Repeated cyclic DSC plot from 0 °C to 170 °C and the following down to 0 °C, and then up scan to 170 °C (black line) with heating rate of 10 °C/min for ZW-UiO-67·TFSA. The red and turquoise plots are for the second and third cyclic data with different heating rates, while the blue line is the last up scan of the a_g ZW-UiO-67·TFSA. **d**, comparison of four up scan plots for ZW-UiO-67·TFA as that in panel **c**.

3. For the typical DSC of ZW-UiO-67·MSA in figure 2a, firstly the really melting-range is indeed confusing for readers. Why the T_m point is in a platform position? More importantly, the authors listed three temperature points of 162.5, 185.4 and 218.3, which seems located at TG range without weight loss. As well known, the melt liquid goes in a thermal stable energy state as homogeneous phase with steady DSC trace. What happens at these points with evident endothermic and exothermic signals? The two-paragraph discussion on thermal behaviors in Page 6 are convoluted.

Response: We thank the reviewer for these critical comments. For the first comment, in the bottom panel of Fig. 2a, the platform position of the offset T_m point is due to the system error between two different instruments, with the 127.2 °C for the top and 125 °C for the bottom. In the revised manuscript, the T_m point of the bottom figure is marked as its actual position (~125 °C) and mentioned at the Fig. 2a caption. For the second comment, please see our responses to questions 1,

2 and 3 of the reviewer 2. More careful analysis based on TGA, DSC and FT-IR spectra has been done, and a more appropriate explanation has been given and highlighted on page 7 and 23 of the revised manuscript. Lastly, we reorganized that two-paragraph discussion on thermal behaviors in page 6 of original manuscript. New version of which is presented as two paragraphs, with the first for the component and stability discussion and the followed paragraph for DSC analysis, as shown in page 7 of revised manuscript as:

“(Supplementary Figs. 8a, 11). Notably, no melting or evaporating event of MSA ($T_m \approx 20^\circ\text{C}$, $b_p \approx 167^\circ\text{C}$) was recorded in the DSC and TGA trace before 180°C . No loss of MSA implies that the Brønsted acid-zwitterion subsystem MIMS·MSA has been successfully incorporated in **ZW-UiO-67·MSA** (Supplementary Fig. 1). This is similar to the scenario for the zwitterion-based acidic ionic liquids^[25] including H₂BPDC-MIMS·MSA which shows no thermal event of a separate specie upon heating (Supplementary Fig. 6d). Therefore, the behavior of **ZW-UiO-67·MSA** is in stark contrast to the thermal properties of neat **ZW-UiO-67** or MSA alone. It can be viewed as a solid ionic liquid^[29] featuring the Brønsted acid-zwitterion structure, but not a simple host-guest composite like those ILs@MOFs^[27].

For the DSC plot within $0\sim 320^\circ\text{C}$, the first feature of the endothermal melting (offset $\sim 127^\circ\text{C}$) is broad and weak, which could be related to the lattice fluctuation with multiple and gradual enthalpy changes, being similar to the melts of some porous organic cages with flexible structures upon heating^[30]. The measured fusion enthalpy change of 6.35 J/g is also comparable to those of 2D metal-triazolates ($5\text{-}10\text{ kJ/mol}$)^[14], zeolitic zinc 2-ethylimidazole^[31] and some silica zeolites^[32]. The following exothermal event with onset at 162.5°C (without weight loss) may be ascribed to polyamorphic phase transitions between distinct structural configurations of neighbor states with different potential energies (Fig. 2a, Supplementary Fig. 5). Therefore, the melting phase of **ZW-UiO-67·MSA** before 162.5°C can be regarded as a stable liquid phase I. At 185.4°C , the endothermal sharp peak closely relates to that event for **ZW-UiO-67** at 180.0°C and **ZW-UiO-67·0.5MSA** at 186.2°C , respectively (Supplementary Fig. 5). Considering the formula mass percentage of acetate (OOCCH_3 wt% = 1.4% in **ZW-UiO-67·MSA**) is close to the $\sim 1.8\text{wt}\%$ weight loss recorded before 250°C , it could be speculated that the endothermal event without weight loss at 185.4°C may relate to the de-association of the acetate group which is followed by less composite decomposition after that temperature. Online gas-phase FT-IR spectra demonstrate that H₂O and CO₂ can be detected with the temperature increase (25°C to 320°C) (Methods and Supplementary Fig. 25).”

4. The authors said “the following exothermal and another event with onset 256.7°C may be polyamorphic phase transitions from the low-density liquid (LDL) to high-density liquid (HDL) driven by different potential energies between distinct structural configurations of neighbor states”. However, comparing to Ref-5, the signals at 256.7°C shows completely different characters. The dramatic exothermic trends are really strange. Without sufficient and credible evidence, such statement of LDL and HDL is improper.

Response: We thank the reviewer for this critical comment. Please see our response to question 2 from Reviewer 2.

5. In page 5, the formular for **ZW-UiO-67·MSA** seems incorrect, the BPDC should contain the grafted MIMS. In addition, why 1MSA and 0.5MSA per **ZW-UiO-67** in page 7. Considering there

are a numbers of compounds in the paper, a list showing all the abbreviation, accurate formular in necessary for readership.

Response: The formula for ZW-UiO-67·MSA has been changed to $Zr_6O_4(OH)_4(BPDC-MIMS)_{5.4} \cdot (O_2CCH_3)_{1.2} \cdot (MSA)_{20.5}$. In addition, we acknowledge the importance of providing a comprehensive list of abbreviations and accurate formulas for all compounds mentioned in the paper. To address this, we have included a dedicated Supplementary Table 2 that lists all abbreviations used for the complexes discussed throughout the manuscript.

Supplementary Table 2 | Compositions of ZW-UiO-67, ZW-UiO-67·0.5MSA, ZW-UiO-67·HA, a_{gr} ZW-UiO-67·MSA, and EIMS·MSA@UiO-67. HA = MSA, TFSA, TFA, ESA.

	ZW-UiO-67 (mg)	HA (μ l)	Stoichiometry Formula ^a (L = BPDC-MIMS)	Formula weight	ZrO ₂ (wt%) ^b Calc./meas.	
ZW-UiO-67	/	/	$Zr_6O_4(OH)_4(L)_{5.4} (O_2CCH_3)_{1.2}$	~3124.8	~23.7/27.6	
ZW-UiO-67·MSA	80	34	$Zr_6O_4(OH)_4(L)_{5.4} \cdot (O_2CCH_3)_{1.2} \cdot (MSA)_{20.5}$	~5092.8	~14.5/15.7	
ZW-UiO-67·0.5MSA	80	17	$Zr_6O_4(OH)_4(L)_{5.4} \cdot (O_2CCH_3)_{1.2} \cdot (MSA)_{10.2}$	~4104.0	~18.0/20.1	
ZW-UiO-67·TFSA	80	34	$Zr_6O_4(OH)_4(L)_{5.4} \cdot (O_2CCH_3)_{1.2} \cdot (HTFSA)_6$	~4923.2	~15.0/16.0	
ZW-UiO-67·TFA	80	34	$Zr_6O_4(OH)_4(L)_{5.4} \cdot (O_2CCH_3)_{1.2} \cdot (TFA)_{15.0}$	~5374.8	~13.8/14.1	
ZW-UiO-67·ESA	80	34	$Zr_6O_4(OH)_4(L)_{5.4} \cdot (O_2CCH_3)_{1.2} \cdot (ESA)_{16.3}$	~4916.6	~15.0/26	
EIMS·MSA@UiO-67	UiO-67 (80 mg) & ILs (81 μ L)		$Zr_6O_4(OH)_4(BPDC)_{5.4} \cdot (O_2CCH_3)_{1.2} \cdot [EIMS]_{4.93} [MSA]_{18.7}$	~4935.7	~15.0/8.73	
a_{gr}ZW-UiO-67·MSA	/	/	$Zr_6O_4(OH)_4(L)_{5.4} \cdot (O_2CCH_3)_{1.2} \cdot (MSA)_{12.3}$	~4305.6	Unmeasured	
HA or ILs used in this study						
HA/ILs^c	EIMS·MSA	EMIM·TFA	MSA	TFSA ^d	TFA	ESA
ρ (g/ml)	1.42	1.38	1.48	1.892	1.696	1.35

^aStoichiometry formula is based on ZW-UiO-67 which was determined through ¹H NMR (Supplementary Fig. 2), and volume of HA or ionic liquid equals to the pore of ZW-UiO-67 used.

^b Data is obtained from the TGA traces (Supplementary Fig.16). ^c Ionic liquids (ILs) were synthesized as that illustrated in Supplementary Fig. 1. ^d Melt point of TFSA (bis(trifluoromethane)sulfonimide) is 52 °C, here used as a liquid under 55 °C.

6. For the description of proton conduction in Page 7, the reported data is not clear for which sample.

Response: We are sorry for this misunderstanding. The reported data are for a_gZW-UiO-67·MSA, which has been explicitly mentioned in page 8-9 of the revised manuscript.

“...The anhydrous proton conductivity of a_gZW-UiO-67·MSA reaches $1.57 \times 10^{-2} \text{ Scm}^{-1}$ at 100 °C, acting as a fast ionic conductor with application prospect^[29]. Almost one order-of-magnitude increase in conduction from 1.96×10^{-3} to $1.65 \times 10^{-2} \text{ Scm}^{-1}$ was observed as the temperature elevated from 80 to 110 °C. Such a marked change in conduction correlates to the glass transition near $T_g (> T_g, 104 \text{ °C})$, reminiscent of that observed for the phase transition in reported solid state conductors^[33]...”.

7. For the recovery experiments, how about the solvent-dependent? Excepting methanol, can other solvent induce the glass-to-crystalline transformation. In addition, to maintain the molding film, is vapor possible to facilitate the recovery?

Response: Thanks for your professional comments and insightful suggestion. We appreciate your interest in the solvent-dependent recovery experiments. Based on our research, we have observed partial recovery of the low 2θ degree X-ray diffraction for a_gZW-UiO-67·MSA when exposed to methanol vapor for 30 minutes. In response to your question, we have also extended our investigations to include other polar and non-polar solvents such as ethanol, water, ethyl acetate, dichloromethane, and acetone. Our findings suggest that a polar solvent is more beneficial for the recovery process. Related updated Supplementary Fig. 15 demonstrates the reappearance of high-degree X-ray diffractions after immersing the samples in the solvent for 20 minutes with stirring at room temperature. The revised manuscript now includes a detailed discussion on this topic, specifically on page 8. Please also see that response to question 6 of reviewer 2 above.

8. It seems enough content of incorporating organic acids is crucial for the success of melting, how about the critical point for the three MOFs?

Response: The reviewer is right. Sufficient amounts of organic acids are critical, as illustrated by the difference in thermal behavior between ZW-UiO-67·MSA and ZW-UiO-67·0.5MSA. Regarding the critical points of the three MOFs, in this study we mainly focused on exploring the addition of same volume of organic acids as the pores of MOFs. To fully characterize the melting of a series of incorporated MOF·HA compounds, more structure determinations need to be involved, including X-ray absorption near edge structure (XANES) spectra, total scattering measurements, morphology and thermal behavior characterizations. Due to the resource and time constraints, we regret to say that our revision is unable to include all the demanding experiments in this context. However, we totally agree with the reviewer that this is a very worthwhile direction to explore as well as investigating the potential linear relationship between organic acid content and T_m/T_g . We will try to look into this in depth in future studies.

9. On the whole, the organization of different parts as well as discussion in the same part is not sufficiently good, which leads to easy lost of main idea of the authors. Though it may be ascribed to the large experimental data, further improvement on the organization and language is necessary.

Response: As the reviewer suggested, we have made several revisions to improve the organization and language of the paper. Specifically, we have reorganized certain paragraphs, such as the

component and stability section, as well as the thermal behavior discussion on page 7. Additionally, we have addressed the recovery phenomenon and conducted a more comprehensive study of the glass on page 8. Furthermore, we have revised our explanation of the melting mechanism, providing a clearer elucidation on page 9. We believe that these revisions contribute to a more effective communication of our research findings.

Reviewer #4 (Remarks to the Author):

There is a great deal of current interest in the new family of glasses derived from metal organic frameworks, but most of the ones that have been reported to date are based on organic linkers that bind to metals via nitrogen. For example, the most widely studied MOF glasses are those based on zeolites imidazolate frameworks (ZIFs). The manuscript submitted by Yue et al describes an ingenious way of making glasses from MOFs based upon carboxylate linkers, which constitute the largest family of MOFs. This work will therefore be of considerable interest to the MOF community. The new carboxylate glasses are mainly formed by starting with derivatives of MOF UiO-67 that have been functionalized by substituting zwitterionic amines onto the aromatic linkers and then introducing a complementary Brønsted acid, such as methylsulfonic acid, into the MOF cavity. The amine substituent and the Brønsted acid form zwitterionic pairs that facilitate the low temperature melting of the MOF. The melts can then in turn be quenched to yield glasses. One of the interesting features of these glasses is that the crystalline MOF precursor can be retrieved from the glass by leaching the acid out of the glass by using a simple organic solvent such as methanol. This reversibility is not usually found in ZIF glasses.

The characterisation of the MOF glasses in the manuscript is very thorough, even though the structures of the crystalline material precursors depend upon powder X-ray diffraction data. The use of other techniques, such as EXAFS, total scattering XRD, vibrational spectroscopy and solid state NMR, has been used to probe the local structures of the glasses and compare them to the crystalline precursors. Overall, I think that the work has been carried out very carefully and that the claims of glass formation are well supported by the data.

I think that this interesting work could be published in Nature Communications after a couple of important points have been addressed:

Response: We thank the reviewer for these enthusiastic comments on the significance of our work.

1. The authors claim that they have developed a universal strategy for making glasses from carboxylate MOFs (see pages 3 and 14). I think that this claim is exaggerated and needs to be toned down. For example, the MOF cavities need to be large enough to accommodate both the zwitterionic amine substituents AND the complementary Brønsted acid molecules. This approach can only therefore be used for MOFs with large cavities such as the UiO family. Carboxylate MOFs with smaller cavities, of which there are very many, will be unsuitable for this approach. The claim of universality must be replaced by a more suitable and justifiable claim.

Response: We agree with the reviewer that we should tone down the applicability of our strategy. We have reorganized our statements to make our claim more appropriate, such as the modified article title “Melt-Quenched Glass Formation of a Family of Metal-Carboxylate Frameworks” and the revised wording “By functionalizing the carboxylate ligands with zwitterionic groups and incorporating the Brønsted acids in the framework channels, a family of metal-carboxylate frameworks (UiO-67, UiO-68 and DUT-5) were successfully made into glasses via the melt-quenched method...” in the Discussion of the last paragraph of the revised manuscript.

2. Most of the work concerned MOFs from the UiO family and these are well-known to be prone to having large concentrations of defects in the form of missing octahedra (from the work of Goodwin).

This should be mentioned in the manuscript since these large defect vacancies may play a role in accommodating the amine substituents and the Brønsted acid molecules. I also wonder if any of the characterisation methods reveal any evidence for the presence of defects in the crystalline precursors.

Response: According to Goodwin's work (*Nat. Commun.* 2014, **5**, 4176), we have thoroughly examined our WAXS data for crystalline ZW-UiO-67 (Supplementary Fig. 27b). The analysis reveals that the fluctuations between 2.5 and 5 degrees (2θ) are nearly negligible, suggesting that cluster defects may be of minimal significance (*Nat. Chem.* 2019, **11**, 622–628). Furthermore, ZW-UiO-67 exhibits a structure with ligand-dominated defects, which has been characterized using ^1H NMR spectra. The stoichiometric formula of ZW-UiO-67 is denoted as $\text{Zr}_6\text{O}_4(\text{OH})_4(\text{BPDC-MIMS})_{5.4}$, with a portion of the BPDC-MIMS ligand being displaced by an acetate anion. Although we cannot dismiss the possibility of a small number of cluster defects in ZW-UiO-67, we believe that the majority of defects can be attributed to ligand defects identified through ^1H NMR, and such defects are not responsible for the meltability of ZW-UiO-67·HA because control ILs@ ZW-UiO-67 is unmeltable. According discussion on the ligand-based defect has been included in the middle of page 5 of the revised version as the following:

“...The stoichiometric formula of $\text{Zr}_6\text{O}_4(\text{OH})_4(\text{BPDC-MIMS})_{5.4}$ is determined via ^1H NMR spectra, with a small portion of BPDC-MIMS ligand displaced by the acetate group...”

Supplementary Fig. 27b. PXRD patterns of WAXS of crystalline ZW-UiO-67, amorphous $a_g\text{ZW-UiO-67}\cdot\text{HA}$, polycarbonate and silicate glass.

If these two points are properly addressed, I think that the paper could be accepted for publication in Nature Communications.

Reviewer #1 (Remarks to the Author):

The authors have addressed all the issues, and thus I recommend it be accepted for publish without further change.

Response: We thank the reviewer for the very positive comment.

Reviewer #2 (Remarks to the Author):

Q1

Point-by-point revisions by the author are highly welcome. The author has added new citations for some metal-carboxylate framework glasses. Additional discussions for metal-carboxylate framework glasses have been added. But does the ambiguous naming of "Several melt-quenched metal complexes" give the impression that the authors are deliberately leading readers to ignore closely related published works? After reading these papers, I firmly believe that they are carboxylate MOF glasses. In addition, I have noticed that another paper based on porous dicarboxylate-bridged MOF glasses has just been published (Chem. Commun., 2023, 10.1039/D3CC04518H). The authors need to make appropriate discussions and revisions.

Response: We agree with the reviewer we need to extend our discussions on the recently developed metal-carboxylate MOF glasses. We have cited the mentioned paper as reference 24. Also, related discussions have been reorganized on page 3 in the revised manuscript as the following:

“...Several carboxylate MOFs, including both carboxylates and bicarboxylates, have been made into glasses very recently, which are derived from discrete solvated complexes or coordination networks through rearrangements of coordination bonds upon desolvation^[22-24]...”

Q2

In addition, agrZW-UiO-67-MSA was obtained by refluxing agrZW-UiO-67-MSA in methanol. The current discussion, which is mainly based on PXRD, cannot support that agZW-UiO-67-MSA is a UiO-67-type framework. That is, it cannot be completely ruled out that the material gradually dissolves and recrystallizes in the refluxing methanol. In fact, it can be seen from the blurred SEM in Fig. 2b that agrZW-UiO-67-MSA are nanocrystals, quite different from that of agZW-UiO-67-MSA. The authors need to provide clear SEMs to study the morphological evolution of the particles and surfaces of agrZW-UiO-67-MSA to agZW-UiO-67-MSA.

Response: We are afraid that the reviewer confused the glass (a_gZW-UiO-67-MSA) and recovered crystalline phase (a_{gr}ZW-UiO-67-MSA). Firstly, the structure of the a_gZW-UiO-67-MSA glass was characterized by EXAFS, PDF, solid NMR and FT-IR spectra and *ab initio* molecular dynamics (AIMD) simulations etc, which showed the distorted Zr₆O₄(OH)₄ nodes with some broken Zr-O_{COO} bonds in the *syn-syn-μ₂* COO bridges. Such a defected structure with continue random network (CRN) is reminiscent of the scenarios in structurally similar silicates and reported MOF based glasses (*Chem. Rev.* 2022, 122, 4163). Additional variable-temperature in-situ FT-IR of ZW-UiO-67-MSA also can prove the network of a_gZW-UiO-67-MSA resemble the UiO-67-type structure in ZW-UiO-67-MSA. Please also see the following response for Q4.

The reviewer mentioned ‘...material gradually dissolves and recrystallizes in the refluxing methanol’. In fact, a_gZW-UiO-67-MSA is indissolvable in methanol, which is shown in Supplementary Fig. 5a. So, soaking the as-made a_gZW-UiO-67-MSA in methanol with stirring at

room temperature for 3 minutes led to high yield (97%) of $a_{gr}ZW-UiO-67 \cdot MSA$ powder with partially recovered Bragg peaks, as shown in Supplementary Fig. 4a and Method Section in Page 29. The less weight loss may be due to the departure of some methylsulfonate acid MSA. To further rule out the possible recrystallization of $a_{gr}ZW-UiO-67 \cdot MSA$ from the gradually dissolving of $a_gZW-UiO-67 \cdot MSA$, the Soxhlet Extractor method was used. The results showed the $a_{gr}ZW-UiO-67 \cdot MSA$ product with partially recovered Bragg diffractions (concentrated between 5 and 7.5°, Supplementary Fig. 15 as shown below). Combined the stimulation of methanol vapor and other kinds of solvents and the EXAFS, PDF, solid-state NMR and FT-IR spectra, it can be concluded that $a_gZW-UiO-67 \cdot MSA$ is a UiO-67-type framework and $a_{gr}ZW-UiO-67 \cdot MSA$ inherited the $a_gZW-UiO-67 \cdot MSA$ structure which is distorted and can be recovered under external stimulation.

Furthermore, the blurred SEM images in Fig. 2b have been updated with better resolution. According to the reviewer's comment, cross-sectional SEM images of $a_gZW-UiO-67 \cdot MSA$, $a_{gr}ZW-UiO-67 \cdot MSA$ obtained from short-time (3 min) soaking in methanol at room temperature and refluxing methanol (3 h) are shown in Supplementary Figs. 10d-f. In addition, the morphological evolution from $a_gZW-UiO-67 \cdot MSA$ to $a_{gr}ZW-UiO-67 \cdot MSA$ can be clearly visualized in from Supplementary Figs. 10c-f.

Supplementary Fig. 15 | PXRD patterns of $a_{gr}ZW-UiO-67 \cdot MSA$ after solvent stimulations in comparison.

Products were obtained via soaking $a_gZW-UiO-67 \cdot MSA$ in different solvents with stirring at room temperature for 20 minutes. Compared to the patterns of the sample soaked of 3 minutes in methanol (Supplementary Fig. 4), peaks at high 2θ degree appear for those of 20 minutes in various solvents. Methanol vapor stimulation is done by exposing sample to vapor for 30 minutes. Soxhlet Extractor method with methanol as solvent is used to rule out the possibility of $a_{gr}ZW-UiO-67 \cdot MSA$ from the gradually dissolving $a_gZW-UiO-67 \cdot MSA$ and recrystallization. Except for the effect of different solvents, the different crystalline phases of $a_{gr}ZW-UiO-67 \cdot MSA$ can also be explained by the effect of different batches, including distinct composites obtained via the incipient wetness technique, uncertain portion departure of MSA and different conditions such as with or without stirring etc.

Supplementary Fig. 10 | SEM images. a, ZW-UiO-67. b, ZW-UiO-67-MSA. c, a_gZW-UiO-67-MSA. d, Cross-sectional image of a_gZW-UiO-67-MSA. PTFE polymer is used as support. e, a_gZW-UiO-67-MSA obtained via soaking in methanol with stirring at room temperature. f, a_gZW-UiO-67-MSA resulted from refluxing methanol. g, a_gZW-UiO-67-TFSA. h, a_gZW-UiO-67-TFA. The different false colors are shown for clarity.

Q3

The authors also found that other solvents are advantageous for the recrystallization process. However, PXRD showed that these solvents caused agZW-UiO-67-MSA to crystallize into a different phase than grZW-UiO-67-MSA. The authors need to try to resolve and discuss this crystalline phase.

Response: We thank the reviewer for this professional comment about the recrystallization process. It is true that other solvents can cause recrystallization as the reviewer mentioned, however, it is somewhat different from the case of methanol (Supplementary Fig. 15). After being stimulated by other solvents, the treated ag_{gr}ZW-UiO-67-MSA tends to show diffraction peaks preferentially at higher angles (19°–25°); while with methanol, the treated ag_{gr}ZW-UiO-67-MSA tends to have diffraction peaks at low angles (5°–12°). The presence of low-angle diffraction peaks could imply a better formation of the MOF framework. This could be attributed to the fact that methanol has the optimal molecular volume, polarity, and affinity for recovering this family of MOF glasses into their crystalline parents. Nevertheless, we also recognize that in addition to the effect of different solvents, the different re-crystalline behaviours of ag_{gr}ZW-UiO-67-MSA can also be explained by the effect of different sample batches, including distinct composites obtained *via* the incipient wetness technique, uncertain portion departure of MSA during stimulation and different conditions such as with or without stirring etc.

Furthermore, in order to address the attribution of peaks emerging from other solvent stimulation, we have tried to Pawley fit the PXRD patterns of DCM stimulated sample (please see the following Fig. R1), which has the highest Bragg intensity in Supplementary Fig. 15 compared to those from other samples. The peak positions and the intensities can be reasonably fitted by the lattice parameters of ZW-UiO-67. We therefore consider the emerged phase could be ZW-UiO-67 with low crystallinity. However, the crystallinity is so low here that we cannot define the exact phase situation. But this is a very interesting phenomenon and we will explore this topic in depth in subsequent work.

For a more accurate description, we have made the following changes to the explanation in page 8 of the new version:

“...Furthermore, the glass to crystalline phase transformation upon external stimulation can be extended to other kinds of solvents, and polar solvents are advantageous for this process (Supplementary Fig. 15).”

Supplementary Fig. 15 | PXRD patterns of a_{gr} ZW-UiO-67-MSA after solvent stimulations in comparison.

Products were obtained via soaking a_{gr} ZW-UiO-67-MSA in different solvents with stirring at room temperature for 20 minutes. Compared to the patterns of the sample soaked of 3 minutes in methanol (Supplementary Fig. 4), peaks at high 2θ degree appear for those of 20 minutes in various solvents. Methanol vapor stimulation is done by exposing sample to vapor for 30 minutes. Soxhlet Extractor method with methanol as solvent is used to rule out the possibility of a_{gr} ZW-UiO-67-MSA from the gradually dissolving a_{gr} ZW-UiO-67-MSA and recrystallization. Except for the effect of different solvents, the different crystalline phases of a_{gr} ZW-UiO-67-MSA can also be explained by the effect of different batches, including distinct composites obtained via the incipient wetness technique, uncertain portion departure of MSA and different conditions such as with or without stirring etc.

Fig. R1 | Pawley fit of a_{gr} ZW-UiO-67-MSA (DCM). The crystal parameters taken from ZW-UiO-67.

Q4

I am curious if the coordination structure is maintained after melting? In situ analyses by PDF, XAFS, FT-IR, etc. may provide evidence. Investigation of the structure after liquefaction helps to

understand what the real structure of agZW-UiO-67-MSA is.

Response:

We thank the reviewer for the valuable suggestion. For convenience, variable temperature in-situ FT-IR of ZW-UiO-67-MSA was conducted using a VERTEX 70 spectrometer with the ATR imaging system. All IR peaks of a_gZW-UiO-67-MSA were maintained after melting. Additional discussion has been added at the bottom of Page 14 as the following:

“...Variable temperature in-situ FT-IR data of ZW-UiO-67-MSA show the gradually broadening and shifting of C=O band from 1693 at room temperature to 1713 cm⁻¹ at 130 °C. However, the ~1693 cm⁻¹ band feature remained in the melting ZW-UiO-67-MSA and the melt-quenched glass a_gZW-UiO-67-MSA (Supplementary Fig. 27). This phenomenon further indicates the coordination network in liquids state which quenched to form the glass structure as mentioned above”

Supplementary Fig. 27 | Variable temperature in-situ FT-IR of ZW-UiO-67-MSA. The right panel of the spectra from 1300 cm⁻¹ to 1800 cm⁻¹ range, with the guiding eye lines to show the change of the carboxylate C=O band.

Reviewer #3 (Remarks to the Author):

After carefully reading the authors response and revised manuscript, most of my previous queries have been well addressed, based on some complementary experiments and further analysis of all the observed results. I also happy to see the organization, writing and quality of the paper have been improved, benefited from the comments of other reviewers. I personally think the current version is suitable for publication in Nat. Commun.

Response: We thank the reviewer for the positive comment.

Reviewer #4 (Remarks to the Author):

I am pleased that the authors have addressed the two important points made in my original review and I am satisfied with the changes that they have made in response to them. I also feel that they

have responded effectively to the comments made by the other reviewers. I am therefore happy to recommend publication of the revised manuscript in Nature Communications.

Response: We thank the reviewer for the enthusiastic comment on this work.

Reviewer #2 (Remarks to the Author):

I am very pleased with the authors' excellent and detailed responses to my comments and those of the other two reviewers. I appreciate the way they have revised the paper and believe it will provide an important breakthrough for future work in glass chemistry and other fields.

Response: We thank the reviewer for the enthusiastic and positive comments on this work.